# UNIFORM LOCALIZED CONVERGENCE AND SHARPER GENERALIZATION BOUNDS FOR MINIMAX PROBLEMS

## ABSTRACT

Minimax problems have achieved widely success in machine learning such as adversarial training, robust optimization, reinforcement learning. Existing studies focus on minimax problems with specific algorithms in stochastic optimization, with only a few work on generalization performance. Current generalization bounds almost all depend on stability, which need case-by-case analyses for specific algorithms. Additionally, recent work provides the $O(\sqrt{d/n})$ generalization bound in expectation based on uniform convergence. In this paper, we study the generalization bounds measured by the gradients of primal functions using the uniform localized convergence. We relax the Lipschitz continuity assumption and give a sharper high probability generalization bound for nonconvex-strongly-concave (NC-SC) stochastic minimax problems considering the localized information. Furthermore, we provide dimension-independent results under Polyak-Lojasiewicz condition for the outer layer. Based on the uniform localized convergence, we analyze some popular algorithms such as the empirical saddle point (ESP), gradient descent ascent (GDA) and stochastic gradient descent ascent (SGDA) and improve the generalization bounds for primal functions. We can even gain approximate $O(1/n^2)$ excess primal risk bounds with further assumptions that the optimal population risks are small, which, to the best of our knowledge, are the sharpest results in minimax problems.

## 1    INTRODUCTION

Modern machine learning settings such as reinforcement learning (Du et al., 2017; Dai et al., 2018), adversarial learning (Goodfellow et al., 2016), robust optimization (Chen et al., 2017; Namkoong & Duchi, 2017) often need to solve minimax problems, which divide the training process into two groups: one for minimization and one for maximization. To solve the problems, various efficient optimization algorithms such as gradient descent ascent (GDA), stochastic gradient descent ascent (SGDA) have been proposed. Most of them were focused on the iteration complexity, which only considered the optimization error. In contrast, the generalization performance analysis is less considered, which is an important measure to foresee their prediction behavior after training.

Recently, Zhang et al. (2022) introduced an expectation generalization error for primal functions in minimax problems using complexity. Naturally, we want to create a high-probability version, preferably using local methods to introduce variance information and obtain a tighter upper bound. A straightforward idea is that we can continue with the traditional localized approach and solve the problem with covering numbers Bartlett et al. (2002). However, these technologies require additional bounded assumptions (Assumption 2), or need certain distributional assumptions for unbounded condition. For example, Mei et al. (2018) introduced the "Hessian statistical noise" assumption when using covering numbers. Fortunately, Xu & Zeevi (2020) developed a novel "uniform localized convergence" framework using generic chaining for the minimization problems and Li & Liu (2021b) extended it to analyze stochastic algorithms.

This novel framework can not only relax the bounded (or specific distribution) assumptions but also impose fewer restrictions on the surrogate function for the localized method, enabling us to design the measurement functional to achieve a sharper bound. Consequently, we introduce this remarkable framework into minimax problems. Our generalization bound uses weaker assumptions

comparing with Zhang et al. (2022) and is sharper in some conditions due to our utilization of variance information.

Introducing this new framework into minimax problems is not straightforward. Zhang et al. (2022) indeed established a connection between inner and outer layers with the loss of primal functions, but we need do this with a new generic chaining approach. Furthermore, while Zhang et al. (2022) only needed to bound the error caused by the connection between inner and outer layer with $O(1/\sqrt{n})$. We need to introduce the variance for a sharper bound, to bound the error involved by the two layers.

Next, we turn to applications. Firstly, for a sharper excess risk bound, we need to introduce the PL-SC condition to establish a connection between excess risk and the gradient of primal functions, leading to our results for ESP. Unfortunately, for GDA and SGDA algorithms, we found that all the related optimization papers in minimax problems focused on the iteration complexity (or gradient complexity). As a result, they only require $\mathbb{E}[\|\nabla\Phi(\mathbf{x}_T)\|]$ for average of round $T$ with an expected outcome. We were compelled to derive the high probability empirical optimization bound ourselves using classical optimization methods under SC-SC conditions.

Notice that even under SC-SC settings, achieving this for SGDA remains difficult. Drawing inspiration from Lei et al. (2021)'s proof of Primal-Dual Risk optimization bound, we eventually derive the optimization bound for primal risk. The proofs for excess risks in applications differ from minimization problems Li & Liu (2021b) and pose challenges. These challenges stem from errors in the inner and outer layers of minimax problems. Consequently, we can only achieve a result close to $O(1/n^2)$. Our contributions are summarized as follows:

1. We introduce local uniform comvergence using new generic chaining techniques. Comparing with traditional uniform convergence results in Zhang et al. (2022), we derive sharper generalization bounds measured by the gradients of primal functions for NC-SC minimax problems. It provides problem independent results that can be used in various minimax algorithms.

2. Under the Polyak-Lojasiewicz condition for the outer layer, we provide dimension-independent results and remove the dimension of parameters $d$ from our generalization bound when the sample size $n$ is large enough, which is, to our knowledge, the first result in minimax problems.

3. We extend our main theorems into various algorithms such as ESP, GDA, SGDA. We establish faster $O(1/n)$ order bounds for excess primal risk. We can even gain approximate $O(1/n^2)$ bounds with further assumptions that the optimal population risk is small. To our best knowledge, it is the first time to gain approximate $O(1/n^2)$ for NC-SC minimax problems in expectation and the first result nearly to $O(1/n^2)$ high probability bound for SC-SC settings.

This paper is organized as follows. In Section 2, we review the related work. In Section 3, we introduce the notations and assumptions about the problems. Section 4 presents our main results. Then we apply our main theorems into various algorithms and give the sharper bounds for different settings in Section 5. Section 6 concludes our paper. All the proofs in our paper are given in Appendix.

## 2 RELATED WORK

**Minimax optimization.** Minimax optimization analysis has been widely studied in different settings. For example, one of the most popular SGDA algorithm and its variants have been analyzed in several recent works including Palaniappan & Bach (2016); Hsieh et al. (2019) for SC-SC cases, Nedić & Ozdaglar (2009); Nemirovski et al. (2009) for convex-concave (C-C) cases, Lin et al. (2020); Luo et al. (2020); Yan et al. (2020); Rafique et al. (2022) for NC-SC problems, Thekumparampil et al. (2019); Yan et al. (2020) for nonconvex-concave (NC-C) cases and Loizou et al. (2020); Liu et al. (2021); Yang et al. (2020) for nonconvex-nonconcave (NC-NC) minimax optimization problems. All these works focus on the iteration complexity (or the gradient complexity) of the algorithms, which only proved the optimization error bounds for the sum of $T$ iteration's gradient of primal empirical function in expectation. Recently Li & Liu (2021a); Lei et al. (2021) gave optimization bounds with high probability for Primal-Dual risk. We notice that the optimization error of the gradients of primal functions with high probability haven't been studied yet.

**Algorithmic stability.** Algorithmic stability is a classical approach, which was presented by Rogers & Wagner (1978). It gives the generalization bound by analyzing the sensitivity of a particular

learning algorithm when changing one data point in the dataset. Modern framework of stability analysis was established by Bousquet & Elisseeff (2002), where they present an important concept called uniform stability. Since then, a lot of works based on uniform stability have emerged. On the one hand, the generalization bound with algorithmic stability have been significantly improved by Bousquet et al. (2020); Feldman & Vondrak (2018; 2019); Klochkov & Zhivotovskiy (2021). On the other hand, different algorithmic stability measures such as uniform argument stability (Liu et al., 2017; Bassily et al., 2020), on average stability (Shalev-Shwartz et al., 2010; Kuzborskij & Lampert, 2018), collective stability (London et al., 2016) have been developed. For minimax problems, many useful stability measures have also been extended, for example, weak stability (Lei et al., 2021), argument stability (Lei et al., 2021; Li & Liu, 2021a), and uniform stability (Lei et al., 2021; Li & Liu, 2021a; Zhang et al., 2021; Farnia & Ozdaglar, 2021; Ozdaglar et al., 2022). Most of them focused on the expectation generalization bounds and only Lei et al. (2021); Li & Liu (2021a) established some high probability bounds.

**Uniform convergence.** Uniform convergence is another popular approach in statistical learning theory to study generalization bounds (Fisher, 1922; Vapnik, 1999; Van der Vaart, 2000). The main idea is to bound the generalization gap by its supremum over the whole (or a subset) of the hypothesis space via some space complexity measures, such as VC dimension, covering number and Rademacher complexity. For finite-dimensional problem, Kleywegt et al. (2002) provided that the generalization error is $O(\sqrt{d/n})$ depended on the sample size $n$ and the dimension of parameters $d$ in high probability. For nonconvex settings, Mei et al. (2018); Davis & Drusvyatskiy (2022) showed that the empirical of generalization error is $O(\sqrt{d/n})$. Xu & Zeevi (2020) developed a novel "uniform localized convergence" framework using generic chaining for the minimization problems and Li & Liu (2021b) extended it to analyze stochastic algorithms. In minimax problems, Zhang et al. (2022) established the first uniform convergence and showed that the empirical generalization error of the gradients for primal functions is $O(\sqrt{d/n})$ under NC-SC settings.

## 3 PRELIMINARIES

Let $\mathcal{X} \in \mathbb{R}^d$ and $\mathcal{Y} \in \mathbb{R}^{d'}$ be two nonempty closed convex parameters spaces. Let $\mathbb{P}$ be a probability measure defined on a sample space $\mathcal{Z}$. We consider the following minimax optimization problem

$$\min_{\mathbf{x} \in \mathcal{X}} \max_{\mathbf{y} \in \mathcal{Y}} F(\mathbf{x}, \mathbf{y}) := \mathbb{E}_{\mathbf{z} \sim \mathbb{P}}[f(\mathbf{x}, \mathbf{y}; \mathbf{z})], \tag{1}$$

where $f : \mathcal{X} \times \mathcal{Y} \times \mathcal{Z} \to \mathbb{R}$ is continuously differentiable and Lipschitz smooth jointly in $\mathbf{x}$ and $\mathbf{y}$ for any $\mathbf{z}$. This above minimax objective called as the population minimax problem represents an expectation of a cost function $f(\mathbf{x}, \mathbf{y}; \mathbf{z})$ for minimization variable $\mathbf{x}$, maximization variable $\mathbf{y}$ and data variable $\mathbf{z}$. In this paper, we focus on the NC-SC problem which means that $f$ is nonconvex in $\mathbf{x}$ and strongly concave in $\mathbf{y}$. Obviously, our goal is to gain the optimal solution $(\mathbf{x}^*, \mathbf{y}^*)$ to (1). Since the distribution $\mathbb{P}$ is unavailable, we can only gain a dataset $S = \{\mathbf{z}_1, \ldots, \mathbf{z}_n\}$ drawn $n$ times independently according to $\mathbb{P}$. Therefore, we solve the following empirical minimax problem instead

$$\min_{\mathbf{x} \in \mathcal{X}} \max_{\mathbf{y} \in \mathcal{Y}} F_S(\mathbf{x}, \mathbf{y}) := \frac{1}{n} \sum_{i=1}^{n} f(\mathbf{x}, \mathbf{y}; \mathbf{z}_i). \tag{2}$$

Next we introduce one of the common measures in minimax problems called primal functions.

**Definition 1** (primal function (empirical/population)). *The primal population function and the primal empirical function are given by*

$$\Phi(\mathbf{x}) := \max_{\mathbf{y} \in \mathcal{Y}} F(\mathbf{x}, \mathbf{y}) \quad and \quad \Phi_S(\mathbf{x}) := \max_{\mathbf{y} \in \mathcal{Y}} F_S(\mathbf{x}, \mathbf{y}).$$

Since $F$ and $F_S$ are nonconvex in $\mathbf{x}$, it is difficult to find the global optimal solution in general. In practice, we design an algorithm $\mathcal{A}$ that finds an $\epsilon$-stationary point

$$\|\nabla\Phi(\mathcal{A}_{\mathbf{x}}(S))\| \le \epsilon, \tag{3}$$

where $\mathcal{A}_{\mathbf{x}}(S)$ is the $x$-component of the output using any algorithm $\mathcal{A}(S) = (\mathcal{A}_{\mathbf{x}}(S), \mathcal{A}_{\mathbf{y}}(S))$ for solving (2). Then the optimization error for solve the population minimax problem (1) can be decomposed into two terms:

$$\|\nabla\Phi(\mathcal{A}_{\mathbf{x}}(S))\| \le \|\nabla\Phi_S(\mathcal{A}_{\mathbf{x}}(S))\| + \|\nabla\Phi(\mathcal{A}_{\mathbf{x}}(S)) - \nabla\Phi_S(\mathcal{A}_{\mathbf{x}}(S))\|, \tag{4}$$

where the first term on the right-hand-side corresponds to the optimization error of solving the empirical minimax problem (2) and the second term corresponds to the generalization error of the gradients of primal function. The above inequality satisfies from the triangle inequality.

Let $\|\cdot\|$ be the Euclidean norm for simplicity and $B(\mathbf{x}_0, R) := \{\mathbf{x} \in \mathbb{R}^d : \|\mathbf{x} - \mathbf{x}_0\| \leq R\}$ denote a ball with center $\mathbf{x}_0 \in \mathbb{R}^d$ and radius $R$. For the closed convex set $\mathcal{X}$, we assume that there is a radius $R_1$ such that $\mathcal{X} \in B(\mathbf{x}^*, R_1)$. Let $\mathcal{A}(S) := (\mathcal{A}_\mathbf{x}(S), \mathcal{A}_\mathbf{y}(S))$ denote the output of an algorithm $\mathcal{A}$ for solving the empirical minimax problem (2) with dataset $S$ and $\nabla f = (\nabla_\mathbf{x} f, \nabla_\mathbf{y} f)$ denote the gradient of a function $f$.

**Definition 2** (Strongly convex function). *Let $\mu_\mathbf{y} > 0$. A differentiable function $g : \mathcal{W} \to \mathbb{R}$ is called $\mu$-strongly-convex in $\mathbf{w}$ if the following inequality holds for every $\mathbf{w}_1$, $\mathbf{w}_2$:*

$$g(\mathbf{w}_1) - g(\mathbf{w}_2) \geq \langle \nabla g(\mathbf{w}_2), \mathbf{w}_1 - \mathbf{w}_2 \rangle + \frac{\mu}{2} \|\mathbf{w}_1 - \mathbf{w}_2\|^2,$$

*we say $g$ is $\mu$-strongly-concave if $-g$ is $\mu$-strongly-convex.*

**Definition 3** (Smooth function). *Let $\beta > 0$. A function $f : \mathcal{X} \times \mathcal{Y} \times \mathcal{Z} \to \mathbb{R}$ is $\beta$-smooth in $(\mathbf{x}, \mathbf{y})$ if the function is continuous differentiable and for any $\mathbf{x}_1, \mathbf{x}_2 \in \mathcal{X}$, $\mathbf{y}_1, \mathbf{y}_2 \in \mathcal{Y}$ and $\mathbf{z} \in \mathcal{Z}$, $f(\mathbf{x}, \mathbf{y}; \mathbf{z})$ satisfies*

$$\left\| \begin{pmatrix} \nabla_\mathbf{x} f(\mathbf{x}_1, \mathbf{y}_1; \mathbf{z}) - \nabla_\mathbf{x} f(\mathbf{x}_2, \mathbf{y}_2; \mathbf{z}) \\ \nabla_\mathbf{y} f(\mathbf{x}_1, \mathbf{y}_1; \mathbf{z}) - \nabla_\mathbf{y} f(\mathbf{x}_2, \mathbf{y}_2; \mathbf{z}) \end{pmatrix} \right\| \leq \beta \left\| \begin{pmatrix} \mathbf{x}_1 - \mathbf{x}_2 \\ \mathbf{y}_1 - \mathbf{y}_2 \end{pmatrix} \right\|.$$

**Assumption 1** (Nonconvex-strongly-concave minimax problem). *In order to obtain meaningful conclusions, we make the following assumptions:*

- *Let $\mu_\mathbf{y} > 0$. The function $f(\mathbf{x}, \mathbf{y}; \mathbf{z})$ is $\mu_\mathbf{y}$-strongly concave in $\mathbf{y} \in \mathcal{Y}$ for any $\mathbf{x} \in \mathcal{X}$ and $\mathbf{z} \in \mathcal{Z}$.*

- *The function $f(\mathbf{x}, \mathbf{y}; \mathbf{z})$ is $\beta$-smooth in $(\mathbf{x}, \mathbf{y}) \in \mathcal{X} \times \mathcal{Y}$ for any $\mathbf{z}$.*

- *$\mathcal{X}$ and $\mathcal{Y}$ are compact convex sets, which means that there exist constants $D_\mathcal{X}, D_\mathcal{Y} > 0$ such that for any $\mathbf{x} \in \mathcal{X}$, $\|\mathbf{x}\|^2 \leq D_\mathcal{X}$ and for any $\mathbf{y} \in \mathcal{Y}$, $\|\mathbf{y}\|^2 \leq D_\mathcal{Y}$.*

The first two assumptions in Assumption 1 are standard in NC-SC minimax problems (Zhang et al., 2021; Farnia & Ozdaglar, 2021; Lei et al., 2021; Li & Liu, 2021a) and the last one in Assumption 1 is widely used in uniform convergence analysis (Kleywegt et al., 2002; Davis & Drusvyatskiy, 2022; Zhang et al., 2022).

**Assumption 2** (Lipschitz continuity). *Let $L > 0$, assume that for any $\mathbf{x} \in \mathcal{X}$ and any $\mathbf{y} \in \mathcal{Y}$ respectively for any $\mathbf{z}$, the function $f(\mathbf{x}, \mathbf{y}; \mathbf{z})$ satisfies*

$$\|\nabla_\mathbf{x} f(\mathbf{x}, \mathbf{y}; \mathbf{z})\| \leq L \quad and \quad \|\nabla_\mathbf{y} f(\mathbf{x}, \mathbf{y}; \mathbf{z})\| \leq L.$$

Lipschitz assmuption is also the standard assumption and widely used in literature such as Zhang et al. (2021); Farnia & Ozdaglar (2021); Lei et al. (2021); Li & Liu (2021a). But we need to emphasize that our main Theorem 1 and Theorem 3 do not require the Lipschitz assumption. Instead, we introduce a weaker assumption called Bernstein condition in minimax problems.

**Definition 4** (Bernstein condition). *Given a random variable $X$ with mean $\mu = \mathbb{E}[X]$ and variance $\sigma^2 = \mathbb{E}[X^2] - \mu^2$, we say that Bernstein's condition holds if there exists $B > 0$ such that for all $2 \leq k \leq n$,*

$$|\mathbb{E}[(X - \mu)^k]| \leq \frac{1}{2} k! \sigma^2 B^{k-2}. \tag{5}$$

**Remark 1.** *Bernstein condition has been widely used to obtain tail bounds that may be tighter than the Hoeffding bounds. It is easy to verify that any bounded variable satisfies the Bernstein condition. Moreover, the Bernstein condition is milder than the bounded assumption of random variables and is also satisfied by various unbounded variables. For example, a random variable is sub-exponetial if it satisfies the Bernstein condition (Wainwright, 2019). Please refer to Wainwright (2019) for more discussions. Next, we introduce a straightforward generalization of the Bernstein condition to minimax problems. We formally state these extension in the following assumptions.*

**Assumption 3.** *In minimax problems, the function $f(\mathbf{x}, \mathbf{y}; \mathbf{z})$ satisfies Bernstein condition in $\mathbf{x}^*$ for $\mathbf{y}^*$: there exists $B_{\mathbf{x}^*} > 0$ such that for all $2 \leq k \leq n$,*

$$\mathbb{E}[\|\nabla_{\mathbf{x}} f(\mathbf{x}^*, \mathbf{y}^*; \mathbf{z})\|^k] \leq \frac{1}{2} k! \mathbb{E}[\|\nabla_{\mathbf{x}} f(\mathbf{x}^*, \mathbf{y}^*; \mathbf{z})\|^2] B_{\mathbf{x}^*}^{k-2}. \tag{6}$$

*And the function $f(\mathbf{x}, \mathbf{y}; \mathbf{z})$ satisfies Bernstein condition in $\mathbf{y}^*(\mathbf{x})$ for any fixed $\mathbf{x}$: there exists $B_{\mathbf{y}^*} > 0$ such that for all $2 \leq k \leq n$,*

$$\mathbb{E}[\|\nabla_{\mathbf{y}} f(\mathbf{x}, \mathbf{y}^*(\mathbf{x}); \mathbf{z})\|^k] \leq \frac{1}{2} k! \mathbb{E}[\|\nabla_{\mathbf{y}} f(\mathbf{x}, \mathbf{y}^*(\mathbf{x}); \mathbf{z})\|^2] B_{\mathbf{y}^*}^{k-2}, \tag{7}$$

*where $\mathbf{y}^*(\mathbf{x}) := \arg\max_{\mathbf{y} \in \mathcal{Y}} F(\mathbf{x}, \mathbf{y})$.*

**Remark 2.** *We can easily obtain that Assumption 2 can derive Assumption 3. For example, if function $f$ is L-Lipschitz continuous, then $\|\nabla_{\mathbf{x}} f(\mathbf{x}, \mathbf{y}; \mathbf{z})\| \leq L$. Thus for any $\mathbf{x} \in \mathcal{X}, \mathbf{y} \in \mathcal{Y}$ and for all $2 \leq k \leq n$, we have $\mathbb{E}[\|\nabla_{\mathbf{y}} f(\mathbf{x}, \mathbf{y}; \mathbf{z})\|^k] \leq \frac{1}{2} k! \mathbb{E}[\|\nabla_{\mathbf{y}} f(\mathbf{x}, \mathbf{y}; \mathbf{z})\|^2] L^{k-2}$, which means that the function $f$ satisfies Bernstein condition for any $\mathbf{x}, \mathbf{y}$. Similarly, $\mathbb{E}[\|\nabla_{\mathbf{x}} f(\mathbf{x}, \mathbf{y}; \mathbf{z})\|^k] \leq \frac{1}{2} k! \mathbb{E}[\|\nabla_{\mathbf{x}} f(\mathbf{x}, \mathbf{y}; \mathbf{z})\|^2] L^{k-2}$ can be easily derived. Furthermore, Bernstein condition assumption is pretty mild since $B_{\mathbf{y}^*}$ only depends on gradients at $(\mathbf{x}, \mathbf{y}^*(\mathbf{x}))$ for any $\mathbf{x} \in \mathcal{X}$ and $B_{\mathbf{x}^*}$ only depends on gradients at $(\mathbf{x}^*, \mathbf{y}^*)$.*

# 4 UNIFORM LOCALIZED CONVERGENCE AND GENERALIZATION BOUNDS

Uniform convergence of the gradients for the primal functions measures the deviation between the gradients of the primal population function $\nabla\Phi(\mathbf{x})$ and the gradients of the primal empirical function $\nabla\Phi_S(\mathbf{x})$. In this section, we provide the sharper uniform convergence of the gradients for the primal functions comparing with Zhang et al. (2022).

**Theorem 1.** *Under Assumption 1 and 3, we have the following inequality that for any $\delta \in (0, 1)$ and for all $\mathbf{x} \in \mathcal{X}$, with probability at least $1 - \delta$,*

$$\|\nabla\Phi(\mathbf{x}) - \nabla\Phi_S(\mathbf{x})\| \leq \frac{\beta}{\mu_{\mathbf{y}}} \left( \sqrt{\frac{2\mathbb{E}\|\nabla_{\mathbf{y}} f(\mathbf{x}, \mathbf{y}^*(\mathbf{x}); \mathbf{z})\|^2 \log \frac{4}{\delta}}{n}} + \frac{B_{\mathbf{y}^*} \log \frac{4}{\delta}}{n} \right)$$

$$+ \sqrt{\frac{2\mathbb{E}\|\nabla_{\mathbf{x}} f(\mathbf{x}^*, \mathbf{y}^*; \mathbf{z})\|^2 \log \frac{8}{\delta}}{n}} + \frac{B_{\mathbf{x}^*} \log \frac{8}{\delta}}{n} + \frac{C\beta(\mu_{\mathbf{y}} + \beta)}{\mu_{\mathbf{y}}} \max\left\{ \|\mathbf{x} - \mathbf{x}^*\|, \frac{1}{n} \right\}$$

$$\times \left( \sqrt{\frac{d + \log \frac{16 \log_2(\sqrt{2}R_1 n + 1)}{\delta}}{n}} + \frac{d + \log \frac{16 \log_2(\sqrt{2}R_1 n + 1)}{\delta}}{n} \right),$$

*where $C$ is a absolute constant.*

There is only one uniform convergence of gradients for primal functions in minimax problems given in Zhang et al. (2022). Here is their main theorem in NC-SC settings.

**Theorem 2** (Theorem in (Zhang et al., 2022)). *Under Assumption 1 and 2, we have*

$$\mathbb{E}\left[ \max_{\mathbf{x} \in \mathcal{X}} \|\nabla\Phi(\mathbf{x}) - \nabla\Phi_S(\mathbf{x})\| \right] = \tilde{O}\left( \frac{L(\mu_{\mathbf{y}} + \beta)}{\mu_{\mathbf{y}}} \sqrt{\frac{d}{n}} \right),$$

*where $\tilde{O}(\cdot)$ hides logarithmic factors.*

**Remark 3.** *We now compare our uniform convergence of gradient for primal functions with Zhang et al. (2022). Firstly, our result is the only one with high-probability format. Besides, we successfully relax the assumptions. Theorem 2 requires the Lipschitz continuity assumption, while our result only needs Bernstein condition assumption. Please refer to Remark 1 Remark 2 for the detailed comparison between these assumptions. Then, the factor in Theorem 2 is $\frac{L(\mu_{\mathbf{y}} + \beta)}{\mu_{\mathbf{y}}}$, while our result in Theorem 1 is $\frac{C\beta(\mu_{\mathbf{y}} + \beta)}{\mu_{\mathbf{y}}} \max\left\{ \|\mathbf{x} - \mathbf{x}^*\|, \frac{1}{n} \right\}$, not involving the term $L$, which may be very large and even infinite without Lipschitz continuity assumption. Finally, while Zhang et al. (2022) studied*

*the worst-case upper bounds on the parameters, results based on generic chaining yield upper bound related to the parameters. As shown Theorem 1, we have the term $\max\{\|\mathbf{x}-\mathbf{x}^*\|, \frac{1}{n}\}$ before the term $O(\sqrt{d/n})$, indicating that our results improve as the calculated parameters of algorithms approach the optimal solution. In the optimal scenario, when $\|\mathbf{x}-\mathbf{x}^*\| \leq \frac{1}{n}$, we can attain the best (sharper) results.*

Next, we privide a dimension-free uniform convergence of gradients for the primal functions when the PL condition is satisfied. Firstly, we introduce the extension of the PL condition to the minimax problem used in Guo et al. (2020); Yang et al. (2020).

**Assumption 4** (x-side $\mu_{\mathbf{x}}$-Polyak-Lojasiewicz condition). *For any $\mathbf{y} \in \mathcal{Y}$, the function $F(\mathbf{x}, \mathbf{y})$ satisfies the x-side $\mu_{\mathbf{x}}$-Polyak-Lojasiewicz (PL) condition with parameter $\mu_{\mathbf{x}} > 0$ on all $\mathbf{x} \in \mathcal{X}$ if*

$$F(\mathbf{x}, \mathbf{y}) - \inf_{\mathbf{x}'} F(\mathbf{x}', \mathbf{y}) \leq \frac{1}{2\mu_{\mathbf{x}}} \|\nabla_{\mathbf{x}} F(\mathbf{x}, \mathbf{y})\|^2.$$

**Remark 4.** *Numerous studies have been conducted on deep learning to provide evidence for the validity of the PL condition in risk minimization problems. This condition has been demonstrated to hold either globally or locally in certain networks with specific structural, activation, or loss function characteristics (Hardt & Ma, 2016; Li & Yuan, 2017; Zhou & Liang, 2017; Li & Liang, 2018; Arora et al., 2018; Charles & Papailiopoulos, 2018; Du et al., 2018; Allen-Zhu et al., 2019). For instance, Du et al. (2018) has exhibited that if a two-layer neural network possesses a sufficiently wide width, the PL condition is upheld within a ball centered at the initial solution, and the global optimum is situated within this same ball. Additionally, Allen-Zhu et al. (2019) have further demonstrated that in overparameterized deep neural networks utilizing ReLU activation, the PL condition is applicable to a global optimum located in the vicinity of a random initial solution.*

**Theorem 3.** *Under Assumption 1 and 3, assume that the population risk $F(\mathbf{x}, \mathbf{y})$ satisfies Assumption 4 with parameter $\mu_{\mathbf{x}}$ and let $c = \max\{16C^2, 1\}$. We have that for all $\mathbf{x} \in \mathcal{X}$, when $n \geq \frac{c\beta^2(\mu_{\mathbf{y}}+\beta)^2(d+\log\frac{8\log_2\sqrt{2}R_1 n+1}{\delta})}{\mu_{\mathbf{y}}^2\mu_{\mathbf{x}}^2}$ with probability at least $1-\delta$*

$$\|\nabla\Phi(\mathbf{x}) - \nabla\Phi_S(\mathbf{x})\| \leq \|\nabla\Phi_S(\mathbf{x})\| + 2\sqrt{\frac{2\mathbb{E}\|\nabla_{\mathbf{x}} f(\mathbf{x}^*, \mathbf{y}^*; \mathbf{z})\|^2 \log\frac{8}{\delta}}{n}}$$

$$+ \frac{2B_{\mathbf{x}^*}\log\frac{8}{\delta}}{n} + \frac{\mu_{\mathbf{x}}}{n} + \frac{2\beta}{\mu_{\mathbf{y}}}\left(\sqrt{\frac{2\mathbb{E}\|\nabla_{\mathbf{y}} f(\mathbf{x}, \mathbf{y}^*(\mathbf{x}); \mathbf{z})\|^2 \log\frac{4}{\delta}}{n}} + \frac{B_{\mathbf{y}^*}\log\frac{4}{\delta}}{n}\right).$$

**Remark 5.** *The following inequality can be easily derived using the norm triangle inequality and Cauchy–Bunyakovsky–Schwarz inequality.*

$$\Phi(\mathbf{x}) - \Phi(\mathbf{x}^*) \leq \frac{8\|\nabla\Phi_S(\mathbf{x})\|^2}{\mu_{\mathbf{x}}} + \frac{16\beta^2\mathbb{E}\|\nabla_{\mathbf{y}} f(\mathbf{x}, \mathbf{y}^*(\mathbf{x}); \mathbf{z})\|^2 \log\frac{4}{\delta}}{\mu_{\mathbf{x}}\mu_{\mathbf{y}}^2 n}$$

$$+ \frac{16\mathbb{E}\|\nabla_{\mathbf{x}} f(\mathbf{x}^*, \mathbf{y}^*; \mathbf{z})\|^2 \log\frac{8}{\delta}}{\mu_{\mathbf{x}} n} + \frac{2\left(\frac{2\beta B_{\mathbf{y}^*}}{\mu_{\mathbf{y}}}\log\frac{4}{\delta} + 2B_{\mathbf{x}^*}\log\frac{8}{\delta} + \mu_{\mathbf{x}}\right)^2}{\mu_{\mathbf{x}} n^2}. \tag{8}$$

*We can easily derive (8) from Theorem 3 to gain the excess primal risk bound, where $\|\nabla\Phi_S(\mathbf{x})\|$ is the empirical optimization error of the primal function. In Theorem 3 and (8), $\|\nabla\Phi_S(\mathbf{x})\|$ can be very tiny since most famous optimization algorithms such as GDA and SGDA, can optimize it small enough. The term $\mathbb{E}\|\nabla_{\mathbf{x}} f(\mathbf{x}^*, \mathbf{y}^*; \mathbf{z})\|^2$ and $\mathbb{E}\|\nabla_{\mathbf{y}} f(\mathbf{x}, \mathbf{y}^*(\mathbf{x}); \mathbf{z})\|^2$ can be also tiny since they only depend on the the gradient of the optima $\mathbf{x}^*$ w.r.t $\mathbf{x}$ and the gradient of the optima $\mathbf{y}^*(\mathbf{x})$ w.r.t. $\mathbf{y}$. Thus, comparing with Theorem 2 in Zhang et al. (2022), this uniform localized convergence bound is clearly tighter when relaxing Lipschitz continuity (Assumption 2) and considering PL condition (Assumption 4). We further analyze these two terms $\mathbb{E}\|\nabla_{\mathbf{x}} f(\mathbf{x}^*, \mathbf{y}^*; \mathbf{z})\|^2$ and $\mathbb{E}\|\nabla_{\mathbf{y}} f(\mathbf{x}, \mathbf{y}^*(\mathbf{x}); \mathbf{z})\|^2$ using "Self-bounding" property for smooth function (Srebro et al., 2010) and considering specific algorithms in Section 5, which can derive to almost $O(1/n^2)$ bounds. Additionally, uniform convergence often implies results with a square-root dependence on the dimension $d$ such as Theorem 1 and Zhang et al. (2022). Another distinctive improvement of Theorem 3 is that we remove the dimension $d$ when the population risk $F(\mathbf{x}, \mathbf{y})$ satisfies the x-side PL condition and the sample size $n$ is large enough.*

**Remark 6.** *There are two mainly challenges in our work for minimax problems. On one hand, comparing with uniform convergence in Zhang et al. (2022), we use a novel uniform localized convergence techniques (Xu & Zeevi, 2020) to construct a functional w.r.t. loss functions on minimax problems. This two layer structure involves difficulties. On the other hand, it is noteworthy that the optimal point $\mathbf{y}^*(\mathbf{x}) := \arg\max_{\mathbf{y} \in \mathcal{Y}} F(\mathbf{x}, \mathbf{y})$ for a given $\mathbf{x}$ differs from $\mathbf{y}_S^*(\mathbf{x}) := \arg\max_{\mathbf{y} \in \mathcal{Y}} F_S(\mathbf{x}, \mathbf{y})$, thus introducing an additional error term $\|\mathbf{y}^*(\mathbf{x}) - \mathbf{y}_S^*(\mathbf{x})\|$. Compared to Zhang et al. (2022), they only need to bound this term with $O(\frac{1}{\sqrt{n}})$. But we need to reach the upper bound of order $O(\frac{1}{n})$ under certain assumptions.*

## 5 APPLICATION

### 5.1 EMPIRICAL SADDLE POINT

Empirical saddle point (ESP) problem, which is also known as sample average approximation (SAA) (Zhang et al., 2021) refers to (2). We denote $(\hat{\mathbf{x}}^*, \hat{\mathbf{y}}^*)$ as one of the ESP solution to (2). Then we can provide some important theorems in this subsection.

**Theorem 4.** *Suppose the empirical saddle point $(\hat{\mathbf{x}}^*, \hat{\mathbf{y}}^*)$ exists and Assumption 1 and 3 hold, for any $\delta \in (0, 1)$, with probability at least $1 - \delta$, we have*

$$\|\nabla \Phi(\hat{\mathbf{x}}^*)\| = O\left(\sqrt{\frac{d + \log \frac{\log n}{\delta}}{n}}\right)$$

**Remark 7.** *When Assumption 1 and 3 hold, Theorem 4 gives that the population optimization error $\|\nabla \Phi(\hat{\mathbf{x}}^*)\|$ is of order $O\left(\sqrt{\frac{d + \log \frac{1}{\delta}}{n}}\right)$ ($\log n$ is small and can be ignored typically). Note that this result doesn't require the Lipschitz continuity assumption (Assumption 2). Although it may be hard to find $(\hat{\mathbf{x}}^*, \hat{\mathbf{y}}^*)$ in NC-SC minimax problems, it is still meaningful when assuming the ESP $(\hat{\mathbf{x}}^*, \hat{\mathbf{y}}^*)$ has been found.*

**Theorem 5.** *Suppose Assumption 1 and 3 hold. Assume that the population risk $F(\mathbf{x}, \mathbf{y})$ satisfies Assumption 4 with parameter $\mu_{\mathbf{x}}$. For any $\delta \in (0, 1)$, with probability at least $1 - \delta$, when $n \geq \frac{c\beta^2(\mu_{\mathbf{y}} + \beta)^2(d + \log \frac{8 \log_2 \sqrt{2} R_1 n + 1}{\delta})}{\mu_{\mathbf{y}}^2 \mu_{\mathbf{x}}^2}$, where $c$ is an absolute constant, we have*

$$\Phi(\hat{\mathbf{x}}^*) - \Phi(\mathbf{x}^*) \leq \frac{12\beta^2 \mathbb{E}\|\nabla_{\mathbf{y}} f(\hat{\mathbf{x}}^*, \mathbf{y}^*(\hat{\mathbf{x}}^*); \mathbf{z})\|^2 \log \frac{4}{\delta}}{\mu_{\mathbf{x}} \mu_{\mathbf{y}}^2 n} + \frac{12 \mathbb{E}\|\nabla_{\mathbf{x}} f(\mathbf{x}^*, \mathbf{y}^*; \mathbf{z})\|^2 \log \frac{8}{\delta}}{\mu_{\mathbf{x}} n}$$

$$+ \frac{3 \left(\frac{2\beta B_{\mathbf{y}^*}}{\mu_{\mathbf{y}}} \log \frac{4}{\delta} + 2B_{\mathbf{x}^*} \log \frac{8}{\delta} + \mu_{\mathbf{x}}\right)^2}{2\mu_{\mathbf{x}} n^2}.$$

*Furthermore, if we let $n \geq \max\left\{\frac{c\beta^2(\mu_{\mathbf{y}} + \beta)^2(d + \log \frac{8 \log_2 \sqrt{2} R_1 n + 1}{\delta})}{\mu_{\mathbf{y}}^2 \mu_{\mathbf{x}}^2}, \frac{48\beta^3 \log \frac{4}{\delta}}{\mu_{\mathbf{x}} \mu_{\mathbf{y}}^2}\right\}$ and assume the function $f(\mathbf{x}, \mathbf{y}; \mathbf{z})$ is non-negative and $\Phi(\mathbf{x}^*) = O\left(\frac{1}{n}\right)$, we have*

$$\Phi(\hat{\mathbf{x}}^*) - \Phi(\mathbf{x}^*) = O\left(\frac{\log^2 \frac{1}{\delta}}{n \left(n - \frac{48\beta^3 \log \frac{4}{\delta}}{\mu_{\mathbf{x}} \mu_{\mathbf{y}}^2}\right)}\right).$$

**Remark 8.** *Theorem 5 shows that when the population minimax risk $F(\mathbf{x}, \mathbf{y})$ satisfies $\mathbf{x}$-side PL condition, we can provide a sharper excess risk bound for primal function, which can be almost $O(1/n^2)$. Note that the optimal population primal function $\Phi(\mathbf{x}^*) = O(1/n)$ is a very common assumption in many researches such as Srebro et al. (2010); Zhang et al. (2017); Liu et al. (2018); Zhang & Zhou (2019); Lei & Ying (2020), which is natural because $F(\mathbf{x}^*, \mathbf{y}^*)$ is the minimal population risk. Now we compare our results with recent related work (Li & Liu, 2021b), which studied the general machine learning settings for $f(\mathbf{w})$ under PL condition. Their empirical risk minimizer (ERM) excess risk bounds provided $O(1/n^2)$ order rates. We analyze the excess risk with primal functions which involve an additional error term. In consequence, our result for ESP just approximate $O(1/n^2)$ order rate.*

**Algorithm 1** Two-timescale GDA for minimax problem

1: **Input:** $(\mathbf{x}_1, \mathbf{y}_1)$, step sizes $\eta_{\mathbf{x}} > 0, \eta_{\mathbf{y}} > 0$ and dataset $S = \{\mathbf{z}_1, \dots, \mathbf{z}_n\}$
2: **for** $t = 1, \dots, T$ **do**
3: $\quad$ update $\mathbf{x}_{t+1} = \mathbf{x}_t - \eta_{\mathbf{x}} \nabla_{\mathbf{x}} F_S(\mathbf{x}_t, \mathbf{y}_t)$
4: $\quad$ update $\mathbf{y}_{t+1} = \mathbf{y}_t + \eta_{\mathbf{y}} \nabla_{\mathbf{y}} F_S(\mathbf{x}_t, \mathbf{y}_t)$

**Algorithm 2** Two-timescale SGDA for minimax problem

1: **Input:** $(\mathbf{x}_1, \mathbf{y}_1) = (0, 0)$, step sizes $\{\eta_{\mathbf{x}_t}\}_t > 0, \{\eta_{\mathbf{y}_t}\}_t > 0$ and dataset $S = \{\mathbf{z}_1, \dots, \mathbf{z}_n\}$
2: **for** $t = 1, \dots, T$ **do**
3: $\quad$ update $\mathbf{x}_{t+1} = \mathbf{x}_t - \eta_{\mathbf{x}_t} \nabla_{\mathbf{x}} f(\mathbf{x}_t, \mathbf{y}_t; \mathbf{z}_{i_t})$
4: $\quad$ update $\mathbf{y}_{t+1} = \mathbf{y}_t + \eta_{\mathbf{y}_t} \nabla_{\mathbf{y}} f(\mathbf{x}_t, \mathbf{y}_t; \mathbf{z}_{i_t})$

## 5.2 GRADIENT DESCENT ASCENT

Gradient descent ascent (GDA) presented in Algorithm 1 is one of the most popular algorithms and has been widely used in minimax problems. In this subsection, we provide the generalization and excess risk bounds of primal functions with the two-timescale GDA Algorithm which is harder to analyze compared to GDMax and multistep GDA (Lin et al., 2020).

**Theorem 6.** *Suppose Assumption 1 and 2 hold. Let $\{\mathbf{x}_t\}_t$ be the sequence produced by Algorithm 1 with the step sizes chosen as $\eta_{\mathbf{x}} = \frac{1}{16(\frac{\beta}{\mu}+1)^2\beta}$ and $\eta_{\mathbf{y}} = \frac{1}{\beta}$, for any $\delta \in (0, 1)$, with probability at least $1 - \delta$, we have*

$$\frac{1}{T}\sum_{t=1}^{T} \|\nabla\Phi(\mathbf{x}_t)\|^2 \leq O\left(\frac{1}{T}\right) + O\left(\frac{d + \log\frac{16\log_2(\sqrt{2}R_1 n + 1)}{\delta}}{n}T\right).$$

*Furthermore, when $T \asymp O\left(\sqrt{\frac{n}{d}}\right)$, we have*

$$\frac{1}{T}\sum_{t=1}^{T} \|\nabla\Phi(\mathbf{x}_t)\|^2 \leq O\left(\frac{d + \log\frac{\log n}{\delta}}{\sqrt{nd}}\right).$$

**Remark 9.** *Theorem 6 reveals that we need to balance the optimization error and the generalization error for GDA. According to the results, the iterative complexity of Algorithm 1 should be chosen as $T \asymp O\left(\sqrt{\frac{n}{d}}\right)$, which achieves the optimal population optimization error of primal function.*

*In comparison to Theorem 4, Theorem 6 derives into population optimization error w.r.t SGD, which is much more difficult. To establish population optimization error, we need to bound the empirical optimization error, an area where no research has been conducted in NC-SC settings with high probability. One possible approach is to construct the martingale difference sequence of step $T$ for primal functions, yet this constitutes a separate topic warranting further exploration. Theorem 6 aims to directly apply Theorem 1 to SGD. Comparing with Theorem 3, Theorem 6 only necessitates smooth and Lipschitz conditions (Assumption 1 and 2) and doesn't require PL conditions. In fact, Theorem 7 constitutes a further extension of Theorem 3.*

Next, we provide the excess risk of primal functions $\Phi(\bar{\mathbf{x}}_T) - \Phi(\mathbf{x}^*)$ for Algorithm 1, where $\bar{\mathbf{x}}_T = \frac{1}{T}\sum_{t=1}^{T} \mathbf{x}_t$. We need to know the empirical optimization error $\|\nabla\Phi_S(\bar{\mathbf{x}}_T)\|$. Unfortunately, although the generalization bounds we proved are in NC-SC settings, we require the SC-SC assumptions to derive the empirical optimization error bound of primal functions, to gain the high probability bound. We relax this SC-SC assumption in Appendix E using existing optimization error bound with expectation format.

**Definition 5.** *A function $g : \mathcal{X} \times \mathcal{Y} \to \mathbb{R}$ is $\mu_{\mathbf{x}}$-strongly-convex-$\mu_{\mathbf{y}}$-strongly-concave if $g(\cdot, \mathbf{y})$ is $\mu_{\mathbf{x}}$-strongly-convex for any $\mathbf{y} \in \mathcal{Y}$ and $g(\mathbf{x}, \cdot)$ is $\mu_{\mathbf{y}}$-strongly-concave for any $\mathbf{x} \in \mathcal{X}$*

**Assumption 5** (Strongly-convex-strongly-concave minimax problem). *Assume Assumption 1 holds and let $\mu_{\mathbf{x}} > 0, \mu_{\mathbf{y}} > 0$. The function $f(\mathbf{x}, \mathbf{y}; \mathbf{z})$ is $\mu_{\mathbf{x}}$-strongly-convex-$\mu_{\mathbf{y}}$-strongly-concave in $\mathbf{y} \in \mathcal{Y}$ for any $\mathbf{x} \in \mathcal{X}$ and $\mathbf{z} \in \mathcal{Z}$.*

**Remark 10.** *Assumption 5 is commonly used in SC-SC problems (Zhang et al., 2021; Li & Liu, 2021a). We require this assumption to derive the empirical **optimization error** bound of primal functions. The detailed proofs of the optimization error bound $\|\nabla\Phi_S(\bar{\mathbf{x}}_T)\|$ are given in Section D.2 for GDA and in Section D.3 for SGDA.*

**Theorem 7.** *Suppose Assumption 3 and 5 hold. Let $\{\mathbf{x}_t\}_t$ be the sequence produced by Algorithm 1 and $\bar{\mathbf{x}}_T = \frac{1}{T}\sum_{t=1}^T \mathbf{x}_t$ with the step sizes chosen as $\eta_{\mathbf{x}} = \frac{1}{16(\frac{\beta}{\mu}+1)^2\beta}$ and $\eta_{\mathbf{y}} = \frac{1}{\beta}$, for any $\delta \in (0,1)$, with probability at least $1-\delta$, when $T \asymp n$ and $n \geq \frac{c\beta^2(\mu_{\mathbf{y}}+\beta)^2(d+\log\frac{8\log_2\sqrt{2}R_1 n+1}{\delta})}{\mu_{\mathbf{y}}^2\mu_{\mathbf{x}}^2}$, where $c$ is an absolute constant, we have*

$$\Phi(\bar{\mathbf{x}}_T) - \Phi(\mathbf{x}^*) = O\left(\frac{\mathbb{E}\|\nabla_{\mathbf{x}}f(\mathbf{x}^*,\mathbf{y}^*;\mathbf{z})\|^2\log\frac{1}{\delta}}{n} + \frac{\mathbb{E}\|\nabla_{\mathbf{y}}f(\bar{\mathbf{x}}_T,\mathbf{y}^*(\bar{\mathbf{x}}_T);\mathbf{z})\|^2\log\frac{1}{\delta}}{n} + \frac{\log^2\frac{1}{\delta}}{n^2}\right).$$

*Furthermore, Let $T \asymp n^2$ and $n \geq \max\left\{\frac{c\beta^2(\mu_{\mathbf{y}}+\beta)^2(d+\log\frac{8\log_2\sqrt{2}R_1 n+1}{\delta})}{\mu_{\mathbf{y}}^2\mu_{\mathbf{x}}^2}, \frac{128\beta^3\log\frac{4}{\delta}}{\mu_{\mathbf{x}}\mu_{\mathbf{y}}^2}\right\}$. Assume the function $f(\mathbf{x},\mathbf{y};\mathbf{z})$ is non-negative and $\Phi(\mathbf{x}^*) = O\left(\frac{1}{n}\right)$, we have*

$$\Phi(\bar{\mathbf{x}}_T) - \Phi(\mathbf{x}^*) = O\left(\frac{\log^2\frac{1}{\delta}}{n\left(n - \frac{128\beta^3\log\frac{4}{\delta}}{\mu_{\mathbf{x}}\mu_{\mathbf{y}}^2}\right)}\right).$$

**Remark 11.** *Theorem 7 shows that the excess risk for primal functions can be bound almost to $O\left(1/n^2\right)$ comparing with the optimal result $O(1/n)$ given in Li & Liu (2021a) when $n$ is large enough. Note that we require the SC-SC assumption to derive the empirical optimization error. If we give this bound in expectation, we can relax the SC-SC assumption with $\mathbf{x}$-side PL-strongly-concave assumption instead.*

## 5.3 STOCHASTIC GRADIENT DESCENT ASCENT

We now analyze the excess risk bound of primal functions for stochastic gradient descent ascent (SGDA). The algorithmic scheme that we study is two-timescale SGDA ($\eta_{\mathbf{x}} \neq \eta_{\mathbf{y}}$) with variable stepsizes, presented in Algorithm 2 which is more nature in the real problems.

**Theorem 8.** *Suppose Assumption 2 and 5 hold, let $\{\mathbf{x}_t\}_t$ be the sequence produced by Algorithm 2 and $\bar{\mathbf{x}}_T = \frac{1}{T}\sum_{t=1}^T \mathbf{x}_t$ with the step sizes chosen as $\eta_{\mathbf{x}_t} = \frac{1}{\mu_{\mathbf{x}}(t+t_0)}$ and $\eta_{\mathbf{y}_t} = \frac{1}{\mu_{\mathbf{y}}(t+t_0)}$, for any $\delta \in (0,1)$, with probability at least $1-\delta$, when $T \asymp n^2$ and $n \geq \frac{c\beta^2(\mu_{\mathbf{y}}+\beta)^2(d+\log\frac{8\log_2\sqrt{2}R_1 n+1}{\delta})}{\mu_{\mathbf{y}}^2\mu_{\mathbf{x}}^2}$, where $c$ is an absolute constant, we have*

$$\Phi(\bar{\mathbf{x}}_T) - \Phi(\mathbf{x}^*) = O\left(\frac{\mathbb{E}\|\nabla_{\mathbf{x}}f(\mathbf{x}^*,\mathbf{y}^*;\mathbf{z})\|^2\log\frac{1}{\delta}}{n} + \frac{\mathbb{E}\|\nabla_{\mathbf{y}}f(\bar{\mathbf{x}}_T,\mathbf{y}^*(\bar{\mathbf{x}}_T);\mathbf{z})\|^2\log\frac{1}{\delta}}{n} + \frac{\log^2\frac{1}{\delta}}{n^2}\right).$$

*Furthermore, let $T \asymp n^4$ and $n \geq \max\left\{\frac{c\beta^2(\mu_{\mathbf{y}}+\beta)^2(d+\log\frac{8\log_2\sqrt{2}R_1 n+1}{\delta})}{\mu_{\mathbf{y}}^2\mu_{\mathbf{x}}^2}, \frac{128\beta^3\log\frac{8}{\delta}}{\mu_{\mathbf{x}}\mu_{\mathbf{y}}^2}\right\}$. Assume the function $f(\mathbf{x},\mathbf{y};\mathbf{z})$ is non-negative and $\Phi(\mathbf{x}^*) = O\left(\frac{1}{n}\right)$, we have*

$$\Phi(\bar{\mathbf{x}}_T) - \Phi(\mathbf{x}^*) = O\left(\frac{\log^2\frac{1}{\delta}}{n\left(n - \frac{128\beta^3\log\frac{8}{\delta}}{\mu_{\mathbf{x}}\mu_{\mathbf{y}}^2}\right)}\right).$$

**Remark 12.** *Theorem 8 reveals that under the SC-SC settings, the excess risk bound can be approximate $O(1/n^2)$ comparing with the optimal result $O(1/n)$ given in (Li & Liu, 2021a). Similarly, since the SC-SC assumption is required to derive the empirical optimization bound, we can relax the assumptions when we only need expectation bounds instead of high probability bounds.*

## 6 CONCLUSION

In this paper, we provide the improved generalization bounds for minimax problems with uniform localized convergence. We firstly provide a sharper bound measured by the gradients of primal functions with weaker assumptions. Then we provide dimension-independent results under PL condition. Finally we extend our main theorems into various algorithms to reach the optimal excess primal risk bounds. We notice that most optimization works focused on the gradient complexity with expectation results. It would be interesting to give the optimization error of $\bar{\mathbf{x}}_T$ or even $\mathbf{x}_T$ with high probability under weaker conditions. Combining with our generalization work, we can get a tighter excess primal risk bound with weaker conditions.

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
