## A ADDITIONAL DEFINITIONS AND LEMMATA

**Lemma 1** (Bernstein's inequality (Dirksen, 2015)). *Let $X_1, \ldots, X_n$ be real-valued, independent, mean-zero random variables and suppose that for some constants $\sigma, B > 0$,*

$$\frac{1}{n} \sum_{i=1}^{n} \mathbb{E}|X_i|^k \leq \frac{k!}{2} \sigma^2 B^{k-2}, \quad k = 2, 3, \ldots$$

*Then, $\forall \delta \in (0, 1)$, with probability at least $1 - \delta$*

$$\left| \frac{1}{n} \sum_{i=1}^{n} X_i \right| \leq \sqrt{\frac{2\sigma^2 \log \frac{2}{\delta}}{n}} + \frac{B \log \frac{2}{\delta}}{n}. \tag{9}$$

**Lemma 2** (A variant of the "uniform localized convergence" argument (Xu & Zeevi, 2020)). *Let $\mathbb{P}$ be a probability measure defined on a sample space $\mathcal{Z}$ and $\mathbb{P}_n$ be the corresponding empirical probability measure. For a function class $\mathcal{H} = \{h_f : f \in \mathcal{F}\}$ and functional $T : \mathcal{F} \to [0, R]$, assume there is a function $\psi(r; \delta)$ (possibly depending on the samples), which is non-decreasing with respect to $r$ and satisfies that $\forall \delta \in (0, 1)$, $\forall r \in [0, R]$, with probability at least $1 - \delta$,*

$$\sup_{f \in \mathcal{F}: T(f) \leq r} (\mathbb{P} - \mathbb{P}_n) h_f \leq \psi(r; \delta).$$

*Then, given any $\delta \in (0, 1)$ and $r_0 \in (0, R]$, with probability at least $1 - \delta$, for all $f \in \mathcal{F}$,*

$$(\mathbb{P} - \mathbb{P}_n) h_f \leq \psi \left( 2T(f) \vee r_0; \frac{\delta}{C_{r_0}} \right),$$

*where $C_{r_0} = 2 \log_2 \frac{2R}{r_0}$.*

**Definition 6** (Orlicz$_\alpha$ norm (Dirksen, 2015)). *For every $\alpha \in (0, +\infty)$ we define the Orlicz$_\alpha$ norm of a random $u$:*

$$\|u\|_{Orlicz_\alpha} = \inf\{K > 0 : \mathbb{E} \exp((|u|/K)^\alpha) \leq 2\}.$$

*A random variable (or vector) $X \in \mathbb{R}^d$ is $K$-sub-Gaussian if $\forall \boldsymbol{\lambda} \in \mathbb{R}^d$, we have*

$$\|\boldsymbol{\lambda}^{\mathrm{T}} X\|_{Orlicz_2} = K \|\boldsymbol{\lambda}\|_2.$$

*A random variable (or vector) $X \in \mathbb{R}^d$ is $K$-sub-exponential if $\forall \boldsymbol{\lambda} \in \mathbb{R}^d$, we have*

$$\|\boldsymbol{\lambda}^{\mathrm{T}} X\|_{Orlicz_1} = K \|\boldsymbol{\lambda}\|_2.$$

**Definition 7** (Orlicz$_\alpha$ processes (Dirksen, 2015)). *Let $\{X_f\}_{f \in \mathcal{F}}$ be a sequence of random variables. $\{X_f\}_{f \in \mathcal{F}}$ is called an Orlicz$_\alpha$ process for a metric $\mathsf{metr}(\cdot, \cdot)$ on $\mathcal{F}$ if*

$$\|X_{f_1} - X_{f_2}\|_{Orlicz_\alpha} \leq \mathsf{metr}(f_1, f_2), \quad \forall f_1, f_2 \in \mathcal{F}.$$

*Typically, the Orlicz$_2$ process is called "process with sub-Gaussian increments" and the Orlicz$_1$ process is called "process with sub-exponential increments".*

**Definition 8** (Mixed sub-Gaussian-sub-exponential increments (Dirksen, 2015)). *We say a process $(X_\theta)_{\theta \in \Theta}$ has mixed sub-Gaussian-sub-exponential increments with respect to the pair $(\mathsf{metr}_1, \mathsf{metr}_2)$ if for all $\theta_1, \theta_2 \in \Theta$,*

$$Prob(\|X_{\theta_1} - X_{\theta_2}\| \geq \sqrt{u} \cdot \mathsf{metr}_2(\theta_1, \theta_2) + u \cdot \mathsf{metr}_1(\theta_1, \theta_2)) \leq 2e^{-u}, \forall u \geq 0,$$

*where "$Prob$" means probability.*

**Definition 9** (Talagrand's $\gamma_\alpha$-functional (Dirksen, 2015)). *A sequence $F = (\mathcal{F}_n)_{n \geq 0}$ of subsets of $\mathcal{F}$ is called admissible if $|\mathcal{F}_0| = 1$ and $|\mathcal{F}_n| \leq 2^{2^n}$ for all $n \geq 1$. For any $0 < \alpha < \infty$, the $\gamma_\alpha$-functional of $(\mathcal{F}, \mathsf{metr})$ is defined by*

$$\gamma_\alpha(F, d) = \inf_F \sup_{f \in \mathcal{F}} \sum_{n=0}^{\infty} 2^{\frac{n}{\alpha}} \mathsf{metr}(f, \mathcal{F}_n),$$

*where the infimum is taken over all admissible sequences and we write $\mathsf{metr}(f, \mathcal{F}_n) = \inf_{s \in \mathcal{F}_n} \mathsf{metr}(f, s)$.*

**Lemma 3** (Bernstein's inequality for sub-exponential random variables (Wainwright, 2019)). *If $X_1, \ldots X_n$ are sub-exponential random variables, the Bernstein's inequality in Lemma 1 holds with*

$$\sigma^2 = \frac{1}{n} \sum_{i=1}^{n} \|X_i\|_{Orlicz_1}^2, \quad B = \max_{1 \leq i \leq n} \|X_i\|_{Orlicz_1}.$$

**Lemma 4** (Vector Bernstein's inequality (Pinelis, 1994; Smale & Zhou, 2007; Xu & Zeevi, 2020)). *Let $\{X_i\}_{i=1}^{n}$ be a sequence of i.i.d. random variables taking values in a real separable Hilbert space. Assume that $\mathbb{E}[X_i] = \mu$, $\mathbb{E}[\|X_i - \mu\|^2] = \sigma^2$, $\forall 1 \leq i \leq n$, we say that vector Bernstein's condition with parameter B holds if for all $1 \leq i \leq n$,*

$$\mathbb{E}[\|X_i - \mu\|^k] \leq \frac{1}{2} k! \sigma^2 B^{k-2}, \quad \forall 2 \leq k \leq n. \tag{10}$$

*If this condition holds, then for all $\delta \in (0, 1)$, with probability at least $1 - \delta$ we have*

$$\left\| \frac{1}{n} \sum_{i=1}^{n} X_i - \mu \right\| \leq \sqrt{\frac{2\sigma^2 \log(\frac{2}{\delta})}{n}} + \frac{B \log \frac{2}{\delta}}{n}. \tag{11}$$

**Definition 10** (Covering number (Wainwright, 2019)). *Assume $(\mathcal{M}, \mathsf{metr})$ is a metric space and $\mathcal{F} \subseteq \mathcal{M}$. For any $\epsilon > 0$, a set $\mathcal{F}_c$ is called an $\epsilon$-cover of $\mathcal{F}$ if for any $f \in \mathcal{F}$ we have an element $g \in \mathcal{F}_c$ such that $\mathsf{metr}(f, g) \leq \epsilon$. We denote $N(\mathcal{F}, \mathsf{metr}, \epsilon)$ the covering number as the cardinality of the minimal $\epsilon$-cover of $\mathcal{F}$:*

$$N(\mathcal{F}, \mathsf{metr}, \epsilon) = \min\{|\mathcal{F}_c| : \mathcal{F}_c \text{is an } \epsilon\text{-cover of } \mathcal{F}\}.$$

**Lemma 5** (Dudley's integral bound for $\gamma_\alpha$ functional (Talagrand, 1996)). *There exist a constant $C_\alpha$ depending only on $\alpha$ such that*

$$\gamma_\alpha(\mathcal{F}, \mathsf{metr}) \leq C_\alpha \int_0^{+\infty} (\log N(\mathcal{F}, \mathsf{metr}, \epsilon))^{\frac{1}{\alpha}} d\epsilon$$

**Lemma 6** (Generic chaining for a process with mixed tail increments in (Dirksen, 2015)). *If $(X_f)_{f \in \mathcal{F}}$ has mixed sub-Gaussian-sub-exponential increments with respect to the pair $(\mathsf{metr}_1, \mathsf{metr}_2)$, there are absolute constants $c, C > 0$ such that for $\forall \delta \in (0, 1)$, with probability at least $1 - \delta$*

$$\sup_{\theta \in \Theta} \|X_f - X_{f_0}\| \leq C(\gamma_2(\mathcal{F}, \mathsf{metr}_2) + \gamma_1(\mathcal{F}, \mathsf{metr}_1)) +$$

$$c \left( \sqrt{\log \frac{1}{\delta}} + \sup_{f_1, f_2 \in \mathcal{F}} [\mathsf{metr}_2(f_1, f_2)] + \log \frac{1}{\delta} \sup_{f_1, f_2 \in \mathcal{F}} [\mathsf{metr}_1(f_1, f_2)] \right).$$

**Lemma 7** ("Self-bounding" property for smooth function (Srebro et al., 2010)). *For a $\beta$-smooth and non-negative function $f : \mathbf{w} \to \mathbb{R}$, for all $\mathbf{w} \in \mathcal{W}$:*

$$\|\nabla f(\mathbf{w})\| \leq \sqrt{4\beta f(\mathbf{w})}$$

# B  SOME BASIC LEMMATA IN MINIMAX PROBLEMS

**Lemma 8** (Smoothness for primal function (Nouiehed et al., 2019)). *Suppose Assumption 1 holds, then the function $\Phi(\mathbf{x})$ and $\Phi_S(\mathbf{x})$ is $\beta + \frac{\beta^2}{\mu_{\mathbf{y}}}$-smooth.*

**Lemma 9** (PL condition for primal function (Yang et al., 2020)). *For NC-SC setting, suppose Assumption 1 holds. Assume that the population risk $F(\mathbf{x}, \mathbf{y})$ satisfies Assumption 4 with parameter $\mu_{\mathbf{x}}$, then function $\Phi(\mathbf{x})$ satisfies the PL condition with $\mu_{\mathbf{x}}$, which means that for all $\mathbf{x} \in \mathcal{X}$*

$$\Phi(\mathbf{x}) - \Phi(\mathbf{x}^*) \leq \frac{1}{2\mu_{\mathbf{x}}} \|\nabla \Phi(\mathbf{x})\|^2.$$

*For SC-SC setting, suppose Assumption 5 holds, then function $\Phi(\mathbf{x})$ and $\Phi_S(\mathbf{x})$ satisfy the PL condition with $\mu_{\mathbf{x}}$, which means that for all $\mathbf{x} \in \mathcal{X}$*

$$\Phi(\mathbf{x}) - \Phi(\mathbf{x}^*) \leq \frac{1}{2\mu_{\mathbf{x}}} \|\nabla \Phi(\mathbf{x})\|^2 \quad and \quad \Phi_S(\mathbf{x}) - \Phi_S(\mathbf{x}^*) \leq \frac{1}{2\mu_{\mathbf{x}}} \|\nabla \Phi_S(\mathbf{x})\|^2.$$

**Definition 11.** *For a given* $\mathbf{x}$*, the empirical optimal point* $\mathbf{y}^*(\mathbf{x})$ *and the population optimal point* $\mathbf{y}_S^*(\mathbf{x})$ *are given as follows,*

$$\mathbf{y}^*(\mathbf{x}) := \arg\max_{\mathbf{y}\in\mathcal{Y}} F(\mathbf{x}, \mathbf{y}) \quad and \quad \mathbf{y}_S^*(\mathbf{x}) := \arg\max_{\mathbf{y}\in\mathcal{Y}} F_S(\mathbf{x}, \mathbf{y}).$$

**Remark 13.** *According to the definition, we can easily derive the following equations that* $\mathbf{y}^* = \mathbf{y}^*(\mathbf{x}^*) = \arg\max_{\mathbf{y}\in\mathcal{Y}} F(\mathbf{x}^*, \mathbf{y})$ *if* $(\mathbf{x}^*, \mathbf{y}^*)$ *is the solution to (1) and* $\hat{\mathbf{y}}^* = \mathbf{y}_S^*(\hat{\mathbf{x}}^*) = \arg\max_{\mathbf{y}\in\mathcal{Y}} F_S(\hat{\mathbf{x}}^*, \mathbf{y})$ *if* $(\hat{\mathbf{x}}^*, \hat{\mathbf{y}}^*)$ *is the solution to (2).*

**Lemma 10** (Concentration of $\mathbf{y}^*$ (Zhang et al., 2022))**.** *For* $\mathbf{y}^*(\mathbf{x})$ *and* $\mathbf{y}_S^*(\mathbf{x})$ *defined in Definition 11, with Assumption 1, we have* $\forall \mathbf{x} \in \mathcal{X}$,

$$\|\mathbf{y}^*(\mathbf{x}) - \mathbf{y}_S^*(\mathbf{x})\| \le \frac{1}{\mu_{\mathbf{y}}} \|\nabla_{\mathbf{y}} F(\mathbf{x}, \mathbf{y}^*(\mathbf{x})) - \nabla_{\mathbf{y}} F_S(\mathbf{x}, \mathbf{y}^*(\mathbf{x}))\|.$$

**Lemma 11.** *Suppose Assumption 1 and 3 hold, For any* $\mathbf{x} \in \mathcal{X}$*, With probability at least* $1 - \delta$ *we have the following inequalities*

$$\|\mathbf{y}^*(\mathbf{x}) - \mathbf{y}_S^*(\mathbf{x})\| \le \frac{1}{\mu_{\mathbf{y}}} \left( \sqrt{\frac{2\mathbb{E}[\|\nabla_{\mathbf{y}} f(\mathbf{x}, \mathbf{y}^*(\mathbf{x}); \mathbf{z})\|^2] \log\frac{2}{\delta}}{n}} + \frac{B_{\mathbf{y}^*} \log\frac{2}{\delta}}{n} \right).$$

*Proof of Lemma 11.* From Bernstein's inequality for vectors (Lemma 4), applying Lemma 10, For any $\mathbf{x} \in \mathcal{X}$, With probability at least $1 - \delta$ we have the following inequalities

$$\|\mathbf{y}^*(\mathbf{x}) - \mathbf{y}_S^*(\mathbf{x})\| \le \frac{1}{\mu_{\mathbf{y}}} \|\nabla_{\mathbf{y}} F(\mathbf{x}, \mathbf{y}^*(\mathbf{x})) - \nabla_{\mathbf{y}} F_S(\mathbf{x}, \mathbf{y}^*(\mathbf{x}))\|$$

$$\le \frac{1}{\mu_{\mathbf{y}}} \left( \sqrt{\frac{2\mathbb{E}[\|\nabla_{\mathbf{y}} f(\mathbf{x}, \mathbf{y}^*(\mathbf{x}); \mathbf{z})\|^2] \log\frac{2}{\delta}}{n}} + \frac{B_{\mathbf{y}^*} \log\frac{2}{\delta}}{n} \right).$$

Notice that Bernstein inequality for vectors (Lemma 4) under Assumption 3 can be used here because for all $\mathbf{x} \in \mathcal{X}$, $\mathbf{y}^*(x)$ and $F_S(\cdot)$ are independent.

Next, we need to point out that we only need pointwise convergence for any fixed $\mathbf{x}$ instead of unifrom convergence on $\mathcal{X}$. We take the standard minimization problem as an example. For population risk $R(\theta) = \mathbb{E}_z \ell(\theta, z)$, empirical risk $r(\theta) = \frac{1}{n}\sum_{i=1}^n \ell(\theta, z_i)$ and fixed $\theta$, we can directly apply concentration inequality. But we can't apply them to $\theta_{\mathrm{ERM}}$ as it is a function of the dataset. We need to establish the sup that $R(\theta_{\mathrm{ERM}}) - r(\theta_{\mathrm{ERM}}) \le \sup_{\theta_{sup}}[R(\theta_{sup}) - r(\theta_{sup})]$, where $\theta_{sup} = \arg\min_\theta R(\theta) - r(\theta)$. Yet, as the dataset changes, the parameters $\theta_{sup}$ change accordingly. Thus, we require uniform convergence for function $R - r$. In Lemma 11 of our proof, for any $\mathbf{x}$, when the dataset changes, $\mathbf{x}$ and $\mathbf{y}^*(\mathbf{x})$ remain unchanged, and $\nabla F_S(\mathbf{x}, \mathbf{y}^*(\mathbf{x}))$ are the only altered random variables/vectors. Consequently, we can directly apply the Bernstein inequality, thus we only need pointwise convergence for the function $y^*(x) - y_S^*(x)$ w.r.t. $x$, which suffices.

$\square$

**Lemma 12** (Zhang et al. (2021))**.** *For* $\mathbf{y}^*$ *and* $\mathbf{y}_S^*$ *defined in Definition 11, with Assumption 1, then for* $\forall \mathbf{x}_1, \mathbf{x}_2 \in \mathcal{X}$,

$$\|\mathbf{y}^*(\mathbf{x}_1) - \mathbf{y}^*(\mathbf{x}_2)\| \le \frac{\beta}{\mu_{\mathbf{y}}} \|\mathbf{x}_1 - \mathbf{x}_2\|.$$

**Lemma 13.** *Suppose a function* $f : \mathcal{X} \times \mathcal{Y} \times \mathcal{Z} \to \mathbb{R}$ *is* $\beta$*-smooth in* $(\mathbf{x}, \mathbf{y})$ *and the function* $f$ *is* $\mu_{\mathbf{y}}$*-strongly concave in* $\mathbf{y} \in \mathcal{Y}$ *for any* $\mathbf{x} \in \mathcal{X}$ *and* $\mathbf{z} \in \mathcal{Z}$*. Then we have* $\frac{\mathbf{u}^{\mathrm{T}}(\nabla_{\mathbf{x}} f(\mathbf{x}_1, \mathbf{y}^*(\mathbf{x}_1); \mathbf{z}) - \nabla_{\mathbf{x}} f(\mathbf{x}_2, \mathbf{y}^*(\mathbf{x}_2); \mathbf{z}))}{\|\mathbf{x}_1 - \mathbf{x}_2\|}$ *is a* $\frac{\beta(\mu_{\mathbf{y}}+\beta)}{\mu_{\mathbf{y}} \ln 2}$*-sub-exponential random vector. That is for any unit vector* $\mathbf{u} \in B(0, 1)$ *and* $\mathbf{x}_1, \mathbf{x}_2 \in \mathcal{X}$,

$$\mathbb{E}\left\{ \exp\left( \frac{|\mathbf{u}^{\mathrm{T}}(\nabla_{\mathbf{x}} f(\mathbf{x}_1, \mathbf{y}^*(\mathbf{x}_1); \mathbf{z}) - \nabla_{\mathbf{x}} f(\mathbf{x}_2, \mathbf{y}^*(\mathbf{x}_2); \mathbf{z}))|}{\frac{\beta(\mu_{\mathbf{y}}+\beta)}{\mu_{\mathbf{y}} \ln 2} \|\mathbf{x}_1 - \mathbf{x}_2\|} \right) \right\} \le 2.$$

*Proof of Lemma 13.* According to Definition 3, for any sample $\mathbf{z} \in \mathcal{Z}$ and $\mathbf{x}_1, \mathbf{x}_2 \in \mathcal{X}$, we have

$$
\begin{aligned}
&\|\nabla_{\mathbf{x}} f(\mathbf{x}_1, \mathbf{y}^*(\mathbf{x}_1); \mathbf{z}) - \nabla_{\mathbf{x}} f(\mathbf{x}_2, \mathbf{y}^*(\mathbf{x}_2); \mathbf{z})\| \\
\leq & \beta \|\mathbf{x}_1 - \mathbf{x}_2\| + \beta \|\mathbf{y}^*(\mathbf{x}_1) - \mathbf{y}^*(\mathbf{x}_2)\| \\
\leq & \beta \|\mathbf{x}_1 - \mathbf{x}_2\| + \frac{\beta^2}{\mu_{\mathbf{y}}} \|\mathbf{x}_1 - \mathbf{x}_2\| \\
= & \frac{\beta(\mu_{\mathbf{y}} + \beta)}{\mu_{\mathbf{y}}} \|\mathbf{x}_1 - \mathbf{x}_2\|,
\end{aligned}
$$

where the first inequality uses the smoothness and the second inequality applies Lemma 12.

Then, for any unit vector $\mathbf{u} \in B(0, 1)$, we have

$$
\begin{aligned}
&\left|\mathbf{u}^{\mathrm{T}}(\nabla_{\mathbf{x}} f(\mathbf{x}_1, \mathbf{y}^*(\mathbf{x}_1); \mathbf{z}) - \nabla_{\mathbf{x}} f(\mathbf{x}_2, \mathbf{y}^*(\mathbf{x}_2); \mathbf{z}))\right| \\
\leq & \|\mathbf{u}^{\mathrm{T}}\| \|\nabla_{\mathbf{x}} f(\mathbf{x}_1, \mathbf{y}^*(\mathbf{x}_1); \mathbf{z}) - \nabla_{\mathbf{x}} f(\mathbf{x}_2, \mathbf{y}^*(\mathbf{x}_2); \mathbf{z})\| \\
\leq & \frac{\beta(\mu_{\mathbf{y}} + \beta)}{\mu_{\mathbf{y}}} \|\mathbf{x}_1 - \mathbf{x}_2\|,
\end{aligned}
$$

which implies

$$
\frac{\left|\mathbf{u}^{\mathrm{T}}(\nabla_{\mathbf{x}} f(\mathbf{x}_1, \mathbf{y}^*(\mathbf{x}_1); \mathbf{z}) - \nabla_{\mathbf{x}} f(\mathbf{x}_2, \mathbf{y}^*(\mathbf{x}_2); \mathbf{z}))\right|}{\frac{\beta(\mu_{\mathbf{y}} + \beta)}{\mu_{\mathbf{y}}} \|\mathbf{x}_1 - \mathbf{x}_2\|} \leq 1.
$$

Then we get

$$
\mathbb{E}\left\{ \exp\left( \frac{\left|\mathbf{u}^{\mathrm{T}}(\nabla_{\mathbf{x}} f(\mathbf{x}_1, \mathbf{y}^*(\mathbf{x}_1); \mathbf{z}) - \nabla_{\mathbf{x}} f(\mathbf{x}_2, \mathbf{y}^*(\mathbf{x}_2); \mathbf{z}))\right|}{\frac{\beta(\mu_{\mathbf{y}} + \beta)}{\mu_{\mathbf{y}} \ln 2} \|\mathbf{x}_1 - \mathbf{x}_2\|} \right) \right\} \leq 2.
$$

The proof is complete. $\qquad\square$

**Lemma 14.** *Suppose Assumption 1 holds, we have the following inequality that for all $\mathbf{x} \in \mathcal{X}$ and for any $\delta \in (0, 1)$, with probability at lest $1 - \delta$,*

$$
\|(\nabla_{\mathbf{x}} F(\mathbf{x}, \mathbf{y}^*(\mathbf{x})) - \nabla_{\mathbf{x}} F_S(\mathbf{x}, \mathbf{y}^*(\mathbf{x}))) - (\nabla_{\mathbf{x}} F(\mathbf{x}^*, \mathbf{y}^*) - \nabla_{\mathbf{x}} F_S(\mathbf{x}^*, \mathbf{y}^*))\|
$$

$$
\leq \frac{C\beta(\mu_{\mathbf{y}} + \beta)}{\mu_{\mathbf{y}}} \max\left\{ \|\mathbf{x} - \mathbf{x}^*\|, \frac{1}{n} \right\} \times \left( \sqrt{\frac{d + \log \frac{4 \log_2(\sqrt{2} R_1 n + 1)}{\delta}}{n}} + \frac{d + \log \frac{4 \log_2(\sqrt{2} R_1 n + 1)}{\delta}}{n} \right),
$$

*where $C$ is an absolute constant.*

*Proof of Lemma 14.* We define $\mathcal{V} = \{\mathbf{v} \in \mathbb{R}^d : \|\mathbf{v}\| \leq \max\{R_1, \frac{1}{n}\}\}$. For all $(\mathbf{x}, \mathbf{v}) \in \mathcal{X} \times \mathcal{V}$, let $g_{(\mathbf{x}, \mathbf{v})}(\mathbf{z}) = (\nabla_{\mathbf{x}} f(\mathbf{x}, \mathbf{y}^*(\mathbf{x}); \mathbf{z}) - \nabla_{\mathbf{x}} f(\mathbf{x}^*, \mathbf{y}^*; \mathbf{z}))^{\mathrm{T}} \mathbf{v}$. Then for any $(\mathbf{x}_1, \mathbf{v}_1)$ and $(\mathbf{x}_2, \mathbf{v}_2) \in \mathcal{X} \times \mathcal{V}$, we define the following norm on the product space $\mathcal{X} \times \mathcal{V}$

$$
\|(\mathbf{x}_1, \mathbf{v}_1) - (\mathbf{x}_2, \mathbf{v}_2)\|_{\mathcal{X} \times \mathcal{V}} = \left( \|\mathbf{x}_1 - \mathbf{x}_2\|^2 + \|\mathbf{v}_1 - \mathbf{v}_2\|^2 \right)^{\frac{1}{2}}.
$$

Then we define a ball on the product space $\mathcal{X} \times \mathcal{V}$ that $B(\sqrt{r}) = \{(\mathbf{x}, \mathbf{v}) \in \mathcal{X} \times \mathcal{V} : \|\mathbf{x} - \mathbf{x}^*\|^2 + \|\mathbf{v}\|^2 \leq r\}$. Given any $(\mathbf{x}_1, \mathbf{v}_1)$ and $(\mathbf{x}_2, \mathbf{v}_2) \in B(\sqrt{r})$, we have the following decomposition

$$
\begin{aligned}
& g_{(\mathbf{x}_1, \mathbf{v}_1)}(\mathbf{z}) - g_{(\mathbf{x}_2, \mathbf{v}_2)}(\mathbf{z}) \\
= & (\nabla_{\mathbf{x}} f(\mathbf{x}_1, \mathbf{y}^*(\mathbf{x}_1); \mathbf{z}) - \nabla_{\mathbf{x}} f(\mathbf{x}^*, \mathbf{y}^*; \mathbf{z}))^{\mathrm{T}} \mathbf{v}_1 \\
& - (\nabla_{\mathbf{x}} f(\mathbf{x}_2, \mathbf{y}^*(\mathbf{x}_2); \mathbf{z}) - \nabla_{\mathbf{x}} f(\mathbf{x}^*, \mathbf{y}^*; \mathbf{z}))^{\mathrm{T}} \mathbf{v}_2 \\
= & (\nabla_{\mathbf{x}} f(\mathbf{x}_1, \mathbf{y}^*(\mathbf{x}_1); \mathbf{z}) - \nabla_{\mathbf{x}} f(\mathbf{x}^*, \mathbf{y}^*; \mathbf{z}))^{\mathrm{T}} (\mathbf{v}_1 - \mathbf{v}_2) \\
& + (\nabla_{\mathbf{x}} f(\mathbf{x}_1, \mathbf{y}^*(\mathbf{x}_1); \mathbf{z}) - \nabla_{\mathbf{x}} f(\mathbf{x}^*, \mathbf{y}^*; \mathbf{z}))^{\mathrm{T}} \mathbf{v}_2 \\
& - (\nabla_{\mathbf{x}} f(\mathbf{x}_2, \mathbf{y}^*(\mathbf{x}_2); \mathbf{z}) - \nabla_{\mathbf{x}} f(\mathbf{x}^*, \mathbf{y}^*; \mathbf{z}))^{\mathrm{T}} \mathbf{v}_2 \\
= & (\nabla_{\mathbf{x}} f(\mathbf{x}_1, \mathbf{y}^*(\mathbf{x}_1); \mathbf{z}) - \nabla_{\mathbf{x}} f(\mathbf{x}^*, \mathbf{y}^*; \mathbf{z}))^{\mathrm{T}} (\mathbf{v}_1 - \mathbf{v}_2) \\
& + (\nabla_{\mathbf{x}} f(\mathbf{x}_1, \mathbf{y}^*(\mathbf{x}_1); \mathbf{z}) - \nabla_{\mathbf{x}} f(\mathbf{x}_2, \mathbf{y}^*(\mathbf{x}_2); \mathbf{z}))^{\mathrm{T}} \mathbf{v}_2.
\end{aligned}
$$

Since $(\mathbf{x}_1, \mathbf{v}_1)$ and $(\mathbf{x}_2, \mathbf{v}_2) \in B(\sqrt{r})$, we have

$$\|\mathbf{x}_1 - \mathbf{x}^*\| \|\mathbf{v}_1 - \mathbf{v}_2\| \leq \sqrt{r} \|\mathbf{v}_1 - \mathbf{v}_2\| \leq \sqrt{r} \|(\mathbf{x}_1, \mathbf{v}_1) - (\mathbf{x}_2, \mathbf{v}_2)\|_{\mathcal{X} \times \mathcal{V}}. \tag{12}$$

Next, from Assumption 1 and Lemma 13, we know that $\frac{\nabla_{\mathbf{x}} f(\mathbf{x}_1, \mathbf{y}^*(\mathbf{x}_1); \mathbf{z}) - \nabla_{\mathbf{x}} f(\mathbf{x}_2, \mathbf{y}^*(\mathbf{x}_2); \mathbf{z})}{\|\mathbf{x}_1 - \mathbf{x}_2\|}$ is a $\frac{\beta(\mu_{\mathbf{y}} + \beta)}{\mu_{\mathbf{y}} \ln 2}$-sub-exponential random vector for all $\mathbf{x}_1, \mathbf{x}_2 \in \mathcal{X}$, which means that

$$\mathbb{E}\left\{ \exp\left( \frac{(\nabla_{\mathbf{x}} f(\mathbf{x}_1, \mathbf{y}^*(\mathbf{x}_1); \mathbf{z}) - \nabla_{\mathbf{x}} f(\mathbf{x}^*, \mathbf{y}^*(\mathbf{x}^*); \mathbf{z}))^{\mathrm{T}} (\mathbf{v}_1 - \mathbf{v}_2)}{\frac{\beta(\mu_{\mathbf{y}} + \beta)}{\mu_{\mathbf{y}} \ln 2} \|\mathbf{x}_1 - \mathbf{x}^*\| \|\mathbf{v}_1 - \mathbf{v}_2\|} \right) \right\} \leq 2. \tag{13}$$

We combine (12) and (13), according to Definition 6, we can easily deduce that $(\nabla_{\mathbf{x}} f(\mathbf{x}_1, \mathbf{y}^*(\mathbf{x}_1); \mathbf{z}) - \nabla_{\mathbf{x}} f(\mathbf{x}^*, \mathbf{y}^*; \mathbf{z}))^{\mathrm{T}} (\mathbf{v}_1 - \mathbf{v}_2)$ is $\frac{\beta\sqrt{r}(\mu_{\mathbf{y}} + \beta)}{\mu_{\mathbf{y}} \ln 2} \|(\mathbf{x}_1, \mathbf{v}_1) - (\mathbf{x}_2, \mathbf{v}_2)\|_{\mathcal{X} \times \mathcal{V}}$-sub-exponential. Similarly, we can derive that

$$\|\mathbf{x}_1 - \mathbf{x}_2\| \|\mathbf{v}_2\| \leq \sqrt{r} \|\mathbf{x}_1 - \mathbf{x}_2\| \leq \sqrt{r} \|(\mathbf{x}_1, \mathbf{v}_1) - (\mathbf{x}_2, \mathbf{v}_2)\|_{\mathcal{X} \times \mathcal{V}}.$$

Also, there holds that

$$\mathbb{E}\left\{ \exp\left( \frac{(\nabla_{\mathbf{x}} f(\mathbf{x}_1, \mathbf{y}^*(\mathbf{x}_1); \mathbf{z}) - \nabla_{\mathbf{x}} f(\mathbf{x}_2, \mathbf{y}^*(\mathbf{x}_2); \mathbf{z}))^{\mathrm{T}} (\mathbf{v}_2)}{\frac{\beta(\mu_{\mathbf{y}} + \beta)}{\mu_{\mathbf{y}} \ln 2} \|\mathbf{x}_1 - \mathbf{x}_2\| \|\mathbf{v}_2\|} \right) \right\} \leq 2.$$

Thus, we have that $(\nabla_{\mathbf{x}} f(\mathbf{x}_1, \mathbf{y}^*(\mathbf{x}_1); \mathbf{z}) - \nabla_{\mathbf{x}} f(\mathbf{x}_2, \mathbf{y}^*(\mathbf{x}_2); \mathbf{z}))^{\mathrm{T}} (\mathbf{v}_2)$ is also $\frac{\beta\sqrt{r}(\mu_{\mathbf{y}} + \beta)}{\mu_{\mathbf{y}} \ln 2} \|(\mathbf{x}_1, \mathbf{v}_1) - (\mathbf{x}_2, \mathbf{v}_2)\|_{\mathcal{X} \times \mathcal{V}}$-sub-exponential. Now for any $(\mathbf{x}_1, \mathbf{v}_1)$ and $(\mathbf{x}_2, \mathbf{v}_2) \in B(\sqrt{r})$, we know

$$\mathbb{E}\left\{ \exp\left( \frac{g_{(\mathbf{x}_1, \mathbf{v}_1)}(\mathbf{z}) - g_{(\mathbf{x}_2, \mathbf{v}_2)}(\mathbf{z})}{\frac{2\beta\sqrt{r}(\mu_{\mathbf{y}} + \beta)}{\mu_{\mathbf{y}} \ln 2} \|(\mathbf{x}_1, \mathbf{v}_1) - (\mathbf{x}_2, \mathbf{v}_2)\|_{\mathcal{X} \times \mathcal{V}}} \right) \right\}$$

$$\leq \mathbb{E}\left\{ \frac{1}{2} \exp\left( \frac{(\nabla_{\mathbf{x}} f(\mathbf{x}_1, \mathbf{y}^*(\mathbf{x}_1); \mathbf{z}) - \nabla_{\mathbf{x}} f(\mathbf{x}^*, \mathbf{y}^*; \mathbf{z}))^{\mathrm{T}} (\mathbf{v}_1 - \mathbf{v}_2)}{\frac{\beta\sqrt{r}(\mu_{\mathbf{y}} + \beta)}{\mu_{\mathbf{y}} \ln 2} \|(\mathbf{x}_1, \mathbf{v}_1) - (\mathbf{x}_2, \mathbf{v}_2)\|_{\mathcal{X} \times \mathcal{V}}} \right) \right\} \tag{14}$$

$$+ \mathbb{E}\left\{ \frac{1}{2} \exp\left( \frac{(\nabla_{\mathbf{x}} f(\mathbf{x}_1, \mathbf{y}^*(\mathbf{x}_1); \mathbf{z}) - \nabla_{\mathbf{x}} f(\mathbf{x}_2, \mathbf{y}^*(\mathbf{x}_2); \mathbf{z}))^{\mathrm{T}} (\mathbf{v}_2)}{\frac{\beta\sqrt{r}(\mu_{\mathbf{y}} + \beta)}{\mu_{\mathbf{y}} \ln 2} \|(\mathbf{x}_1, \mathbf{v}_1) - (\mathbf{x}_2, \mathbf{v}_2)\|_{\mathcal{X} \times \mathcal{V}}} \right) \right\} \leq 2,$$

where the first inequality follows from Jensen's inequality. (14) implies that $g_{(\mathbf{x}_1, \mathbf{v}_1)}(\mathbf{z}) - g_{(\mathbf{x}_2, \mathbf{v}_2)}(\mathbf{z})$ is a $\frac{2\beta\sqrt{r}(\mu_{\mathbf{y}} + \beta)}{\mu_{\mathbf{y}} \ln 2} \|(\mathbf{x}_1, \mathbf{v}_1) - (\mathbf{x}_2, \mathbf{v}_2)\|_{\mathcal{X} \times \mathcal{V}}$-sub-exponential random variable, for which we have

$$\|g_{(\mathbf{x}_1, \mathbf{v}_1)}(\mathbf{z}) - g_{(\mathbf{x}_2, \mathbf{v}_2)}(\mathbf{z})\|_{\mathrm{Orlicz}_1} \leq \frac{2\beta\sqrt{r}(\mu_{\mathbf{y}} + \beta)}{\mu_{\mathbf{y}} \ln 2} \|(\mathbf{x}_1, \mathbf{v}_1) - (\mathbf{x}_2, \mathbf{v}_2)\|_{\mathcal{X} \times \mathcal{V}}. \tag{15}$$

From Bernstein inequality for sub-exponential variables (Lemma 3), for any fixed $u \leq 0$ and $(\mathbf{x}_1. \mathbf{v}_1), (\mathbf{x}_2, \mathbf{v}_2) \in \mathcal{X} \times \mathcal{V}$, we have

$$Pr\left( |(\mathbb{P} - \mathbb{P}_n)[g_{(\mathbf{x}_1, \mathbf{v}_1)}(\mathbf{z}) - g_{(\mathbf{x}_2, \mathbf{v}_2)}(\mathbf{z})]| \geq \frac{2\beta\sqrt{r}(\mu_{\mathbf{y}} + \beta)}{\mu_{\mathbf{y}} \ln 2} \|(\mathbf{x}_1, \mathbf{v}_1) - (\mathbf{x}_2, \mathbf{v}_2)\|_{\mathcal{X} \times \mathcal{V}} \sqrt{\frac{2u}{n}} \right.$$

$$\left. + \frac{2u\beta\sqrt{r}(\mu_{\mathbf{y}} + \beta)}{n\mu_{\mathbf{y}} \ln 2} \|(\mathbf{x}_1, \mathbf{v}_1) - (\mathbf{x}_2, \mathbf{v}_2)\|_{\mathcal{X} \times \mathcal{V}} \right) \leq 2e^{-u}. \tag{16}$$

According to Definition 8, we know that $(\mathbb{P} - \mathbb{P}_n) g_{(\mathbf{x}, \mathbf{v})}(\mathbf{z})$ is a mixed sub-Gaussian-sub-exponential increments w.r.t. the metrics $\left( \frac{2\beta\sqrt{r}(\mu_{\mathbf{y}} + \beta)}{n\mu_{\mathbf{y}} \ln 2} \| \cdot \|_{\mathcal{X} \times \mathcal{V}}, \frac{2\beta\sqrt{2r}(\mu_{\mathbf{y}} + \beta)}{\sqrt{n}\mu_{\mathbf{y}} \ln 2} \| \cdot \|_{\mathcal{X} \times \mathcal{V}} \right)$ from (16).

Then from the generic chaining for a process with mixed tail increments in Lemma 6, we have the following inequality that for all $\delta \in (0, 1)$, with probability at least $1 - \delta$

$$\sup_{\|\mathbf{x}_1 - \mathbf{x}^*\|^2 + \|\mathbf{v}\|^2 \leq r} |(\mathbb{P} - \mathbb{P}_n)g_{(\mathbf{x}, \mathbf{v})}(\mathbf{z})|$$

$$\leq C\left(\gamma_2\left(B(\sqrt{r}), \frac{2\beta\sqrt{2r}(\mu_{\mathbf{y}} + \beta)}{\sqrt{n}\mu_{\mathbf{y}} \ln 2}\| \cdot \|_{\mathcal{X} \times \mathcal{V}}\right) + \gamma_1\left(B(\sqrt{r}), \frac{2\beta\sqrt{r}(\mu_{\mathbf{y}} + \beta)}{n\mu_{\mathbf{y}} \ln 2}\| \cdot \|_{\mathcal{X} \times \mathcal{V}}\right)\right.$$

$$\left. + \frac{r\beta(\mu_{\mathbf{y}} + \beta)}{\mu_{\mathbf{y}} \ln 2}\sqrt{\frac{\log \frac{1}{\delta}}{n}} + \frac{r\beta(\mu_{\mathbf{y}} + \beta)}{\mu_{\mathbf{y}} \ln 2}\frac{\log \frac{1}{\delta}}{n}\right).$$

Next we use the Dudley's integral (Lemma 5) to bound the $\gamma_1$ and $\gamma_2$ functional. So there exists an absolute constant $C$ such that for any $\delta \in (0, 1)$, with probability at least $1 - \delta$

$$\sup_{\|\mathbf{x}_1 - \mathbf{x}^*\|^2 + \|\mathbf{v}\|^2 \leq r} |(\mathbb{P} - \mathbb{P}_n)g_{(\mathbf{x}, \mathbf{v})}(\mathbf{z})| \leq \frac{rC\beta(\mu_{\mathbf{y}} + \beta)}{\mu_{\mathbf{y}} \ln 2}\left(\sqrt{\frac{d + \log \frac{1}{\delta}}{n}} + \frac{d + \log \frac{1}{\delta}}{n}\right). \quad (17)$$

Then, the next step is to apply Lemma 2 to (17). We denote $T(f) = \|\mathbf{x} - \mathbf{x}^*\|^2 + \|\mathbf{v}\|^2$, $\psi(r; \delta) = \frac{rC\beta(\mu_{\mathbf{y}} + \beta)}{\mu_{\mathbf{y}} \ln 2}\left(\sqrt{\frac{d + \log \frac{1}{\delta}}{n}} + \frac{d + \log \frac{1}{\delta}}{n}\right)$. Since $\|\mathbf{x} - \mathbf{x}^*\|^2 + \|\mathbf{v}\|^2 \leq R_1^2 + R_1^2 + \frac{1}{n^2}$, we set $R^2 = 2R_1^2 + \frac{1}{n^2}$ and $r_0 = \frac{2}{n^2}$. According to Lemma 2, we have the following inequality that for any $\delta \in (0, 1)$, with probability at least $1 - \delta$, for all $\mathbf{x} \in \mathcal{X}$ and $\mathbf{v} \in \mathcal{V}$,

$$(\mathbb{P} - \mathbb{P}_n)g_{(\mathbf{x}, \mathbf{v})}(\mathbf{z})$$

$$= (\mathbb{P} - \mathbb{P}_n)[(\nabla_{\mathbf{x}} f(\mathbf{x}, \mathbf{y}^*(\mathbf{x}); \mathbf{z}) - \nabla_{\mathbf{x}} f(\mathbf{x}^*, \mathbf{y}^*; \mathbf{z}))^{\mathrm{T}}\mathbf{v}]$$

$$\leq \psi\left(\max\left\{\|\mathbf{x} - \mathbf{x}^*\|^2 + \|\mathbf{v}\|^2, \frac{2}{n^2}\right\}; \frac{\delta}{2\log_2(Rn^2)}\right)$$

$$\leq \frac{C\beta(\mu_{\mathbf{y}} + \beta)}{\mu_{\mathbf{y}} \ln 2}\max\left\{\|\mathbf{x} - \mathbf{x}^*\|^2 + \|\mathbf{v}\|^2, \frac{2}{n^2}\right\} \times \left(\sqrt{\frac{d + \log \frac{2\log_2(Rn^2)}{\delta}}{n}} + \frac{d + \log \frac{2\log_2(Rn^2)}{\delta}}{n}\right).$$

$$(18)$$

Finally, we choose $\mathbf{v} = \max\left\{\|\mathbf{x} - \mathbf{x}^*\|, \frac{1}{n}\right\}\frac{(\mathbb{P} - \mathbb{P}_n)(\nabla_{\mathbf{x}} f(\mathbf{x}, \mathbf{y}^*(\mathbf{x}); \mathbf{z}) - \nabla_{\mathbf{x}} f(\mathbf{x}^*, \mathbf{y}^*; \mathbf{z}))}{\|(\mathbb{P} - \mathbb{P}_n)(\nabla_{\mathbf{x}} f(\mathbf{x}, \mathbf{y}^*(\mathbf{x}); \mathbf{z}) - \nabla_{\mathbf{x}} f(\mathbf{x}^*, \mathbf{y}^*; \mathbf{z}))\|}$. It is clear that $\|\mathbf{v}\| = \max\{\|\mathbf{x} - \mathbf{x}^*\|, \frac{1}{n}\} \leq \max\{R_1, \frac{1}{n}\}$, which belongs to the space $\mathcal{V}$. Plugging this $\mathbf{v}$ into (18), we have the following inequality that for any $\delta \in (0, 1)$, with probability at least $1 - \delta$, for all $\mathbf{x} \in \mathcal{X}$,

$$\|(\mathbb{P} - \mathbb{P}_n)(\nabla_{\mathbf{x}} f(\mathbf{x}, \mathbf{y}^*(\mathbf{x}); \mathbf{z}) - \nabla_{\mathbf{x}} f(\mathbf{x}^*, \mathbf{y}^*; \mathbf{z}))\|$$

$$\leq \frac{C\beta(\mu_{\mathbf{y}} + \beta)}{\mu_{\mathbf{y}} \ln 2}\max\left\{\|\mathbf{x} - \mathbf{x}^*\|, \frac{1}{n}\right\} \times \left(\sqrt{\frac{d + \log \frac{2\log_2(Rn^2)}{\delta}}{n}} + \frac{d + \log \frac{2\log_2(Rn^2)}{\delta}}{n}\right)$$

$$= \frac{C\beta(\mu_{\mathbf{y}} + \beta)}{\mu_{\mathbf{y}} \ln 2}\max\left\{\|\mathbf{x} - \mathbf{x}^*\|, \frac{1}{n}\right\} \times \left(\sqrt{\frac{d + \log \frac{4\log_2(\sqrt{2}R_1n + 1)}{\delta}}{n}} + \frac{d + \log \frac{4\log_2(\sqrt{2}R_1n + 1)}{\delta}}{n}\right).$$

$$(19)$$

Finally we have

$$\|(\nabla_{\mathbf{x}} F(\mathbf{x}, \mathbf{y}^*(\mathbf{x})) - \nabla_{\mathbf{x}} F_S(\mathbf{x}, \mathbf{y}^*(\mathbf{x}))) - (\nabla_{\mathbf{x}} F(\mathbf{x}^*, \mathbf{y}^*) - \nabla_{\mathbf{x}} F_S(\mathbf{x}^*, \mathbf{y}^*))\|$$

$$= \|(\mathbb{P} - \mathbb{P}_n)(\nabla_{\mathbf{x}} f(\mathbf{x}, \mathbf{y}^*(\mathbf{x}); \mathbf{z}) - \nabla_{\mathbf{x}} f(\mathbf{x}^*, \mathbf{y}^*(\mathbf{x}^*); \mathbf{z}))\|$$

$$\leq \frac{C\beta(\mu_{\mathbf{y}} + \beta)}{\mu_{\mathbf{y}}}\max\left\{\|\mathbf{x} - \mathbf{x}^*\|, \frac{1}{n}\right\} \times \left(\sqrt{\frac{d + \log \frac{4\log_2(\sqrt{2}R_1n + 1)}{\delta}}{n}} + \frac{d + \log \frac{4\log_2(\sqrt{2}R_1n + 1)}{\delta}}{n}\right),$$

where $C$ is an absolute constant.

The proof is complete. $\qquad \square$

**Lemma 15.** *Suppose Assumption 1 holds, we have the following inequality that for all $\mathbf{x} \in \mathcal{X}$ and for any $\delta \in (0, 1)$, with probability at lest $1 - \delta$,*

$$\|\nabla_{\mathbf{x}} F(\mathbf{x}, \mathbf{y}^*(\mathbf{x})) - \nabla_{\mathbf{x}} F_S(\mathbf{x}, \mathbf{y}^*(\mathbf{x}))\| \leq \sqrt{\frac{2\mathbb{E}\|\nabla_{\mathbf{x}} f(\mathbf{x}^*, \mathbf{y}^*(\mathbf{x}^*); \mathbf{z})\|^2 \log \frac{4}{\delta}}{n}} + \frac{B_{\mathbf{x}^*} \log \frac{4}{\delta}}{n}$$

$$+ \frac{C\beta(\mu_{\mathbf{y}} + \beta)}{\mu_{\mathbf{y}}} \max \left\{ \|\mathbf{x} - \mathbf{x}^*\|, \frac{1}{n} \right\} \times \left( \sqrt{\frac{d + \log \frac{8 \log_2(\sqrt{2}R_1 n + 1)}{\delta}}{n}} + \frac{d + \log \frac{8 \log_2(\sqrt{2}R_1 n + 1)}{\delta}}{n} \right).$$

*Proof of Lemma 15.* According to Lemma 14, we have the following inequality that for any $\delta \in (0, 1)$, with probability at least $1 - \frac{\delta}{2}$

$$\|\nabla_{\mathbf{x}} F(\mathbf{x}, \mathbf{y}^*(\mathbf{x})) - \nabla_{\mathbf{x}} F_S(\mathbf{x}, \mathbf{y}^*(\mathbf{x}))\| \leq \|\nabla_{\mathbf{x}} F(\mathbf{x}^*, \mathbf{y}^*) - \nabla_{\mathbf{x}} F_S(\mathbf{x}^*, \mathbf{y}^*)\|$$

$$+ \frac{C\beta(\mu_{\mathbf{y}} + \beta)}{\mu_{\mathbf{y}}} \max \left\{ \|\mathbf{x} - \mathbf{x}^*\|, \frac{1}{n} \right\} \times \left( \sqrt{\frac{d + \log \frac{8 \log_2(\sqrt{2}R_1 n + 1)}{\delta}}{n}} + \frac{d + \log \frac{8 \log_2(\sqrt{2}R_1 n + 1)}{\delta}}{n} \right),$$

(20)

where this inequality applies norm triangle inequality. Then we need to bound $\|\nabla_{\mathbf{x}} F(\mathbf{x}^*, \mathbf{y}^*) - \nabla_{\mathbf{x}} F_S(\mathbf{x}^*, \mathbf{y}^*)\|$.

According to Lemma 4, with probability at least $1 - \frac{\delta}{2}$

$$\|\nabla_{\mathbf{x}} F(\mathbf{x}^*, \mathbf{y}^*) - \nabla_{\mathbf{x}} F_S(\mathbf{x}^*, \mathbf{y}^*)\| \leq \sqrt{\frac{2\mathbb{E}[\|\nabla_{\mathbf{x}} f(\mathbf{x}^*, \mathbf{y}^*; \mathbf{z})\|^2] \log \frac{4}{\delta}}{n}} + \frac{B_{\mathbf{x}^*} \log \frac{4}{\delta}}{n}. \quad (21)$$

Combining (20) and (21), for any $\delta \in (0, 1)$, with probability at least $1 - \delta$, we have

$$\|\nabla_{\mathbf{x}} F(\mathbf{x}, \mathbf{y}^*(\mathbf{x})) - \nabla_{\mathbf{x}} F_S(\mathbf{x}, \mathbf{y}^*(\mathbf{x}))\| \leq \sqrt{\frac{2\mathbb{E}[\|\nabla_{\mathbf{x}} f(\mathbf{x}^*, \mathbf{y}^*; \mathbf{z})\|^2] \log \frac{4}{\delta}}{n}} + \frac{B_{\mathbf{x}^*} \log \frac{4}{\delta}}{n}$$

$$+ \frac{C\beta(\mu_{\mathbf{y}} + \beta)}{\mu_{\mathbf{y}}} \max \left\{ \|\mathbf{x} - \mathbf{x}^*\|, \frac{1}{n} \right\} \times \left( \sqrt{\frac{d + \log \frac{8 \log_2(\sqrt{2}R_1 n + 1)}{\delta}}{n}} + \frac{d + \log \frac{8 \log_2(\sqrt{2}R_1 n + 1)}{\delta}}{n} \right).$$

The proof is complete. $\qquad \square$

## C    PROOFS IN SECTION 4

*Proof of Theorem 1.* Firstly, for all $\mathbf{x} \in \mathcal{X}$, we divide $\|\nabla\Phi(\mathbf{x}) - \nabla\Phi_S(\mathbf{x})\|$ into two terms

$$
\begin{aligned}
&\|\nabla\Phi(\mathbf{x}) - \nabla\Phi_S(\mathbf{x})\| \\
&= \left\| \mathbb{E}_{\mathbf{z}} \nabla_{\mathbf{x}} f(\mathbf{x}, \mathbf{y}^*(\mathbf{x}); \mathbf{z}) - \frac{1}{n} \sum_{i=1}^n \nabla_{\mathbf{x}} f(\mathbf{x}, \mathbf{y}_S^*(\mathbf{x}); \mathbf{z}_i) \right\| \\
&= \left\| \mathbb{E}_{\mathbf{z}} \nabla_{\mathbf{x}} f(\mathbf{x}, \mathbf{y}^*(\mathbf{x}); \mathbf{z}) - \frac{1}{n} \sum_{i=1}^n \nabla_{\mathbf{x}} f(\mathbf{x}, \mathbf{y}^*(\mathbf{x}); \mathbf{z}_i) \right. \\
&\qquad \left. + \frac{1}{n} \sum_{i=1}^n \nabla_{\mathbf{x}} f(\mathbf{x}, \mathbf{y}^*(\mathbf{x}); \mathbf{z}_i) - \frac{1}{n} \sum_{i=1}^n \nabla_{\mathbf{x}} f(\mathbf{x}, \mathbf{y}_S^*(\mathbf{x}); \mathbf{z}_i) \right\| \\
&\leq \left\| \mathbb{E}_{\mathbf{z}} \nabla_{\mathbf{x}} f(\mathbf{x}, \mathbf{y}^*(\mathbf{x}); \mathbf{z}) - \frac{1}{n} \sum_{i=1}^n \nabla_{\mathbf{x}} f(\mathbf{x}, \mathbf{y}^*(\mathbf{x}); \mathbf{z}_i) \right\| \\
&\qquad + \left\| \frac{1}{n} \sum_{i=1}^n \nabla_{\mathbf{x}} f(\mathbf{x}, \mathbf{y}^*(\mathbf{x}); \mathbf{z}_i) - \frac{1}{n} \sum_{i=1}^n \nabla_{\mathbf{x}} f(\mathbf{x}, \mathbf{y}_S^*(\mathbf{x}); \mathbf{z}_i) \right\| \\
&\leq \left\| \mathbb{E}_{\mathbf{z}} \nabla_{\mathbf{x}} f(\mathbf{x}, \mathbf{y}^*(\mathbf{x}); \mathbf{z}) - \frac{1}{n} \sum_{i=1}^n \nabla_{\mathbf{x}} f(\mathbf{x}, \mathbf{y}^*(\mathbf{x}); \mathbf{z}_i) \right\| + \beta \|\mathbf{y}^*(\mathbf{x}) - \mathbf{y}_S^*(\mathbf{x})\| \\
&= \|\nabla_{\mathbf{x}} F(\mathbf{x}, \mathbf{y}^*(\mathbf{x})) - \nabla_{\mathbf{x}} F_S(\mathbf{x}, \mathbf{y}^*(\mathbf{x}))\| + \beta \|\mathbf{y}^*(\mathbf{x}) - \mathbf{y}_S^*(\mathbf{x})\|,
\end{aligned}
\tag{22}
$$

where the first inequality satisfies from the triangle inequality, the second inequality holds by the smoothness of $f$. Next we need to upper bound these two terms respectively.

Firstly we need to upper bound $\|\nabla_{\mathbf{x}} F(\mathbf{x}, \mathbf{y}^*(\mathbf{x})) - \nabla_{\mathbf{x}} F_S(\mathbf{x}, \mathbf{y}^*(\mathbf{x}))\|$. According to Lemma 15, for all $\mathbf{x} \in \mathcal{X}$ and for any $\delta \in (0, 1)$, with probability at least $1 - \frac{\delta}{2}$, we have

$$
\|\nabla_{\mathbf{x}} F(\mathbf{x}, \mathbf{y}^*(\mathbf{x})) - \nabla_{\mathbf{x}} F_S(\mathbf{x}, \mathbf{y}^*(\mathbf{x}))\| \leq \sqrt{\frac{2\mathbb{E}[\|\nabla_{\mathbf{x}} f(\mathbf{x}^*, \mathbf{y}^*; \mathbf{z})\|^2] \log \frac{8}{\delta}}{n}} + \frac{B_{\mathbf{x}^*} \log \frac{8}{\delta}}{n}
$$
$$
+ \frac{C\beta(\mu_{\mathbf{y}} + \beta)}{\mu_{\mathbf{y}}} \max\left\{ \|\mathbf{x} - \mathbf{x}^*\|, \frac{1}{n} \right\} \times \left( \sqrt{\frac{d + \log \frac{16\log_2(\sqrt{2}R_1 n + 1)}{\delta}}{n}} + \frac{d + \log \frac{16\log_2(\sqrt{2}R_1 n + 1)}{\delta}}{n} \right).
\tag{23}
$$

Next we will bound $\|\mathbf{y}^*(\mathbf{x}) - \mathbf{y}_S^*(\mathbf{x})\|$. According to Lemma 11, with probability at least $1 - \frac{\delta}{2}$ we have the following inequalities

$$
\|\mathbf{y}^*(\mathbf{x}) - \mathbf{y}_S^*(\mathbf{x})\| \leq \frac{1}{\mu_{\mathbf{y}}} \left( \sqrt{\frac{2\mathbb{E}[\|\nabla_{\mathbf{y}} f(\mathbf{x}, \mathbf{y}^*(\mathbf{x}); \mathbf{z})\|^2] \log \frac{4}{\delta}}{n}} + \frac{B_{\mathbf{y}^*} \log \frac{4}{\delta}}{n} \right).
\tag{24}
$$

Finally, we plug (23) and (24) into (22), we obtain the result that for any $\delta \in (0, 1)$, with probability at least $1 - \delta$,

$$
\|\nabla\Phi(\mathbf{x}) - \nabla\Phi_S(\mathbf{x})\| \leq \frac{\beta}{\mu_{\mathbf{y}}} \left( \sqrt{\frac{2\mathbb{E}[\|\nabla_{\mathbf{y}} f(\mathbf{x}, \mathbf{y}^*(\mathbf{x}); \mathbf{z})\|^2] \log \frac{4}{\delta}}{n}} + \frac{B_{\mathbf{y}^*} \log \frac{4}{\delta}}{n} \right)
$$
$$
+ \sqrt{\frac{2\mathbb{E}[\|\nabla_{\mathbf{x}} f(\mathbf{x}^*, \mathbf{y}^*; \mathbf{z})\|^2] \log \frac{8}{\delta}}{n}} + \frac{B_{\mathbf{x}^*} \log \frac{8}{\delta}}{n} + \frac{C\beta(\mu_{\mathbf{y}} + \beta)}{\mu_{\mathbf{y}}} \max\left\{ \|\mathbf{x} - \mathbf{x}^*\|, \frac{1}{n} \right\} \times
$$
$$
\left( \sqrt{\frac{d + \log \frac{16\log_2(\sqrt{2}R_1 n + 1)}{\delta}}{n}} + \frac{d + \log \frac{16\log_2(\sqrt{2}R_1 n + 1)}{\delta}}{n} \right).
$$

The proof is complete.    □

*Proof of Theorem 3.* Firstly, for all $\mathbf{x} \in \mathcal{X}$, we divide $\|\nabla\Phi(\mathbf{x}) - \nabla\Phi_S(\mathbf{x})\|$ into two terms which is the same as (22)

$$\|\nabla\Phi(\mathbf{x}) - \nabla\Phi_S(\mathbf{x})\| \leq \|\nabla_\mathbf{x} F(\mathbf{x}, \mathbf{y}^*(\mathbf{x})) - \nabla_\mathbf{x} F_S(\mathbf{x}, \mathbf{y}^*(\mathbf{x}))\| + \beta\,\|\mathbf{y}^*(\mathbf{x}) - \mathbf{y}_S^*(\mathbf{x})\|. \qquad (25)$$

Firstly, we start to bound $\|\nabla_\mathbf{x} F(\mathbf{x}, \mathbf{y}^*(\mathbf{x})) - \nabla_\mathbf{x} F_S(\mathbf{x}, \mathbf{y}^*(\mathbf{x}))\|$. According to Lemma 15, applying norm triangle inequality, for any $\delta \in (0, 1)$, with probability at least $1 - \frac{\delta}{2}$, we have

$$\|\nabla_\mathbf{x} F(\mathbf{x}, \mathbf{y}^*(\mathbf{x}))\| - \|\nabla_\mathbf{x} F_S(\mathbf{x}, \mathbf{y}^*(\mathbf{x}))\| \leq \sqrt{\frac{2\mathbb{E}[\|\nabla_\mathbf{x} f(\mathbf{x}^*, \mathbf{y}^*; \mathbf{z})\|^2]\log\frac{8}{\delta}}{n}} + \frac{B_{\mathbf{x}^*}\log\frac{8}{\delta}}{n}$$

$$+ \frac{C\beta(\mu_\mathbf{y} + \beta)}{\mu_\mathbf{y}}\max\left\{\|\mathbf{x} - \mathbf{x}^*\|, \frac{1}{n}\right\} \times \left(\sqrt{\frac{d + \log\frac{16\log_2(\sqrt{2}R_1 n + 1)}{\delta}}{n}} + \frac{d + \log\frac{16\log_2(\sqrt{2}R_1 n + 1)}{\delta}}{n}\right).$$
$$(26)$$

Since the population risk $F(\mathbf{x}, \mathbf{y})$ satisfies Assumption 4 with parameter $\mu_\mathbf{x}$, according to Lemma 9, $\Phi(\mathbf{x})$ satisfies the PL condition with $\mu_\mathbf{x}$, there holds the error bound property (refer to Theorem 2 in (Karimi et al., 2016))

$$\mu_\mathbf{x}\|\mathbf{x} - \mathbf{x}^*\| \leq \|\Phi(\mathbf{x})\| = \|\nabla_\mathbf{x} F(\mathbf{x}, \mathbf{y}^*(\mathbf{x}))\|. \qquad (27)$$

Thus, combing (26) and (27) we have

$$\mu_\mathbf{x}\|\mathbf{x} - \mathbf{x}^*\| \leq \|\nabla_\mathbf{x} F(\mathbf{x}, \mathbf{y}^*(\mathbf{x}))\|$$

$$\leq \frac{C\beta(\mu_\mathbf{y} + \beta)}{\mu_\mathbf{y}}\max\left\{\|\mathbf{x} - \mathbf{x}^*\|, \frac{1}{n}\right\} \times \left(\sqrt{\frac{d + \log\frac{16\log_2(\sqrt{2}R_1 n + 1)}{\delta}}{n}} + \frac{d + \log\frac{16\log_2(\sqrt{2}R_1 n + 1)}{\delta}}{n}\right)$$

$$+ \|\nabla_\mathbf{x} F_S(\mathbf{x}, \mathbf{y}^*(\mathbf{x}))\| + \sqrt{\frac{2\mathbb{E}[\|\nabla_\mathbf{x} f(\mathbf{x}^*, \mathbf{y}^*; \mathbf{z})\|^2]\log\frac{8}{\delta}}{n}} + \frac{B_{\mathbf{x}^*}\log\frac{8}{\delta}}{n}.$$
$$(28)$$

On the other hand, according to Lemma 8 with Assumption 1, the function $\Phi(\mathbf{x})$ is $\beta + \frac{\beta^2}{\mu_\mathbf{y}}$-smooth in $\mathbf{x} \in \mathcal{X}$. According to (Nesterov, 2003), $\Phi(\mathbf{x})$ holds the following property

$$\frac{1}{2(\beta + \frac{\beta^2}{\mu_\mathbf{y}})}\|\nabla\Phi(\mathbf{x})\|^2 \leq \Phi(\mathbf{x}) - \Phi(\mathbf{x}^*).$$

We know that $\Phi(\mathbf{x})$ satisfies the PL condition with $\mu_\mathbf{x}$. Thus, we have

$$\frac{1}{2(\beta + \frac{\beta^2}{\mu_\mathbf{y}})}\|\nabla\Phi(\mathbf{x})\| \leq \Phi(\mathbf{x}) - \Phi(\mathbf{x}^*) \leq \frac{1}{2\mu_\mathbf{x}}\|\nabla\Phi(\mathbf{x})\|^2,$$

which means that $\frac{\mu_\mathbf{x}\mu_\mathbf{y}}{\beta(\mu_\mathbf{y} + \beta)} \leq 1$. Then let $c = \max\{16C^2, 1\}$, when

$$n \geq \frac{c\beta^2(\mu_\mathbf{y} + \beta)^2(d + \log\frac{8\log_2\sqrt{2}R_1 n + 1}{\delta})}{\mu_\mathbf{y}^2\mu_\mathbf{x}^2},$$

we have

$$\frac{C\beta(\mu_\mathbf{y} + \beta)}{\mu_\mathbf{y}}\left(\sqrt{\frac{d + \log\frac{16\log_2(\sqrt{2}R_1 n + 1)}{\delta}}{n}} + \frac{d + \log\frac{16\log_2(\sqrt{2}R_1 n + 1)}{\delta}}{n}\right) \leq \frac{\mu_\mathbf{x}}{2}, \qquad (29)$$

with the fact that $\frac{\mu_\mathbf{x}\mu_\mathbf{y}}{\beta(\mu_\mathbf{y} + \beta)} \leq 1$.

Next, plugging (29) into (28), we obtain that

$$\|\mathbf{x} - \mathbf{x}^*\| \leq \frac{2}{\mu_\mathbf{x}}\left(\|\nabla_\mathbf{x} F_S(\mathbf{x}, \mathbf{y}^*(\mathbf{x}))\| + \sqrt{\frac{2\mathbb{E}[\|\nabla_\mathbf{x} f(\mathbf{x}^*, \mathbf{y}^*; \mathbf{z})\|^2]\log\frac{8}{\delta}}{n}} + \frac{B_{\mathbf{x}^*}\log\frac{8}{\delta}}{n} + \frac{\mu_\mathbf{x}}{2n}\right).$$

Then plugging (30) into (26), we derive that for all $\mathbf{x} \in \mathcal{X}$, when $n \geq \frac{c\beta^2(\mu_{\mathbf{y}}+\beta)^2(d+\log\frac{8\log_2\sqrt{2}R_1 n+1}{\delta})}{\mu_{\mathbf{y}}^2\mu_{\mathbf{x}}^2}$, with probability at least $1 - \frac{\delta}{2}$,

$$
\begin{aligned}
\|\nabla_{\mathbf{x}}F(\mathbf{x}, \mathbf{y}^*(\mathbf{x})) - \nabla_{\mathbf{x}}F_S(\mathbf{x}, \mathbf{y}^*(\mathbf{x}))\| &\leq \|\nabla_{\mathbf{x}}F_S(\mathbf{x}, \mathbf{y}^*(\mathbf{x}))\| \\
&+ 2\sqrt{\frac{2\mathbb{E}[\|\nabla_{\mathbf{x}}f(\mathbf{x}^*, \mathbf{y}^*; \mathbf{z})\|^2]\log\frac{8}{\delta}}{n}} + \frac{2B_{\mathbf{x}^*}\log\frac{8}{\delta}}{n} + \frac{\mu_{\mathbf{x}}}{n}.
\end{aligned}
\tag{30}
$$

Next, we need to bound $\|\nabla_{\mathbf{x}}F_S(\mathbf{x}, \mathbf{y}^*(\mathbf{x}))\|$. We make the following decomposition

$$
\begin{aligned}
&\|\nabla_{\mathbf{x}}F_S(\mathbf{x}, \mathbf{y}^*(\mathbf{x}))\| \\
=&\|\nabla_{\mathbf{x}}F_S(\mathbf{x}, \mathbf{y}^*(\mathbf{x})) - \nabla_{\mathbf{x}}F_S(\mathbf{x}, \mathbf{y}_S^*(\mathbf{x})) + \nabla_{\mathbf{x}}F_S(\mathbf{x}, \mathbf{y}_S^*(\mathbf{x}))\| \\
\leq&\|\nabla_{\mathbf{x}}F_S(\mathbf{x}, \mathbf{y}^*(\mathbf{x})) - \nabla_{\mathbf{x}}F_S(\mathbf{x}, \mathbf{y}_S^*(\mathbf{x}))\| + \|\nabla_{\mathbf{x}}F_S(\mathbf{x}, \mathbf{y}_S^*(\mathbf{x}))\| \\
\leq&\beta\|\mathbf{y}^*(\mathbf{x}) - \mathbf{y}_S^*(\mathbf{x})\| + \|\nabla\Phi_S(\mathbf{x})\|,
\end{aligned}
\tag{31}
$$

where the first inequality holds with the norm triangle inequality and the second inequality holds with the fact that $f$ is $\beta$-smooth.

According to Lemma 11, for any $\mathbf{x} \in \mathcal{X}$, With probability at least $1 - \frac{\delta}{2}$ we have the following inequalities

$$
\|\mathbf{y}^*(\mathbf{x}) - \mathbf{y}_S^*(\mathbf{x})\| \leq \frac{1}{\mu_{\mathbf{y}}}\left(\sqrt{\frac{2\mathbb{E}[\|\nabla_{\mathbf{y}}f(\mathbf{x}, \mathbf{y}^*(\mathbf{x}); \mathbf{z})\|^2]\log\frac{4}{\delta}}{n}} + \frac{B_{\mathbf{y}^*}\log\frac{4}{\delta}}{n}\right).
\tag{32}
$$

Combing (30), (31), (25) and (32), we have that for all $\mathbf{x} \in \mathcal{X}$, let $c = \max\{16C^2, 1\}$, when $n \geq \frac{c\beta^2(\mu_{\mathbf{y}}+\beta)^2(d+\log\frac{8\log_2\sqrt{2}R_1 n+1}{\delta})}{\mu_{\mathbf{y}}^2\mu_{\mathbf{x}}^2}$, with probability at least $1 - \delta$

$$
\begin{aligned}
\|\nabla\Phi(\mathbf{x}) - \nabla\Phi_S(\mathbf{x})\| \leq& \|\nabla\Phi_S(\mathbf{x})\| + 2\sqrt{\frac{2\mathbb{E}[\|\nabla_{\mathbf{x}}f(\mathbf{x}^*, \mathbf{y}^*(\mathbf{x}^*); \mathbf{z})\|^2]\log\frac{8}{\delta}}{n}} \\
&+ \frac{2B_{\mathbf{x}^*}\log\frac{8}{\delta}}{n} + \frac{\mu_{\mathbf{x}}}{n} + \frac{2\beta}{\mu_{\mathbf{y}}}\left(\sqrt{\frac{2\mathbb{E}[\|\nabla_{\mathbf{y}}f(\mathbf{x}, \mathbf{y}^*(\mathbf{x}); \mathbf{z})\|^2]\log\frac{4}{\delta}}{n}} + \frac{B_{\mathbf{y}^*}\log\frac{4}{\delta}}{n}\right).
\end{aligned}
$$

The proof is complete. $\qquad\square$

*Proof of Remark 5.* Here we briefly prove the results given in Remark 5. Since $\Phi(\mathbf{x})$ satisfies the PL condition with $\mu_{\mathbf{x}}$, we have we have

$$
\Phi(\mathbf{x}) - \Phi(\mathbf{x}^*) \leq \frac{\|\nabla\Phi(\mathbf{x})\|^2}{2\mu_{\mathbf{x}}}.
\tag{33}
$$

Therefore, we need to bound $\|\nabla\Phi(\mathbf{x})\|^2$. According to Theorem 3, for any $\delta \in (0, 1)$, when $n \geq \frac{c\beta^2(\mu_{\mathbf{y}}+\beta)^2(d+\log\frac{8\log_2\sqrt{2}R_1 n+1}{\delta})}{\mu_{\mathbf{y}}^2\mu_{\mathbf{x}}^2}$, with probability at least $1 - \delta$,

$$
\begin{aligned}
\|\nabla\Phi(\mathbf{x})\| \leq& 2\|\nabla\Phi_S(\mathbf{x})\| + 2\sqrt{\frac{2\mathbb{E}[\|\nabla_{\mathbf{x}}f(\mathbf{x}^*, \mathbf{y}^*; \mathbf{z})\|^2]\log\frac{8}{\delta}}{n}} \\
&+ \frac{2B_{\mathbf{x}^*}\log\frac{8}{\delta}}{n} + \frac{\mu_{\mathbf{x}}}{n} + \frac{2\beta}{\mu_{\mathbf{y}}}\left(\sqrt{\frac{2\mathbb{E}[\|\nabla_{\mathbf{y}}f(\mathbf{x}, \mathbf{y}^*(\mathbf{x}); \mathbf{z})\|^2]\log\frac{4}{\delta}}{n}} + \frac{B_{\mathbf{y}^*}\log\frac{4}{\delta}}{n}\right).
\end{aligned}
\tag{34}
$$

Then, substituting (34) into (33), we have

$$
\begin{aligned}
&\Phi(\mathbf{x}) - \Phi(\mathbf{x}^*) \\
&\leq \frac{\|\nabla\Phi(\mathbf{x})\|^2}{2\mu_{\mathbf{x}}} \\
&\leq \frac{1}{2\mu_{\mathbf{x}}} \Bigg\{ 2\|\nabla\Phi_S(\mathbf{x})\| + \frac{2\beta}{\mu_{\mathbf{y}}} \left( \sqrt{\frac{2\mathbb{E}[\|\nabla_{\mathbf{y}}f(\mathbf{x},\mathbf{y}^*(\mathbf{x});\mathbf{z})\|^2]\log\frac{4}{\delta}}{n}} + \frac{B_{\mathbf{y}^*}\log\frac{4}{\delta}}{n} \right) \\
&\qquad\qquad + 2\sqrt{\frac{2\mathbb{E}[\|\nabla_{\mathbf{x}}f(\mathbf{x}^*,\mathbf{y}^*;\mathbf{z})\|^2]\log\frac{8}{\delta}}{n}} + \frac{2B_{\mathbf{x}^*}\log\frac{8}{\delta}}{n} + \frac{\mu_{\mathbf{x}}}{n} \Bigg\}^2 \\
&\leq \frac{8\|\nabla\Phi_S(\mathbf{x})\|^2}{\mu_{\mathbf{x}}} + \frac{16\beta^2\mathbb{E}[\|\nabla_{\mathbf{y}}f(\mathbf{x},\mathbf{y}^*(\mathbf{x});\mathbf{z})\|^2]\log\frac{4}{\delta}}{\mu_{\mathbf{x}}\mu_{\mathbf{y}}^2 n} \\
&\qquad + \frac{16\mathbb{E}[\|\nabla_{\mathbf{x}}f(\mathbf{x}^*,\mathbf{y}^*;\mathbf{z})\|^2]\log\frac{8}{\delta}}{\mu_{\mathbf{x}} n} + \frac{2\left(\frac{2\beta B_{\mathbf{y}^*}}{\mu_{\mathbf{y}}}\log\frac{4}{\delta} + 2B_{\mathbf{x}^*}\log\frac{8}{\delta} + \mu_{\mathbf{x}}\right)^2}{\mu_{\mathbf{x}} n^2},
\end{aligned}
$$

where the second inequality holds with Cauchy–Bunyakovsky–Schwarz inequality.

The proof is complete. $\qquad\square$

## D APPLICATION

### D.1 EMPIRICAL SADDLE POINT

*Proof of Theorem 4.* Plugging $\hat{\mathbf{x}}^*$ into Theorem 1, for any $\delta \in (0,1)$, with probability at least $1 - \delta$, we have

$$
\begin{aligned}
\|\nabla\Phi(\hat{\mathbf{x}}^*)\| - \|\nabla\Phi_S(\hat{\mathbf{x}}^*)\| &\leq \frac{\beta}{\mu_{\mathbf{y}}}\left( \sqrt{\frac{2\mathbb{E}[\|\nabla_{\mathbf{y}}f(\hat{\mathbf{x}}^*,\mathbf{y}^*(\hat{\mathbf{x}}^*);\mathbf{z})\|^2]\log\frac{4}{\delta}}{n}} + \frac{B_{\mathbf{y}^*}\log\frac{4}{\delta}}{n} \right) \\
&+ \sqrt{\frac{2\mathbb{E}[\|\nabla_{\mathbf{x}}f(\mathbf{x}^*,\mathbf{y}^*;\mathbf{z})\|^2]\log\frac{8}{\delta}}{n}} + \frac{B_{\mathbf{x}^*}\log\frac{8}{\delta}}{n} + \frac{C\beta(\mu_{\mathbf{y}}+\beta)}{\mu_{\mathbf{y}}}\max\left\{\|\mathbf{x}-\mathbf{x}^*\|,\frac{1}{n}\right\} \\
&\times \left( \sqrt{\frac{d + \log\frac{16\log_2(\sqrt{2}R_1 n + 1)}{\delta}}{n}} + \frac{d + \log\frac{16\log_2(\sqrt{2}R_1 n + 1)}{\delta}}{n} \right).
\end{aligned}
$$

Since $\hat{\mathbf{x}}^*$ is the solution of (2), there holds that $\|\nabla\Phi_S(\hat{\mathbf{x}}^*)\| = 0$. Thus, we can derive that

$$
\begin{aligned}
\|\nabla\Phi(\hat{\mathbf{x}}^*)\| &\leq \frac{\beta}{\mu_{\mathbf{y}}}\left( \sqrt{\frac{2\mathbb{E}[\|\nabla_{\mathbf{y}}f(\hat{\mathbf{x}}^*,\mathbf{y}^*(\hat{\mathbf{x}}^*);\mathbf{z})\|^2]\log\frac{4}{\delta}}{n}} + \frac{B_{\mathbf{y}^*}\log\frac{4}{\delta}}{n} \right) \\
&+ \sqrt{\frac{2\mathbb{E}[\|\nabla_{\mathbf{x}}f(\mathbf{x}^*,\mathbf{y}^*;\mathbf{z})\|^2]\log\frac{8}{\delta}}{n}} + \frac{B_{\mathbf{x}^*}\log\frac{8}{\delta}}{n} + \frac{C\beta(\mu_{\mathbf{y}}+\beta)}{\mu_{\mathbf{y}}}\left(R_1 + \frac{1}{n}\right) \\
&\times \left( \sqrt{\frac{d + \log\frac{16\log_2(\sqrt{2}R_1 n + 1)}{\delta}}{n}} + \frac{d + \log\frac{16\log_2(\sqrt{2}R_1 n + 1)}{\delta}}{n} \right) \\
&= O\left( \sqrt{\frac{d + \log\frac{\log n}{\delta}}{n}} \right).
\end{aligned}
$$

The proof is complete. $\qquad\square$

*Proof of Theorem 5.* According to Lemma 9, $\Phi(\mathbf{x})$ satisfies the PL condition with $\mu_{\mathbf{x}}$, we have

$$\Phi(\hat{\mathbf{x}}^*) - \Phi(\mathbf{x}^*) \leq \frac{\|\nabla\Phi(\hat{\mathbf{x}}^*)\|^2}{2\mu_{\mathbf{x}}}. \tag{35}$$

Therefore, we need to bound $\|\nabla\Phi(\hat{\mathbf{x}}^*)\|^2$. Plugging $\hat{\mathbf{x}}^*$ into Theorem 3, for any $\delta \in (0,1)$, when $n \geq \frac{c\beta^2(\mu_{\mathbf{y}}+\beta)^2(d+\log\frac{8\log_2\sqrt{2}R_1n+1}{\delta})}{\mu_{\mathbf{y}}^2\mu_{\mathbf{x}}^2}$, with probability at least $1-\delta$

$$\|\nabla\Phi(\hat{\mathbf{x}}^*)\| \leq 2\|\nabla\Phi_S(\hat{\mathbf{x}}^*)\| + 2\sqrt{\frac{2\mathbb{E}[\|\nabla_{\mathbf{x}}f(\mathbf{x}^*,\mathbf{y}^*;\mathbf{z})\|^2]\log\frac{8}{\delta}}{n}}$$

$$+ \frac{2B_{\mathbf{x}^*}\log\frac{8}{\delta}}{n} + \frac{\mu_{\mathbf{x}}}{n} + \frac{2\beta}{\mu_{\mathbf{y}}}\left(\sqrt{\frac{2\mathbb{E}[\|\nabla_{\mathbf{y}}f(\hat{\mathbf{x}}^*,\mathbf{y}^*(\hat{\mathbf{x}}^*);\mathbf{z})\|^2]\log\frac{4}{\delta}}{n}} + \frac{B_{\mathbf{y}^*}\log\frac{4}{\delta}}{n}\right). \tag{36}$$

Since $\nabla\Phi_S(\hat{\mathbf{x}}^*) = \nabla_{\mathbf{x}}F_S(\hat{\mathbf{x}}^*,\hat{\mathbf{y}}^*) = \mathbf{0}$, we have $\|\nabla\Phi_S(\hat{\mathbf{x}}^*)\| = \|\nabla_{\mathbf{x}}F_S(\hat{\mathbf{x}}^*,\hat{\mathbf{y}}^*)\| = 0$. By plugging (36) into (35), we have

$$\Phi(\hat{\mathbf{x}}^*) - \Phi(\mathbf{x}^*)$$

$$\leq \frac{\|\nabla\Phi(\hat{\mathbf{x}}^*)\|^2}{2\mu_{\mathbf{x}}}$$

$$\leq \frac{1}{2\mu_{\mathbf{x}}}\left\{\frac{2\beta}{\mu_{\mathbf{y}}}\left(\sqrt{\frac{2\mathbb{E}[\|\nabla_{\mathbf{y}}f(\hat{\mathbf{x}}^*,\mathbf{y}^*(\hat{\mathbf{x}}^*);\mathbf{z})\|^2]\log\frac{4}{\delta}}{n}} + \frac{B_{\mathbf{y}^*}\log\frac{4}{\delta}}{n}\right)\right.$$

$$\left. + 2\sqrt{\frac{2\mathbb{E}[\|\nabla_{\mathbf{x}}f(\mathbf{x}^*,\mathbf{y}^*;\mathbf{z})\|^2]\log\frac{8}{\delta}}{n}} + \frac{2B_{\mathbf{x}^*}\log\frac{8}{\delta}}{n} + \frac{\mu_{\mathbf{x}}}{n}\right\}^2$$

$$\leq \frac{12\beta^2\mathbb{E}[\|\nabla_{\mathbf{y}}f(\hat{\mathbf{x}}^*,\mathbf{y}^*(\hat{\mathbf{x}}^*);\mathbf{z})\|^2]\log\frac{4}{\delta}}{\mu_{\mathbf{x}}\mu_{\mathbf{y}}^2n} + \frac{12\mathbb{E}[\|\nabla_{\mathbf{x}}f(\mathbf{x}^*,\mathbf{y}^*;\mathbf{z})\|^2]\log\frac{8}{\delta}}{\mu_{\mathbf{x}}n}$$

$$+ \frac{3\left(\frac{2\beta B_{\mathbf{y}^*}}{\mu_{\mathbf{y}}}\log\frac{4}{\delta} + 2B_{\mathbf{x}^*}\log\frac{8}{\delta} + \mu_{\mathbf{x}}\right)^2}{2\mu_{\mathbf{x}}n^2},$$

where the second inequality holds with Cauchy–Bunyakovsky–Schwarz inequality.

Next, if we further assume $\Phi(\mathbf{x}^*) = O\left(\frac{1}{n}\right)$, according to Lemma 7, we have $\mathbb{E}\|\nabla_{\mathbf{x}}f(\mathbf{x}^*,\mathbf{y}^*;\mathbf{z})\|^2 \leq 4\beta\mathbb{E}[f(\mathbf{x}^*,\mathbf{y}^*;\mathbf{z})]$ and $\mathbb{E}\|\nabla_{\mathbf{y}}f(\hat{\mathbf{x}}^*,\mathbf{y}^*(\hat{\mathbf{x}}^*);\mathbf{z})\|^2 \leq 4\beta\mathbb{E}[f(\hat{\mathbf{x}}^*,\mathbf{y}^*(\hat{\mathbf{x}}^*);\mathbf{z})]$. Then we have

$$\Phi(\hat{\mathbf{x}}^*) - \Phi(\mathbf{x}^*) \leq \frac{48\beta^3\mathbb{E}[f(\hat{\mathbf{x}}^*,\mathbf{y}^*(\hat{\mathbf{x}}^*);\mathbf{z})]\log\frac{4}{\delta}}{\mu_{\mathbf{x}}\mu_{\mathbf{y}}^2n} + \frac{48\beta\mathbb{E}[f(\mathbf{x}^*,\mathbf{y}^*;\mathbf{z})]\log\frac{8}{\delta}}{\mu_{\mathbf{x}}n}$$

$$+ \frac{3\left(\frac{2\beta B_{\mathbf{y}^*}}{\mu_{\mathbf{y}}}\log\frac{4}{\delta} + 2B_{\mathbf{x}^*}\log\frac{8}{\delta} + \mu_{\mathbf{x}}\right)^2}{2\mu_{\mathbf{x}}n^2}$$

$$= \frac{48\beta^3\Phi(\hat{\mathbf{x}}^*)\log\frac{4}{\delta}}{\mu_{\mathbf{x}}\mu_{\mathbf{y}}^2n} + \frac{48\beta\Phi(\mathbf{x}^*)\log\frac{8}{\delta}}{\mu_{\mathbf{x}}n} + \frac{3\left(\frac{2\beta B_{\mathbf{y}^*}}{\mu_{\mathbf{y}}}\log\frac{4}{\delta} + 2B_{\mathbf{x}^*}\log\frac{8}{\delta} + \mu_{\mathbf{x}}\right)^2}{2\mu_{\mathbf{x}}n^2},$$

which implies that

$$\left(1 - \frac{48\beta^3\log\frac{4}{\delta}}{\mu_{\mathbf{x}}\mu_{\mathbf{y}}^2n}\right)(\Phi(\hat{\mathbf{x}}^*) - \Phi(\mathbf{x}^*))$$

$$\leq \frac{48\beta\Phi(\mathbf{x}^*)\log\frac{8}{\delta}}{\mu_{\mathbf{x}}n} + \frac{48\beta^3\Phi(\mathbf{x}^*)\log\frac{4}{\delta}}{\mu_{\mathbf{x}}\mu_{\mathbf{y}}^2n} + \frac{3\left(\frac{2\beta B_{\mathbf{y}^*}}{\mu_{\mathbf{y}}}\log\frac{4}{\delta} + 2B_{\mathbf{x}^*}\log\frac{8}{\delta} + \mu_{\mathbf{x}}\right)^2}{2\mu_{\mathbf{x}}n^2}.$$

When $n \geq \max\left\{\frac{c\beta^2(\mu_\mathbf{y}+\beta)^2(d+\log\frac{8\log_2\sqrt{2}R_1n+1}{\delta})}{\mu_\mathbf{y}^2\mu_\mathbf{x}^2}, \frac{48\beta^3\log\frac{4}{\delta}}{\mu_\mathbf{x}\mu_\mathbf{y}^2}\right\}$, we have

$$\Phi(\hat{\mathbf{x}}^*) - \Phi(\mathbf{x}^*) = O\left(\frac{\Phi(\mathbf{x}^*)\log\frac{1}{\delta}}{n - \frac{48\beta^3\log\frac{4}{\delta}}{\mu_\mathbf{x}\mu_\mathbf{y}^2}} + \frac{\log^2\frac{1}{\delta}}{n\left(n - \frac{48\beta^3\log\frac{4}{\delta}}{\mu_\mathbf{x}\mu_\mathbf{y}^2}\right)}\right).$$

The proof is complete. $\qquad\square$

### D.2 GRADIENT DESCENT ASCENT

Firstly, we introduce the optimization error bound for GDA given in (Lin et al., 2020).

**Lemma 16** (Optimization error bound for NC-SC minimax problems (Lin et al., 2020))**.** *under Assumption 1, and letting the step sizes be chosen as* $\eta_\mathbf{x} = \frac{1}{16(\frac{\beta}{\mu}+1)^2\beta}$ *and* $\eta_\mathbf{y} = \frac{1}{\beta}$*, then the optimization error bound of Algorithm 1 can be bounded by*

$$\frac{1}{T}\sum_{t=1}^{T}\|\nabla\Phi_S(\mathbf{x}_t)\|^2 \leq \frac{128\beta^3\Delta_\Phi}{\mu_\mathbf{y}^2T} + \frac{5\beta^3D_\mathcal{Y}}{\mu_\mathbf{y}T},$$

*where* $\Delta_\Phi = \Phi_S(\mathbf{x}_0) - \min_\mathbf{x}\Phi_S(\mathbf{x})$*.*

*Proof of Theorem 6.* Firstly, we have

$$
\begin{aligned}
\frac{1}{T}\sum_{t=1}^{T}\|\nabla\Phi(\mathbf{x}_t)\|^2 &\leq \frac{2}{T}\sum_{t=1}^{T}\|\nabla\Phi(\mathbf{x}_t) - \nabla_\mathbf{x}\Phi_S(\mathbf{x}_t)\|^2 + \frac{2}{T}\sum_{t=1}^{T}\|\nabla_\mathbf{x}\Phi_S(\mathbf{x}_t)\|^2 \\
&\leq \frac{2}{T}\sum_{t=1}^{T}\max_{t=1,\cdots,T}\|\nabla\Phi(\mathbf{x}_t) - \nabla_\mathbf{x}\Phi_S(\mathbf{x}_t)\|^2 + \frac{2}{T}\sum_{t=1}^{T}\|\nabla_\mathbf{x}\Phi_S(\mathbf{x}_t)\|^2 \qquad (37) \\
&\leq \frac{2}{T}\sum_{t=1}^{T}\max_{t=1,\cdots,T}\|\nabla\Phi(\mathbf{x}_t) - \nabla_\mathbf{x}\Phi_S(\mathbf{x}_t)\|^2 + \frac{256\beta^3\Delta_\Phi}{\mu_\mathbf{y}^2T} + \frac{10\beta^3D_\mathcal{Y}}{\mu_\mathbf{y}T},
\end{aligned}
$$

Thus, with probability at least $1 - \delta$, we have

$$
\begin{aligned}
\frac{1}{T}\sum_{t=1}^{T}\|\nabla\Phi(\mathbf{x}_t)\|^2 \leq {} & O\left(\frac{1}{T}\right) + O\left(\max_{t=1,\cdots,T}\left[\frac{C\beta(\mu_\mathbf{y}+\beta)}{\mu_\mathbf{y}}\max\left\{\|\mathbf{x}_t-\mathbf{x}^*\|, \frac{1}{n}\right\}\right.\right. \\
& \left.\left.\times\left(\sqrt{\frac{d+\log\frac{16\log_2(\sqrt{2}R_1n+1)}{\delta}}{n}} + \frac{d+\log\frac{16\log_2(\sqrt{2}R_1n+1)}{\delta}}{n}\right)\right]^2\right).
\end{aligned}
\qquad (38)
$$

where the inequality holds according to Theorem 1.

Next, we need to bound $\|\mathbf{x}_t-\mathbf{x}^*\|$. Since we assume that $\mathbf{x}_1 = 0$, and $\mathbf{x}_{t+1} = \mathbf{x}_t - \eta_\mathbf{x}\nabla_\mathbf{x}F_S(\mathbf{x}_t, \mathbf{y}_t)$, we have $\mathbf{x}_{t+1} = -\eta_\mathbf{x}\sum_{k=1}^{t}\nabla_\mathbf{x}F_S(\mathbf{x}_t, \mathbf{y}_t)$. then we have

$$
\begin{aligned}
\|\mathbf{x}_{t+1} - \mathbf{x}^*\| &\leq \|\mathbf{x}_{t+1}\| + \|\mathbf{x}^*\| \\
&\leq \left\|\eta_\mathbf{x}\sum_{k=1}^{t}\nabla_\mathbf{x}F_S(\mathbf{x}_t, \mathbf{y}_t)\right\| + \|\mathbf{x}^*\| = O(\eta_\mathbf{x}Lt).
\end{aligned}
\qquad (39)
$$

Then plugging (39) into (38), with probability at least $1 - \delta$

$$\frac{1}{T}\sum_{t=1}^{T}\|\nabla\Phi(\mathbf{x}_t)\|^2 \leq O\left(\frac{1}{T}\right) + O\left(\frac{d+\log\frac{16\log_2(\sqrt{2}R_1n+1)}{\delta}}{n}T\right).$$

Let $T \asymp O\left(\sqrt{\frac{n}{d}}\right)$, we can derive that

$$\frac{1}{T}\sum_{t=1}^{T}\|\nabla\Phi(\mathbf{x}_t)\|^2 \leq O\left(\frac{d^{\frac{1}{2}}+d^{-\frac{1}{2}}\log\frac{\log n}{\delta}}{n^{\frac{1}{2}}}\right).$$

The proof is complete. $\qquad\square$

*Proof of Theorem 7.* According to Lemma 9, $\Phi(\mathbf{x})$ satisfies the PL assumption with parameter $\mu_{\mathbf{x}}$, we have

$$\Phi(\mathbf{x}) - \Phi(\mathbf{x}^*) \leq \frac{\|\nabla\Phi(\mathbf{x})\|^2}{2\mu_{\mathbf{x}}}. \tag{40}$$

To bound $\Phi(\bar{\mathbf{x}}_T) - \Phi(\mathbf{x}^*)$, we firstly need to bound the term $\|\nabla\Phi(\bar{\mathbf{x}}_T)\|^2$. There holds that

$$\|\nabla\Phi(\bar{\mathbf{x}}_T)\|^2 \leq 2\|\nabla\Phi(\bar{\mathbf{x}}_T) - \nabla\Phi_S(\bar{\mathbf{x}}_T)\|^2 + 2\|\nabla\Phi_S(\bar{\mathbf{x}}_T)\|^2. \tag{41}$$

From Theorem 3, under Assumption 3 and 5, plugging $\bar{\mathbf{x}}_T$ into Theorem 3, for any $\delta \in (0, 1)$, when $n \geq \frac{c\beta^2(\mu_{\mathbf{y}}+\beta)^2(d+\log\frac{8\log_2\sqrt{2}R_1n+1}{\delta})}{\mu_{\mathbf{y}}^2\mu_{\mathbf{x}}^2}$, with probability at least $1 - \delta$

$$
\begin{aligned}
\|\nabla\Phi(\bar{\mathbf{x}}_T) - \nabla\Phi_S(\bar{\mathbf{x}}_T)\| &\leq \|\nabla\Phi_S(\bar{\mathbf{x}}_T)\| + 2\sqrt{\frac{2\mathbb{E}[\|\nabla_{\mathbf{x}}f(\mathbf{x}^*, \mathbf{y}^*(\mathbf{x}^*); \mathbf{z})\|^2]\log\frac{8}{\delta}}{n}} \\
&+ \frac{2B_{\mathbf{x}^*}\log\frac{8}{\delta}}{n} + \frac{\mu_{\mathbf{x}}}{n} + \frac{2\beta}{\mu_{\mathbf{y}}}\left(\sqrt{\frac{2\mathbb{E}[\|\nabla_{\mathbf{y}}f(\bar{\mathbf{x}}_T, \mathbf{y}^*(\bar{\mathbf{x}}_T); \mathbf{z})\|^2]\log\frac{4}{\delta}}{n}} + \frac{B_{\mathbf{y}^*}\log\frac{4}{\delta}}{n}\right).
\end{aligned}
\tag{42}
$$

Next, we need to bound the optimization error $\|\nabla\Phi_S(\bar{\mathbf{x}}_T)\|$. According to Lemma 9, $\Phi_S(\mathbf{x})$ satisfies the PL assumption with parameter $\mu_{\mathbf{x}}$, then using Lemma 16 we have

$$\frac{1}{T}\sum_{t=1}^T \Phi_S(\mathbf{x}_t) - \Phi_S(\mathbf{x}^*) \leq \frac{1}{2\mu_{\mathbf{x}}T}\sum_{t=1}^T \|\nabla\Phi_S(\mathbf{x}_t)\|^2 \leq \frac{64\beta^3\Delta_\Phi}{\mu_{\mathbf{x}}\mu_{\mathbf{y}}^2T} + \frac{5\beta^3D_{\mathcal{Y}}}{2\mu_{\mathbf{x}}\mu_{\mathbf{y}}T}.$$

From the convexity of $F_S(\cdot, \mathbf{y})$, we get

$$\Phi_S(\bar{\mathbf{x}}_T) - \Phi_S(\mathbf{x}^*) \leq \frac{1}{T}\sum_{t=1}^T \Phi_S(\mathbf{x}_t) - \Phi_S(\mathbf{x}^*) \leq \frac{64\beta^3\Delta_\Phi}{\mu_{\mathbf{x}}\mu_{\mathbf{y}}^2T} + \frac{5\beta^3D_{\mathcal{Y}}}{2\mu_{\mathbf{x}}\mu_{\mathbf{y}}T}.$$

According to (Nesterov, 2003) and Lemma 8, there holds the following property for $\beta + \frac{\beta^2}{\mu_{\mathbf{y}}}$ function $\Phi_S(\mathbf{x})$, we have

$$\frac{1}{2\left(\beta+\frac{\beta^2}{\mu_{\mathbf{y}}}\right)}\|\nabla\Phi_S(\bar{\mathbf{x}}_T)\|^2 \leq \Phi_S(\bar{\mathbf{x}}_T) - \Phi_S(\mathbf{x}^*) \leq \frac{64\beta^3\Delta_\Phi}{\mu_{\mathbf{x}}\mu_{\mathbf{y}}^2T} + \frac{5\beta^3D_{\mathcal{Y}}}{2\mu_{\mathbf{x}}\mu_{\mathbf{y}}T}. \tag{43}$$

Plugging (43) into (42), according to Cauchy–Bunyakovsky–Schwarz inequality, we can derive that

$$
\begin{aligned}
\|\nabla\Phi(\bar{\mathbf{x}}_T) - \nabla\Phi_S(\bar{\mathbf{x}}_T)\|^2 &\leq O\left(\frac{1}{T}\right) + \frac{32\beta^2\mathbb{E}[\|\nabla_{\mathbf{y}}f(\bar{\mathbf{x}}_T, \mathbf{y}^*(\bar{\mathbf{x}}_T); \mathbf{z})\|^2]\log\frac{4}{\delta}}{\mu_{\mathbf{y}}^2n} \\
&+ \frac{32\mathbb{E}[\|\nabla_{\mathbf{x}}f(\mathbf{x}^*, \mathbf{y}^*; \mathbf{z})\|^2]\log\frac{8}{\delta}}{n} + \frac{4\left(\frac{2\beta B_{\mathbf{y}^*}}{\mu_{\mathbf{y}}}\log\frac{4}{\delta} + 2B_{\mathbf{x}^*}\log\frac{8}{\delta} + \mu_{\mathbf{x}}\right)^2}{n^2}.
\end{aligned}
\tag{44}
$$

Then substituting (44), (43) into (41), we derive that

$$
\begin{aligned}
\|\nabla\Phi(\bar{\mathbf{x}}_T)\|^2 &\leq O\left(\frac{1}{T}\right) + \frac{64\beta^2\mathbb{E}[\|\nabla_{\mathbf{y}}f(\bar{\mathbf{x}}_T, \mathbf{y}^*(\bar{\mathbf{x}}_T); \mathbf{z})\|^2]\log\frac{4}{\delta}}{\mu_{\mathbf{y}}^2n} \\
&+ \frac{64\mathbb{E}[\|\nabla_{\mathbf{x}}f(\mathbf{x}^*, \mathbf{y}^*; \mathbf{z})\|^2]\log\frac{8}{\delta}}{n} + \frac{8\left(\frac{2\beta B_{\mathbf{y}^*}}{\mu_{\mathbf{y}}}\log\frac{4}{\delta} + 2B_{\mathbf{x}^*}\log\frac{8}{\delta} + \mu_{\mathbf{x}}\right)^2}{n^2}.
\end{aligned}
\tag{45}
$$

Finally, we plug (45) into (40) and choose $T \asymp O(n)$, with probability at least $1 - \delta$

$$\Phi(\bar{\mathbf{x}}_T) - \Phi(\mathbf{x}^*) = O\left(\frac{\mathbb{E}[\|\nabla_{\mathbf{x}}f(\mathbf{x}^*, \mathbf{y}^*; \mathbf{z})\|^2]\log\frac{1}{\delta}}{n} + \frac{\mathbb{E}[\|\nabla_{\mathbf{y}}f(\bar{\mathbf{x}}_T, \mathbf{y}^*(\bar{\mathbf{x}}_T); \mathbf{z})\|^2]\log\frac{1}{\delta}}{n} + \frac{\log^2\frac{1}{\delta}}{n^2}\right).$$

Next, if we further assume $\Phi(\mathbf{x}^*) = O\left(\frac{1}{n}\right)$, According to Lemma 7, we have $\mathbb{E}\|\nabla_{\mathbf{x}} f(\mathbf{x}^*, \mathbf{y}^*; \mathbf{z})\|^2 \leq 4\beta\mathbb{E}[f(\mathbf{x}^*, \mathbf{y}^*; \mathbf{z})]$ and $\mathbb{E}\|\nabla_{\mathbf{y}} f(\bar{\mathbf{x}}_T, \mathbf{y}^*(\bar{\mathbf{x}}_T); \mathbf{z})\|^2 \leq 4\beta\mathbb{E}[f(\bar{\mathbf{x}}_T, \mathbf{y}^*(\bar{\mathbf{x}}_T); \mathbf{z})]$. Plugging (45) into (40), we have

$$\Phi(\bar{\mathbf{x}}_T) - \Phi(\mathbf{x}^*)$$

$$\leq \frac{\|\nabla\Phi(\bar{\mathbf{x}}_T)\|^2}{2\mu_{\mathbf{x}}}$$

$$\leq O\left(\frac{1}{T}\right) + \frac{128\beta^3\mathbb{E}[f(\hat{\mathbf{x}}^*, \mathbf{y}^*(\hat{\mathbf{x}}^*); \mathbf{z})]\log\frac{4}{\delta}}{\mu_{\mathbf{x}}\mu_{\mathbf{y}}^2 n} + \frac{128\beta\mathbb{E}[f(\mathbf{x}^*, \mathbf{y}^*; \mathbf{z})]\log\frac{8}{\delta}}{\mu_{\mathbf{x}} n}$$

$$+ \frac{4\left(\frac{2\beta B_{\mathbf{y}^*}}{\mu_{\mathbf{y}}}\log\frac{4}{\delta} + 2B_{\mathbf{x}^*}\log\frac{8}{\delta} + \mu_{\mathbf{x}}\right)^2}{\mu_{\mathbf{x}} n^2}$$

$$= O\left(\frac{1}{T}\right) + \frac{128\beta^3\Phi(\hat{\mathbf{x}}^*)\log\frac{4}{\delta}}{\mu_{\mathbf{x}}\mu_{\mathbf{y}}^2 n} + \frac{128\beta\Phi(\mathbf{x}^*)\log\frac{8}{\delta}}{\mu_{\mathbf{x}} n} + \frac{4\left(\frac{2\beta B_{\mathbf{y}^*}}{\mu_{\mathbf{y}}}\log\frac{4}{\delta} + 2B_{\mathbf{x}^*}\log\frac{8}{\delta} + \mu_{\mathbf{x}}\right)^2}{\mu_{\mathbf{x}} n^2},$$

which implies that

$$\left(1 - \frac{128\beta^3\log\frac{4}{\delta}}{\mu_{\mathbf{x}}\mu_{\mathbf{y}}^2 n}\right)(\Phi(\bar{\mathbf{x}}_T) - \Phi(\mathbf{x}^*))$$

$$= O\left(\frac{1}{T}\right) + \frac{128\beta\Phi(\mathbf{x}^*)\log\frac{8}{\delta}}{\mu_{\mathbf{x}} n} + \frac{128\beta^3\Phi(\mathbf{x}^*)\log\frac{4}{\delta}}{\mu_{\mathbf{x}}\mu_{\mathbf{y}}^2 n} + \frac{4\left(\frac{2\beta B_{\mathbf{y}^*}}{\mu_{\mathbf{y}}}\log\frac{4}{\delta} + 2B_{\mathbf{x}^*}\log\frac{8}{\delta} + \mu_{\mathbf{x}}\right)^2}{\mu_{\mathbf{x}} n^2}.$$

When $n \geq \max\left\{\frac{c\beta^2(\mu_{\mathbf{y}}+\beta)^2(d+\log\frac{8\log_2\sqrt{2}R_1 n+1}{\delta})}{\mu_{\mathbf{y}}^2\mu_{\mathbf{x}}^2}, \frac{128\beta^3\log\frac{4}{\delta}}{\mu_{\mathbf{x}}\mu_{\mathbf{y}}^2}\right\}$, and $T \asymp n^2$ we have

$$\Phi(\bar{\mathbf{x}}_T) - \Phi(\mathbf{x}^*) = O\left(\frac{\Phi(\mathbf{x}^*)\log\frac{1}{\delta}}{n - \frac{128\beta^3\log\frac{4}{\delta}}{\mu_{\mathbf{x}}\mu_{\mathbf{y}}^2}} + \frac{\log^2\frac{1}{\delta}}{n\left(n - \frac{128\beta^3\log\frac{4}{\delta}}{\mu_{\mathbf{x}}\mu_{\mathbf{y}}^2}\right)}\right).$$

The proof is complete. □

### D.3 STOCHASTIC GRADIENT DESCENT ASCENT

In this subsection, we present empirical optimization error bounds of primal functions for SGDA, which are motivated by (Lei et al., 2021; Li & Liu, 2021a). The proofs are standard in the literature (Nedić & Ozdaglar, 2009; Nemirovski et al., 2009) and we give the optimization error bounds with high probability. Firstly, we introduce two concentration inequalities for martingales.

**Lemma 17** ((Boucheron et al., 2013)). *Let $z_1, \ldots, z_n$ be a sequence of random variables such that $z_k$ may depend the previous variables $z_1, \ldots, z_{k-1}$ for all $k = 1, \ldots, n$. Consider a sequence of functionals $\xi_k(z_1, \ldots, z_k), k = 1, \ldots, n$. Assume $|\xi_k - \mathbb{E}_{z_k}[\xi_k]| \leq b_k$ for each $k$. Let $\delta \in (0, 1)$. With probability at least $1 - \delta$*

$$\sum_{k=1}^{n} \xi_k - \sum_{k=1}^{n} \mathbb{E}_{z_k}[\xi_k] \leq \left(2\sum_{k=1}^{n} b_k^2 \log\frac{1}{\delta}\right)^{\frac{1}{2}}.$$

**Lemma 18** ((Tarres & Yao, 2014)). *Let $\{\xi_k\}_{k\in\mathbb{N}}$ be a martingale difference sequence in $\mathbb{R}^d$. Suppose that almost surely $\|\xi_k\| \leq D$ and $\sum_{k=1}^{t} \mathbb{E}[\|\xi_k\|^2|\xi_1, \ldots, \xi_{k-1}] \leq \sigma_t^2$. Then, for any $\delta \in (0, 1)$, the following inequality holds with probability at least $1 - \delta$*

$$\max_{1 \leq j \leq t}\left\|\sum_{k=1}^{j} \xi_k\right\| \leq 2\left(\frac{D}{3} + \sigma_t\right)\log\frac{2}{\delta}.$$

The following lemma shows the optimization error bounds of primal function for SGDA.

**Lemma 19.** *Suppose Assumption 5 and 2 hold and let the stepsizes be chosen as $\eta_{\mathbf{x}_t} = \frac{1}{\mu_{\mathbf{x}}(t+t_0)}$ and $\eta_{\mathbf{y}_t} = \frac{1}{\mu_{\mathbf{y}}(t+t_0)}$, for any $\delta \in (0,1)$, with probability at least $1 - \delta$, then the optimization error of Algorithm 2 can be bounded by*

$$\Phi_S(\bar{\mathbf{x}}_T) - \Phi_S(\hat{\mathbf{x}}^*) \leq \frac{t_0(\mu_{\mathbf{x}}D_{\mathbf{x}} + \mu_{\mathbf{y}}D_{\mathbf{y}})}{2T} + \frac{L^2 \log(eT)}{2T}\left(\frac{1}{\mu_{\mathbf{x}}} + \frac{1}{\mu_{\mathbf{y}}}\right)$$

$$+ \frac{2(\sqrt{D_{\mathcal{X}}} + \sqrt{D_{\mathcal{Y}}})}{T}\left(\frac{2L}{3} + 2L\sqrt{T}\right)\log\frac{6}{\delta} + \frac{2L(\sqrt{D_{\mathcal{X}}} + \sqrt{D_{\mathcal{Y}}})\left(2T\log\frac{6}{\delta}\right)^{\frac{1}{2}}}{T}.$$

*Proof of Lemma 19.* This proof mainly follows from (Lei et al., 2021; Li & Liu, 2021a). Firstly, we have

$$\|\mathbf{x}_{t+1} - \mathbf{x}\|^2 = \|\mathbf{x}_t - \eta_{\mathbf{x}_t}\nabla_{\mathbf{x}}f(\mathbf{x}_t, \mathbf{y}_t; \mathbf{z}_{i_t}) - \mathbf{x}\|^2$$

$$= \|\mathbf{x}_t - \mathbf{x}\|^2 + \eta_{\mathbf{x}_t}^2\|\nabla_{\mathbf{x}}f(\mathbf{x}_t, \mathbf{y}_t; \mathbf{z}_{i_t})\|^2 + 2\eta_{\mathbf{x}_t}\langle\mathbf{x} - \mathbf{x}_t, \nabla_{\mathbf{x}}f(\mathbf{x}_t, \mathbf{y}_t; \mathbf{z}_{i_t})\rangle$$

$$\leq \|\mathbf{x}_t - \mathbf{x}\|^2 + \eta_{\mathbf{x}_t}^2 L^2 + 2\eta_{\mathbf{x}_t}\langle\mathbf{x} - \mathbf{x}_t, \nabla_{\mathbf{x}}f(\mathbf{x}_t, \mathbf{y}_t; \mathbf{z}_{i_t}) - \nabla_{\mathbf{x}}F_S(\mathbf{x}_t, \mathbf{y}_t)\rangle$$

$$+ 2\eta_{\mathbf{x}_t}\langle\mathbf{x} - \mathbf{x}_t, \nabla_{\mathbf{x}}F_S(\mathbf{x}_t, \mathbf{y}_t)\rangle,$$

where the first inequality holds because of Assumption 2. According to the strong convexity of $F_S(\cdot, \mathbf{y}_t)$, we have

$$2\eta_{\mathbf{x}_t}(F_S(\mathbf{x}_t, \mathbf{y}_t) - F_S(\mathbf{x}, \mathbf{y}_t)) \leq (1 - \eta_{\mathbf{x}_t}\mu_{\mathbf{x}})\|\mathbf{x}_t - \mathbf{x}\|^2 - \|\mathbf{x}_{t+1} - \mathbf{x}\|^2 + \eta_{\mathbf{x}_t}^2 L^2$$

$$+ 2\eta_{\mathbf{x}_t}\langle\mathbf{x} - \mathbf{x}_t, \nabla_{\mathbf{x}}f(\mathbf{x}_t, \mathbf{y}_t; \mathbf{z}_{i_t}) - \nabla_{\mathbf{x}}F_S(\mathbf{x}_t, \mathbf{y}_t)\rangle.$$

Let $\eta_{\mathbf{x}_t} = \frac{1}{\mu_{\mathbf{x}}(t+t_0)}$, we further get

$$\frac{2}{\mu_{\mathbf{x}}(t+t_0)}(F_S(\mathbf{x}_t, \mathbf{y}_t) - F_S(\mathbf{x}, \mathbf{y}_t)) \leq \left(1 - \frac{1}{t+t_0}\right)\|\mathbf{x}_t - \mathbf{x}\|^2 - \|\mathbf{x}_{t+1} - \mathbf{x}\|^2$$

$$+ \left(\frac{L}{\mu_{\mathbf{x}}(t+t_0)}\right)^2 + \frac{2}{\mu_{\mathbf{x}}(t+t_0)}\langle\mathbf{x} - \mathbf{x}_t, \nabla_{\mathbf{x}}f(\mathbf{x}_t, \mathbf{y}_t; \mathbf{z}_{i_t}) - \nabla_{\mathbf{x}}F_S(\mathbf{x}_t, \mathbf{y}_t)\rangle.$$

Multiplying both sides by $t + t_0$, we have

$$\frac{2}{\mu_{\mathbf{x}}}(F_S(\mathbf{x}_t, \mathbf{y}_t) - F_S(\mathbf{x}, \mathbf{y}_t)) \leq (t + t_0 - 1)\|\mathbf{x}_t - \mathbf{x}\|^2 - (t+t_0)\|\mathbf{x}_{t+1} - \mathbf{x}\|^2$$

$$+ \frac{L^2}{\mu_{\mathbf{x}}^2(t+t_0)} + \frac{2}{\mu_{\mathbf{x}}}\langle\mathbf{x} - \mathbf{x}_t, \nabla_{\mathbf{x}}f(\mathbf{x}_t, \mathbf{y}_t; \mathbf{z}_{i_t}) - \nabla_{\mathbf{x}}F_S(\mathbf{x}_t, \mathbf{y}_t)\rangle.$$

Since $\mathbf{x}_1 = 0$ and $\sum_{t=1}^T t^{-1} \leq \log(eT)$, by taking a summation of the above inequality from $t = 1$ to $T$, we have

$$\sum_{t=1}^T (F_S(\mathbf{x}_t, \mathbf{y}_t) - F_S(\mathbf{x}, \mathbf{y}_t)) \leq \frac{\mu_{\mathbf{x}}}{2}t_0 D_{\mathcal{X}} + \frac{L^2 \log(eT)}{2\mu_{\mathbf{x}}}$$

$$+ \sum_{t=1}^T \langle\mathbf{x}, \nabla_{\mathbf{x}}f(\mathbf{x}_t, \mathbf{y}_t; \mathbf{z}_{i_t}) - \nabla_{\mathbf{x}}F_S(\mathbf{x}_t, \mathbf{y}_t)\rangle + \sum_{t=1}^T \langle\mathbf{x}_t, \nabla_{\mathbf{x}}F_S(\mathbf{x}_t, \mathbf{y}_t) - \nabla_{\mathbf{x}}f(\mathbf{x}_t, \mathbf{y}_t; \mathbf{z}_{i_t})\rangle.$$

Since $\mathbf{y}_S^*(\mathbf{x}) = \arg\max_{y \in \mathcal{Y}} F_S(\mathbf{x}, \mathbf{y})$, for any $\mathbf{x} \in \mathcal{X}$, we obtain that $F_S(\mathbf{x}, \mathbf{y}_t) \leq F_S(\mathbf{x}, \mathbf{y}_S^*(\mathbf{x}))$. Then we have

$$\sum_{t=1}^T (F_S(\mathbf{x}_t, \mathbf{y}_t) - F_S(\mathbf{x}, \mathbf{y}_S^*(\mathbf{x}))) \leq \frac{\mu_{\mathbf{x}}}{2}t_0 D_{\mathcal{X}} + \frac{L^2 \log(eT)}{2\mu_{\mathbf{x}}}$$

$$+ \sum_{t=1}^T \langle\mathbf{x}, \nabla_{\mathbf{x}}f(\mathbf{x}_t, \mathbf{y}_t; \mathbf{z}_{i_t}) - \nabla_{\mathbf{x}}F_S(\mathbf{x}_t, \mathbf{y}_t)\rangle + \sum_{t=1}^T \langle\mathbf{x}_t, \nabla_{\mathbf{x}}F_S(\mathbf{x}_t, \mathbf{y}_t) - \nabla_{\mathbf{x}}f(\mathbf{x}_t, \mathbf{y}_t; \mathbf{z}_{i_t})\rangle.$$

Since this inequality holds for any $\mathbf{x}$, we get

$$
\sum_{t=1}^{T}(F_S(\mathbf{x}_t, \mathbf{y}_t) - \inf_{\mathbf{x}\in\mathcal{X}} F_S(\mathbf{x}, \mathbf{y}_S^*(\mathbf{x}))) \leq \frac{\mu_{\mathbf{x}}}{2}t_0 D_{\mathcal{X}} + \frac{L^2 \log(eT)}{2\mu_{\mathbf{x}}}
$$
$$
+ \sum_{t=1}^{T} \sup_{x\in\mathcal{X}}\langle \mathbf{x}, \nabla_{\mathbf{x}}f(\mathbf{x}_t, \mathbf{y}_t; \mathbf{z}_{i_t}) - \nabla_{\mathbf{x}}F_S(\mathbf{x}_t, \mathbf{y}_t)\rangle + \sum_{t=1}^{T}\langle \mathbf{x}_t, \nabla_{\mathbf{x}}F_S(\mathbf{x}_t, \mathbf{y}_t) - \nabla_{\mathbf{x}}f(\mathbf{x}_t, \mathbf{y}_t; \mathbf{z}_{i_t})\rangle,
$$

which implies that

$$
\sum_{t=1}^{T}(F_S(\mathbf{x}_t, \mathbf{y}_t) - \Phi_S(\hat{\mathbf{x}}^*)) \leq \frac{\mu_{\mathbf{x}}}{2}t_0 D_{\mathcal{X}} + \frac{L^2 \log(eT)}{2\mu_{\mathbf{x}}}
$$
$$
+ \sum_{t=1}^{T} \sup_{x\in\mathcal{X}}\langle \mathbf{x}, \nabla_{\mathbf{x}}f(\mathbf{x}_t, \mathbf{y}_t; \mathbf{z}_{i_t}) - \nabla_{\mathbf{x}}F_S(\mathbf{x}_t, \mathbf{y}_t)\rangle + \sum_{t=1}^{T}\langle \mathbf{x}_t, \nabla_{\mathbf{x}}F_S(\mathbf{x}_t, \mathbf{y}_t) - \nabla_{\mathbf{x}}f(\mathbf{x}_t, \mathbf{y}_t; \mathbf{z}_{i_t})\rangle.
$$

By Schwarz's inequality, we have

$$
\sum_{t=1}^{T}(F_S(\mathbf{x}_t, \mathbf{y}_t) - \Phi_S(\hat{\mathbf{x}}^*)) \leq \frac{\mu_{\mathbf{x}}}{2}t_0 D_{\mathcal{X}} + \frac{L^2 \log(eT)}{2\mu_{\mathbf{x}}}
$$
$$
+ \sqrt{D_{\mathcal{X}}}\left\|\sum_{t=1}^{T}(\nabla_{\mathbf{x}}f(\mathbf{x}_t, \mathbf{y}_t; \mathbf{z}_{i_t}) - \nabla_{\mathbf{x}}F_S(\mathbf{x}_t, \mathbf{y}_t))\right\| + \sum_{t=1}^{T}\langle \mathbf{x}_t, \nabla_{\mathbf{x}}F_S(\mathbf{x}_t, \mathbf{y}_t) - \nabla_{\mathbf{x}}f(\mathbf{x}_t, \mathbf{y}_t; \mathbf{z}_{i_t})\rangle.
$$

Denote that $\xi_t = \langle \mathbf{x}_t, \nabla_{\mathbf{x}}F_S(\mathbf{x}_t, \mathbf{y}_t) - \nabla_{\mathbf{x}}f(\mathbf{x}_t, \mathbf{x}_t; \mathbf{z}_{i_t})\rangle$. Since $\mathbb{E}_{i_t}[\langle \mathbf{x}_t, \nabla_{\mathbf{x}}F_S(\mathbf{x}_t, \mathbf{y}_t) - \nabla_{\mathbf{x}}f(\mathbf{x}_t, \mathbf{y}_t; \mathbf{z}_{i_t})\rangle] = 0$, so $\{\xi_t | t = 1, \ldots, T\}$ is a martingale difference sequence. By Schwarz's inequality and Assumption 2, we know that $|\langle \mathbf{x}_t, \nabla_{\mathbf{x}}F_S(\mathbf{x}_t, \mathbf{y}_t) - \nabla_{\mathbf{x}}f(\mathbf{x}_t, \mathbf{y}_t; \mathbf{z}_{i_t})\rangle| \leq 2L\sqrt{D_{\mathcal{X}}}$. Then according to Lemma 17, we have the following inequality with probability at least $1 - \frac{\delta}{6}$

$$
\sum_{t=1}^{T}\langle \mathbf{x}_t, \nabla_{\mathbf{x}}F_S(\mathbf{x}_t, \mathbf{y}_t) - \nabla_{\mathbf{x}}f(\mathbf{x}_t, \mathbf{y}_t; \mathbf{z}_{i_t})\rangle \leq 2L\sqrt{D_{\mathcal{X}}}\left(2T\log\frac{6}{\delta}\right)^{\frac{1}{2}}.
$$

Define $\xi_t' = \nabla_{\mathbf{x}}f(\mathbf{x}_t, \mathbf{y}_t; \mathbf{z}_{i_t}) - \nabla_{\mathbf{x}}F_S(\mathbf{x}_t, \mathbf{y}_t)$. Then we get $\|\xi_t'\| \leq 2L$ and

$$
\sum_{t=1}^{T}\mathbb{E}[\|\xi_t'\|^2 | \xi_1', \ldots, \xi_{t-1}'] \leq 4TL^2.
$$

Applying Lemma 18 to the martingale difference sequence $\{\xi_t'\}$, we have the following inequality with probability at least $1 - \frac{\delta}{3}$

$$
\left\|\sum_{t=1}^{T}\xi_t'\right\| \leq 2\left(\frac{2L}{3} + 2L\sqrt{T}\right)\log\frac{6}{\delta}.
$$

This implies that with probability at least $1 - \frac{\delta}{3}$

$$
\left\|\sum_{t=1}^{T}(\nabla_{\mathbf{x}}f(\mathbf{x}_t, \mathbf{y}_t; \mathbf{z}_{i_t}) - \nabla_{\mathbf{x}}F_S(\mathbf{x}_t, \mathbf{y}_t))\right\| \leq 2\left(\frac{2L}{3} + 2L\sqrt{T}\right)\log\frac{6}{\delta}.
$$

Combined with the above results, we finally have the following inequality with probability at least $1 - \frac{\delta}{2}$

$$
\frac{1}{T}\sum_{t=1}^{T}(F_S(\mathbf{x}_t, \mathbf{y}_t) - \Phi_S(\hat{\mathbf{x}}^*)) \leq \frac{\mu_{\mathbf{x}}t_0 D_{\mathcal{X}}}{2T} + \frac{L^2 \log(eT)}{2\mu_{\mathbf{x}}T}
$$
$$
+ \frac{2\sqrt{D_{\mathcal{X}}}}{T}\left(\frac{2L}{3} + 2L\sqrt{T}\right)\log\frac{6}{\delta} + \frac{2L\sqrt{D_{\mathcal{X}}}\left(2T\log\frac{6}{\delta}\right)^{\frac{1}{2}}}{T}. \tag{46}
$$

Similarly, we can bound $\Phi(\bar{\mathbf{x}}_T) - \frac{1}{T}\sum_{t=1}^{T} F_S(\mathbf{x}_t, \mathbf{y}_t)$. Firstly, we have

$$
\begin{aligned}
\|\mathbf{y}_{t+1} - \mathbf{y}\|^2 &= \|\mathbf{y}_t + \eta_{\mathbf{y}_t}\nabla_{\mathbf{y}}f(\mathbf{x}_t, \mathbf{y}_t; \mathbf{z}_{i_t}) - \mathbf{y}\|^2 \\
&= \|\mathbf{y}_t - \mathbf{y}\|^2 + \eta_{\mathbf{y}_t}^2\|\nabla_{\mathbf{y}}f(\mathbf{x}_t, \mathbf{y}_t; \mathbf{z}_{i_t})\|^2 + 2\eta_{\mathbf{x}_y}\langle\mathbf{y}_t - \mathbf{y}, \nabla_{\mathbf{y}}f(\mathbf{x}_t, \mathbf{y}_t; \mathbf{z}_{i_t})\rangle \\
&\le \|\mathbf{y}_t - \mathbf{y}\|^2 + \eta_{\mathbf{y}_t}^2 L^2 + 2\eta_{\mathbf{y}_t}\langle\mathbf{y}_t - \mathbf{y}, \nabla_{\mathbf{y}}f(\mathbf{x}_t, \mathbf{y}_t; \mathbf{z}_{i_t}) - \nabla_{\mathbf{y}}F_S(\mathbf{x}_t, \mathbf{y}_t)\rangle \\
&\quad + 2\eta_{\mathbf{y}_t}\langle\mathbf{y}_t - \mathbf{y}, \nabla_{\mathbf{x}}F_S(\mathbf{x}_t, \mathbf{y}_t)\rangle,
\end{aligned}
$$

where the first inequality holds because of Assumption 2. According to the strong concavity of $F_S(\mathbf{x}_t, \cdot)$, we have

$$
\begin{aligned}
2\eta_{\mathbf{y}_t}(F_S(\mathbf{x}_t, \mathbf{y}) - F_S(\mathbf{x}_t, \mathbf{y}_t)) &\le (1 - \eta_{\mathbf{y}_t}\mu_{\mathbf{y}})\|\mathbf{y}_t - \mathbf{y}\|^2 - \|\mathbf{y}_{t+1} - \mathbf{y}\|^2 + \eta_{\mathbf{y}_t}^2 L^2 \\
&\quad + 2\eta_{\mathbf{y}_t}\langle\mathbf{y}_t - \mathbf{y}, \nabla_{\mathbf{y}}f(\mathbf{x}_t, \mathbf{y}_t; \mathbf{z}_{i_t}) - \nabla_{\mathbf{y}}F_S(\mathbf{x}_t, \mathbf{y}_t)\rangle.
\end{aligned}
$$

Let $\eta_{\mathbf{y}_t} = \frac{1}{\mu_{\mathbf{y}}(t+t_0)}$, we further get

$$
\begin{aligned}
\frac{2}{\mu_{\mathbf{y}}(t+t_0)}(F_S(\mathbf{x}_t, \mathbf{y}) - F_S(\mathbf{x}_t, \mathbf{y}_t)) &\le \left(1 - \frac{1}{t+t_0}\right)\|\mathbf{y}_t - \mathbf{y}\|^2 - \|\mathbf{y}_{t+1} - \mathbf{y}\|^2 \\
&\quad + \left(\frac{L}{\mu_{\mathbf{y}}(t+t_0)}\right)^2 + \frac{2}{\mu_{\mathbf{y}}(t+t_0)}\langle\mathbf{y}_t - \mathbf{y}, \nabla_{\mathbf{y}}f(\mathbf{x}_t, \mathbf{y}_t; \mathbf{z}_{i_t}) - \nabla_{\mathbf{y}}F_S(\mathbf{x}_t, \mathbf{y}_t)\rangle.
\end{aligned}
$$

Multiplying both sides by $t + t_0$, we have

$$
\begin{aligned}
\frac{2}{\mu_{\mathbf{y}}}(F_S(\mathbf{x}_t, \mathbf{y}) - F_S(\mathbf{x}_t, \mathbf{y}_t)) &\le (t+t_0-1)\|\mathbf{y}_t - \mathbf{y}\|^2 - (t+t_0)\|\mathbf{y}_{t+1} - \mathbf{y}\|^2 \\
&\quad + \frac{L^2}{\mu_{\mathbf{y}}^2(t+t_0)} + \frac{2}{\mu_{\mathbf{y}}}\langle\mathbf{y}_t - \mathbf{y}, \nabla_{\mathbf{y}}f(\mathbf{x}_t, \mathbf{y}_t; \mathbf{z}_{i_t}) - \nabla_{\mathbf{y}}F_S(\mathbf{x}_t, \mathbf{y}_t)\rangle.
\end{aligned}
$$

Since $\mathbf{y}_1 = 0$ and $\sum_{t=1}^{T} t^{-1} \le \log(eT)$, by taking a summation of the above inequality from $t = 1$ to $T$, we have

$$
\begin{aligned}
\sum_{t=1}^{T}(F_S(\mathbf{x}_t, \mathbf{y}) - F_S(\mathbf{x}_t, \mathbf{y}_t)) &\le \frac{\mu_{\mathbf{y}}}{2}t_0 D_{\mathcal{Y}} + \frac{L^2\log(eT)}{2\mu_{\mathbf{y}}} \\
&\quad + \sum_{t=1}^{T}\langle\mathbf{y}_t, \nabla_{\mathbf{y}}f(\mathbf{x}_t, \mathbf{y}_t; \mathbf{z}_{i_t}) - \nabla_{\mathbf{y}}F_S(\mathbf{x}_t, \mathbf{y}_t)\rangle + \sum_{t=1}^{T}\langle\mathbf{y}, \nabla_{\mathbf{y}}F_S(\mathbf{x}_t, \mathbf{y}_t) - \nabla_{\mathbf{y}}f(\mathbf{x}_t, \mathbf{y}_t; \mathbf{z}_{i_t})\rangle.
\end{aligned}
$$

From the convexity of $F_S(\cdot, \mathbf{y})$, we get

$$
\begin{aligned}
\sum_{t=1}^{T}(F_S(\bar{\mathbf{x}}_T, \mathbf{y}) - F_S(\mathbf{x}_t, \mathbf{y}_t)) &\le \frac{\mu_{\mathbf{y}}}{2}t_0 D_{\mathcal{Y}} + \frac{L^2\log(eT)}{2\mu_{\mathbf{y}}} \\
&\quad + \sum_{t=1}^{T}\langle\mathbf{y}_t, \nabla_{\mathbf{y}}f(\mathbf{x}_t, \mathbf{y}_t; \mathbf{z}_{i_t}) - \nabla_{\mathbf{y}}F_S(\mathbf{x}_t, \mathbf{y}_t)\rangle + \sum_{t=1}^{T}\langle\mathbf{y}, \nabla_{\mathbf{y}}F_S(\mathbf{x}_t, \mathbf{y}_t) - \nabla_{\mathbf{y}}f(\mathbf{x}_t, \mathbf{y}_t; \mathbf{z}_{i_t})\rangle.
\end{aligned}
$$

Since this inequality holds for any $\mathbf{y}$, we get

$$
\begin{aligned}
\sum_{t=1}^{T}(\sup_{y\in\mathcal{Y}} F_S(\bar{\mathbf{x}}_T, \mathbf{y}) - F_S(\mathbf{x}_t, \mathbf{y}_t)) &\le \frac{\mu_{\mathbf{y}}}{2}t_0 D_{\mathcal{Y}} + \frac{L^2\log(eT)}{2\mu_{\mathbf{y}}} \\
&\quad + \sum_{t=1}^{T}\langle\mathbf{y}_t, \nabla_{\mathbf{y}}f(\mathbf{x}_t, \mathbf{y}_t; \mathbf{z}_{i_t}) - \nabla_{\mathbf{y}}F_S(\mathbf{x}_t, \mathbf{y}_t)\rangle + \sum_{t=1}^{T}\sup_{y\in\mathcal{Y}}\langle\mathbf{y}, \nabla_{\mathbf{y}}F_S(\mathbf{x}_t, \mathbf{y}_t) - \nabla_{\mathbf{y}}f(\mathbf{x}_t, \mathbf{y}_t; \mathbf{z}_{i_t})\rangle,
\end{aligned}
$$

which implies that

$$
\begin{aligned}
\sum_{t=1}^{T}(\Phi_S(\bar{\mathbf{x}}_T) - F_S(\mathbf{x}_t, \mathbf{y}_t)) &\le \frac{\mu_{\mathbf{y}}}{2}t_0 D_{\mathcal{Y}} + \frac{L^2\log(eT)}{2\mu_{\mathbf{y}}} \\
&\quad + \sum_{t=1}^{T}\langle\mathbf{y}_t, \nabla_{\mathbf{y}}f(\mathbf{x}_t, \mathbf{y}_t; \mathbf{z}_{i_t}) - \nabla_{\mathbf{y}}F_S(\mathbf{x}_t, \mathbf{y}_t)\rangle + \sum_{t=1}^{T}\sup_{y\in\mathcal{Y}}\langle\mathbf{y}, \nabla_{\mathbf{y}}F_S(\mathbf{x}_t, \mathbf{y}_t) - \nabla_{\mathbf{y}}f(\mathbf{x}_t, \mathbf{y}_t; \mathbf{z}_{i_t})\rangle.
\end{aligned}
$$

By Schwarz's inequality, we have

$$
\sum_{t=1}^{T}(\Phi_S(\bar{\mathbf{x}}_T) - F_S(\mathbf{x}_t, \mathbf{y}_t)) \le \frac{\mu_{\mathbf{y}}}{2} t_0 D_{\mathcal{Y}} + \frac{L^2 \log(eT)}{2\mu_{\mathbf{y}}}
$$
$$
+ \sum_{t=1}^{T}\langle \mathbf{y}_t, \nabla_{\mathbf{y}} f(\mathbf{x}_t, \mathbf{y}_t; \mathbf{z}_{i_t}) - \nabla_{\mathbf{y}} F_S(\mathbf{x}_t, \mathbf{y}_t)\rangle + D_{\mathcal{Y}} \left\| \sum_{t=1}^{T}(\nabla_{\mathbf{y}} F_S(\mathbf{x}_t, \mathbf{y}_t) - \nabla_{\mathbf{y}} f(\mathbf{x}_t, \mathbf{y}_t; \mathbf{z}_{i_t})) \right\|.
$$

Denote that $\tilde{\xi}_t = \langle \mathbf{y}_t, \nabla_{\mathbf{y}} f(\mathbf{x}_t, \mathbf{x}_t; \mathbf{z}_{i_t}) - \nabla_{\mathbf{y}} F_S(\mathbf{x}_t, \mathbf{y}_t)\rangle$. Since $\mathbb{E}_{i_t}[\langle \mathbf{y}_t, \nabla_{\mathbf{y}} f(\mathbf{x}_t, \mathbf{x}_t; \mathbf{z}_{i_t}) - \nabla_{\mathbf{y}} F_S(\mathbf{x}_t, \mathbf{y}_t)\rangle] = 0$, so $\{\tilde{\xi}_t | t = 1, \dots, T\}$ is a martingale difference sequence. By Schwarz's inequality and Assumption 2, we know that $|\langle \mathbf{y}_t, \nabla_{\mathbf{y}} f(\mathbf{x}_t, \mathbf{x}_t; \mathbf{z}_{i_t}) - \nabla_{\mathbf{y}} F_S(\mathbf{x}_t, \mathbf{y}_t)\rangle| \le 2L\sqrt{D_{\mathcal{Y}}}$. Then according to Lemma 17 we have the following inequality with probability at least $1 - \frac{\delta}{6}$

$$
\sum_{t=1}^{T}\langle \mathbf{y}_t, \nabla_{\mathbf{y}} f(\mathbf{x}_t, \mathbf{x}_t; \mathbf{z}_{i_t}) - \nabla_{\mathbf{y}} F_S(\mathbf{x}_t, \mathbf{y}_t)\rangle \le 2L\sqrt{D_{\mathcal{Y}}} \left( 2T \log \frac{6}{\delta} \right)^{\frac{1}{2}}.
$$

Define $\tilde{\xi}'_t = \nabla_{\mathbf{y}} F_S(\mathbf{x}_t, \mathbf{y}_t) - \nabla_{\mathbf{y}} f(\mathbf{x}_t, \mathbf{y}_t; \mathbf{z}_{i_t})$. Then we get $\|\tilde{\xi}'_t\| \le 2L$ and

$$
\sum_{t=1}^{T} \mathbb{E}[\|\tilde{\xi}'_t\|^2 | \tilde{\xi}'_1, \dots, \tilde{\xi}'_{t-1}] \le 4TL^2.
$$

Applying Lemma 18 to the martingale difference sequence $\{\tilde{\xi}'_t\}$, we have the following inequality with probability at least $1 - \frac{\delta}{3}$

$$
\left\| \sum_{t=1}^{T} \tilde{\xi}'_t \right\| \le 2\left( \frac{2L}{3} + 2L\sqrt{T} \right) \log \frac{6}{\delta}.
$$

This implies that with probability at least $1 - \frac{\delta}{3}$

$$
\left\| \sum_{t=1}^{T} (\nabla_{\mathbf{y}} F_S(\mathbf{x}_t, \mathbf{y}_t) - \nabla_{\mathbf{y}} f(\mathbf{x}_t, \mathbf{y}_t; \mathbf{z}_{i_t})) \right\| \le 2\left( \frac{2L}{3} + 2L\sqrt{T} \right) \log \frac{6}{\delta}.
$$

Combined with the above results, we finally have the following inequality with probability at least $1 - \frac{\delta}{2}$

$$
\frac{1}{T}\sum_{t=1}^{T}(\Phi_S(\bar{\mathbf{x}}_T) - F_S(\mathbf{x}_t, \mathbf{y}_t)) \le \frac{\mu_{\mathbf{y}} t_0 D_{\mathcal{Y}}}{2T} + \frac{L^2 \log(eT)}{2\mu_{\mathbf{y}} T}
$$
$$
+ \frac{2\sqrt{D_{\mathcal{Y}}}}{T}\left( \frac{2L}{3} + 2L\sqrt{T} \right) \log \frac{6}{\delta} + \frac{2L\sqrt{D_{\mathcal{Y}}} \left( 2T \log \frac{6}{\delta} \right)^{\frac{1}{2}}}{T}.
\tag{47}
$$

Combing (46) and (47) together, with probability at least $1 - \delta$ we get the following inequality

$$
\Phi_S(\bar{\mathbf{x}}_T) - \Phi_S(\hat{\mathbf{x}}^*) \le \frac{t_0(\mu_{\mathbf{x}} D_{\mathbf{x}} + \mu_{\mathbf{y}} D_{\mathbf{y}})}{2T} + \frac{L^2 \log(eT)}{2T}\left( \frac{1}{\mu_{\mathbf{x}}} + \frac{1}{\mu_{\mathbf{y}}} \right)
$$
$$
+ \frac{2(\sqrt{D_{\mathcal{X}}} + \sqrt{D_{\mathcal{Y}}})}{T}\left( \frac{2L}{3} + 2L\sqrt{T} \right) \log \frac{6}{\delta} + \frac{2L(\sqrt{D_{\mathcal{X}}} + \sqrt{D_{\mathcal{Y}}}) \left( 2T \log \frac{6}{\delta} \right)^{\frac{1}{2}}}{T}.
$$

$\square$

*Proof of Theorem 8.* According to Lemma 9, $\Phi(\mathbf{x})$ satisfies the PL assumption with parameter $\mu_{\mathbf{x}}$, we have

$$
\Phi(\mathbf{x}) - \Phi(\mathbf{x}^*) \le \frac{\|\nabla\Phi(\mathbf{x})\|^2}{2\mu_{\mathbf{x}}}.
\tag{48}
$$

To bound $\Phi(\bar{\mathbf{x}}_T) - \Phi(\mathbf{x}^*)$, we need to bound the term $\|\nabla\Phi(\bar{\mathbf{x}}_T)\|^2$. There holds that

$$
\|\nabla\Phi(\bar{\mathbf{x}}_T)\|^2 \le 2\|\nabla\Phi(\bar{\mathbf{x}}_T) - \nabla\Phi_S(\bar{\mathbf{x}}_T)\|^2 + 2\|\nabla\Phi_S(\bar{\mathbf{x}}_T)\|^2.
\tag{49}
$$

From Theorem 3, under Assumption 3 and 5. Plugging $\bar{\mathbf{x}}_T$ into Theorem 3, for any $\delta \in (0, 1)$, when $n \geq \frac{c\beta^2(\mu_{\mathbf{y}}+\beta)^2(d+\log\frac{8\log_2\sqrt{2}R_1 n+1}{\delta})}{\mu_{\mathbf{y}}^2\mu_{\mathbf{x}}^2}$, with probability at least $1 - \frac{\delta}{2}$

$$\|\nabla\Phi(\bar{\mathbf{x}}_T) - \nabla\Phi_S(\bar{\mathbf{x}}_T)\| \leq \|\nabla\Phi_S(\bar{\mathbf{x}}_T)\| + 2\sqrt{\frac{2\mathbb{E}[\|\nabla_{\mathbf{x}}f(\mathbf{x}^*, \mathbf{y}^*(\mathbf{x}^*); \mathbf{z})\|^2]\log\frac{16}{\delta}}{n}}$$
$$+ \frac{2B_{\mathbf{x}^*}\log\frac{16}{\delta}}{n} + \frac{\mu_{\mathbf{x}}}{n} + \frac{2\beta}{\mu_{\mathbf{y}}}\left(\sqrt{\frac{2\mathbb{E}[\|\nabla_{\mathbf{y}}f(\bar{\mathbf{x}}_T, \mathbf{y}^*(\bar{\mathbf{x}}_T); \mathbf{z})\|^2]\log\frac{8}{\delta}}{n}} + \frac{B_{\mathbf{y}^*}\log\frac{8}{\delta}}{n}\right). \tag{50}$$

Next, we need to bound the optimization error bound $\|\nabla\Phi_S(\bar{\mathbf{x}}_T)\|$. Firstly, according to Lemma 19, with probability at least $1 - \delta$, we have

$$\Phi_S(\bar{\mathbf{x}}_T) - \Phi_S(\mathbf{x}^*) = O\left(\frac{\log\frac{1}{\delta}}{\sqrt{T}}\right).$$

According to (Nesterov, 2003) and Lemma 8, there holds the following property for $\beta + \frac{\beta^2}{\mu_{\mathbf{y}}}$ function $\Phi_S(\mathbf{x})$, we have

$$\frac{1}{2\left(\beta + \frac{\beta^2}{\mu_{\mathbf{y}}}\right)}\|\nabla\Phi_S(\bar{\mathbf{x}}_T)\|^2 \leq \Phi_S(\bar{\mathbf{x}}_T) - \Phi_S(\mathbf{x}^*) = O\left(\frac{\log\frac{1}{\delta}}{\sqrt{T}}\right). \tag{51}$$

Plugging (51) into (50), according to Cauchy–Bunyakovsky–Schwarz inequality, we can derive that

$$\|\nabla\Phi(\bar{\mathbf{x}}_T) - \nabla\Phi_S(\bar{\mathbf{x}}_T)\|^2 \leq O\left(\frac{\log\frac{1}{\delta}}{\sqrt{T}}\right) + \frac{32\beta^2\mathbb{E}[\|\nabla_{\mathbf{y}}f(\bar{\mathbf{x}}_T, \mathbf{y}^*(\bar{\mathbf{x}}_T); \mathbf{z})\|^2]\log\frac{8}{\delta}}{\mu_{\mathbf{y}}^2 n}$$
$$+ \frac{32\mathbb{E}[\|\nabla_{\mathbf{x}}f(\mathbf{x}^*, \mathbf{y}^*; \mathbf{z})\|^2]\log\frac{16}{\delta}}{n} + \frac{4\left(\frac{2\beta B_{\mathbf{y}^*}}{\mu_{\mathbf{y}}}\log\frac{8}{\delta} + 2B_{\mathbf{x}^*}\log\frac{16}{\delta} + \mu_{\mathbf{x}}\right)^2}{n^2}. \tag{52}$$

Then substituting (52), (51) into (49), we derive that

$$\|\nabla\Phi(\bar{\mathbf{x}}_T)\|^2 \leq O\left(\frac{\log\frac{1}{\delta}}{\sqrt{T}}\right) + \frac{64\beta^2\mathbb{E}[\|\nabla_{\mathbf{y}}f(\bar{\mathbf{x}}_T, \mathbf{y}^*(\bar{\mathbf{x}}_T); \mathbf{z})\|^2]\log\frac{8}{\delta}}{\mu_{\mathbf{y}}^2 n}$$
$$+ \frac{64\mathbb{E}[\|\nabla_{\mathbf{x}}f(\mathbf{x}^*, \mathbf{y}^*; \mathbf{z})\|^2]\log\frac{16}{\delta}}{n} + \frac{8\left(\frac{2\beta B_{\mathbf{y}^*}}{\mu_{\mathbf{y}}}\log\frac{8}{\delta} + 2B_{\mathbf{x}^*}\log\frac{16}{\delta} + \mu_{\mathbf{x}}\right)^2}{n^2}. \tag{53}$$

Finally, we plug (53) into (48) and choose $T \asymp O\left(n^2\right)$, with probability at least $1 - \delta$

$$\Phi(\bar{\mathbf{x}}_T) - \Phi(\mathbf{x}^*) = O\left(\frac{\mathbb{E}[\|\nabla_{\mathbf{x}}f(\mathbf{x}^*, \mathbf{y}^*; \mathbf{z})\|^2]\log\frac{1}{\delta}}{n} + \frac{\mathbb{E}[\|\nabla_{\mathbf{y}}f(\bar{\mathbf{x}}_T, \mathbf{y}^*(\bar{\mathbf{x}}_T); \mathbf{z})\|^2]\log\frac{1}{\delta}}{n} + \frac{\log^2\frac{1}{\delta}}{n^2}\right).$$

Next, if we further assume $\Phi(\mathbf{x}^*) = O\left(\frac{1}{n}\right)$, According to Lemma 7, we have $\mathbb{E}\|\nabla_{\mathbf{x}}f(\mathbf{x}^*, \mathbf{y}^*; \mathbf{z})\|^2 \leq 4\beta\mathbb{E}[f(\mathbf{x}^*, \mathbf{y}^*; \mathbf{z})]$ and $\mathbb{E}\|\nabla_{\mathbf{y}}f(\bar{\mathbf{x}}_T, \mathbf{y}^*(\bar{\mathbf{x}}_T); \mathbf{z})\|^2 \leq 4\beta\mathbb{E}[f(\bar{\mathbf{x}}_T, \mathbf{y}^*(\bar{\mathbf{x}}_T); \mathbf{z})]$. Plugging (53) into (48), then we have

$$\Phi(\bar{\mathbf{x}}_T) - \Phi(\mathbf{x}^*)$$
$$\leq \frac{\|\nabla\Phi(\bar{\mathbf{x}}_T)\|^2}{2\mu_{\mathbf{x}}}$$
$$\leq O\left(\frac{\log\frac{1}{\delta}}{\sqrt{T}}\right) + \frac{128\beta^3\mathbb{E}[f(\hat{\mathbf{x}}^*, \mathbf{y}^*(\hat{\mathbf{x}}^*); \mathbf{z})]\log\frac{8}{\delta}}{\mu_{\mathbf{x}}\mu_{\mathbf{y}}^2 n} + \frac{128\beta\mathbb{E}[f(\mathbf{x}^*, \mathbf{y}^*; \mathbf{z})]\log\frac{16}{\delta}}{\mu_{\mathbf{x}} n}$$
$$+ \frac{4\left(\frac{2\beta B_{\mathbf{y}^*}}{\mu_{\mathbf{y}}}\log\frac{8}{\delta} + 2B_{\mathbf{x}^*}\log\frac{16}{\delta} + \mu_{\mathbf{x}}\right)^2}{\mu_{\mathbf{x}} n^2}$$
$$= O\left(\frac{\log\frac{1}{\delta}}{\sqrt{T}}\right) + \frac{128\beta^3\Phi(\hat{\mathbf{x}}^*)\log\frac{8}{\delta}}{\mu_{\mathbf{x}}\mu_{\mathbf{y}}^2 n} + \frac{128\beta\Phi(\mathbf{x}^*)\log\frac{16}{\delta}}{\mu_{\mathbf{x}} n} + \frac{4\left(\frac{2\beta B_{\mathbf{y}^*}}{\mu_{\mathbf{y}}}\log\frac{8}{\delta} + 2B_{\mathbf{x}^*}\log\frac{16}{\delta} + \mu_{\mathbf{x}}\right)^2}{\mu_{\mathbf{x}} n^2},$$

| Reference | Algorithm | Assumption | Measure | Bounds |
|-----------|-----------|------------|---------|--------|
| Lei | SGDA | C-SC, Lip, S | (E.) $\Phi(\bar{\mathbf{x}}_T) - \Phi(\mathbf{x}^*)$ | $O(1/\sqrt{n})$ |
| | | C-SC, Lip, S | (HP.) $\Phi(\bar{\mathbf{x}}_T) - \Phi(\mathbf{x}^*)$ | $O(\log n/\sqrt{n})$ |
| | AGDA | PL-SC, Lip, S | (E.) $\Phi(\bar{\mathbf{x}}_T) - \Phi(\mathbf{x}^*)$ | $O(n^{-\frac{c\beta+1}{2c\beta+1}})$ |
| Li | ESP | SC-SC, Lip, S, LN | (HP.) $\Phi(\hat{\mathbf{x}}^*) - \Phi(\mathbf{x}^*)$ | $O(\log n/n)$ |
| | GDA | SC-SC, Lip, S, LN | (HP.) $\Phi(\bar{\mathbf{x}}_T) - \Phi(\mathbf{x}^*)$ | $O(\log n/n)$ |
| | SGDA | SC-SC, Lip, S, LN | (HP.) $\Phi(\bar{\mathbf{x}}_T) - \Phi(\mathbf{x}^*)$ | $O(\log n/n)$ |
| Zhang | Convergence | NC-SC, Lip, S | (E.) $\|\nabla\Phi(\mathbf{x}) - \nabla\Phi_S(\mathbf{x})\|$ | $O(\sqrt{d/n})$ |
| This work | Convergence | NC-SC, B, S | (H.P.) $\|\nabla\Phi(\mathbf{x}) - \nabla\Phi_S(\mathbf{x})\|$ | $O(\sqrt{d/n})$ |
| | ESP | PL-SC, B, S, A | (HP.) $\Phi(\hat{\mathbf{x}}^*) - \Phi(\mathbf{x}^*)$ | app. $O(1/n^2)$ |
| | GDA | PL-SC, B, S, A | (E.) $\Phi(\bar{\mathbf{x}}_T) - \Phi(\mathbf{x}^*)$ | app. $O(1/n^2)$ |
| | | SC-SC, B, S, A | (HP.) $\Phi(\bar{\mathbf{x}}_T) - \Phi(\mathbf{x}^*)$ | app. $O(1/n^2)$ |
| | SGDA | PL-SC, B, S, A | (E.) $\Phi(\bar{\mathbf{x}}_T) - \Phi(\mathbf{x}^*)$ | app. $O(1/n^2)$ |
| | | SC-SC, Lip, S, A | (HP.) $\Phi(\bar{\mathbf{x}}_T) - \Phi(\mathbf{x}^*)$ | app. $O(1/n^2)$ |
| | AGDA | PL-SC, B, S, A | (E.) $\Phi(\bar{\mathbf{x}}_T) - \Phi(\mathbf{x}^*)$ | app. $O(1/n^2)$ |

Table 1: Summary of the results. Lei means Lei et al. (2021), Li means Li & Liu (2021a), Zhang means Zhang et al. (2022), The bounds are established by choosing optimal iterate number $T$.

which implies that

$$\left(1 - \frac{128\beta^3 \log\frac{8}{\delta}}{\mu_{\mathbf{x}}\mu_{\mathbf{y}}^2 n}\right)(\Phi(\bar{\mathbf{x}}_T) - \Phi(\mathbf{x}^*))$$

$$=O\left(\frac{\log\frac{1}{\delta}}{\sqrt{T}}\right) + \frac{128\beta\Phi(\mathbf{x}^*)\log\frac{16}{\delta}}{\mu_{\mathbf{x}}n} + \frac{128\beta^3\Phi(\mathbf{x}^*)\log\frac{8}{\delta}}{\mu_{\mathbf{x}}\mu_{\mathbf{y}}^2 n} + \frac{4\left(\frac{2\beta B_{\mathbf{y}^*}}{\mu_{\mathbf{y}}}\log\frac{8}{\delta} + 2B_{\mathbf{x}^*}\log\frac{16}{\delta} + \mu_{\mathbf{x}}\right)^2}{\mu_{\mathbf{x}}n^2}.$$

When $n \geq \max\left\{\frac{c\beta^2(\mu_{\mathbf{y}}+\beta)^2(d+\log\frac{8\log_2\sqrt{2}R_1 n+1}{\delta})}{\mu_{\mathbf{y}}^2\mu_{\mathbf{x}}^2}, \frac{128\beta^3\log\frac{8}{\delta}}{\mu_{\mathbf{x}}\mu_{\mathbf{y}}^2}\right\}$, and $T \asymp n^4$ we have

$$\Phi(\bar{\mathbf{x}}_T) - \Phi(\mathbf{x}^*) = O\left(\frac{\Phi(\mathbf{x}^*)\log\frac{1}{\delta}}{n - \frac{128\beta^3\log\frac{8}{\delta}}{\mu_{\mathbf{x}}\mu_{\mathbf{y}}^2}} + \frac{\log^2\frac{1}{\delta}}{n\left(n - \frac{128\beta^3\log\frac{8}{\delta}}{\mu_{\mathbf{x}}\mu_{\mathbf{y}}^2}\right)}\right).$$

The proof is complete.

$\square$

# E  SOME IMPROVED BOUNDS WITH EXPECTATION FORMATS

Table 1 gives the summary of the existing results. Convergence means the uniform (localized) convergence results. AGDA is alternating gradient descent ascent algorithm proposed in Yang et al. (2020). Lip means Lipschitz continuity. S means smoothness. B means Bernstein condition. A means that we assume $\Phi(\mathbf{x}^*)$ is of order $O\left(\frac{1}{n}\right)$. LN means low noise condition. PL-SC means $\mathbf{x}$-side PL condition strongly concave settings. E. means expectation results. HP. means high probability results and app. $O(1/n^2)$ means the result is approximate $O(1/n^2)$ when $n$ is large enough. Since most of the existing work on optimization error is the expectation format, and our high probability results of generalization error can be transformed into the expectation results. so we give the proofs of the expectation result to relax some assumptions such as SC-SC condition in this section.

Firstly, we translate our high probability result of Theorem 3 into an expectation result.

**Theorem 9.** *Under Assumption 1 and 3, assume that the population risk $F(\mathbf{x}, \mathbf{y})$ satisfies Assumption 4 with parameter $\mu_{\mathbf{x}}$ and let $c = \max\{16C^2, 1\}$. We have that for all $\mathbf{x} \in \mathcal{X}$, when*

$n \geq \frac{c\beta^2(\mu_{\mathbf{y}}+\beta)^2(d+\log\frac{8\log_2\sqrt{2}R_1 n+1}{\delta})}{\mu_{\mathbf{y}}^2\mu_{\mathbf{x}}^2}$, *the excess risks of primal functions can be bounded by*

$$\mathbb{E}\|\Phi(\mathbf{x}) - \Phi(\mathbf{x}^*)\| \leq \frac{8\mathbb{E}[\|\nabla\Phi_S(\mathbf{x})\|^2]}{\mu_{\mathbf{x}}} + O\left(\frac{\mathbb{E}[\|\nabla_{\mathbf{x}}f(\mathbf{x}^*,\mathbf{y}^*;\mathbf{z})\|^2]}{\mu_{\mathbf{x}}n}\right.$$
$$\left. + \frac{\beta^2\mathbb{E}[\|\nabla_{\mathbf{y}}f(\mathbf{x},\mathbf{y}^*(\mathbf{x});\mathbf{z})\|^2]}{\mu_{\mathbf{x}}\mu_{\mathbf{y}}^2 n} + \frac{1}{n^2}\right).$$

*Proof of Theorem 9.* According to Theorem 3, we have that for all $\mathbf{x} \in \mathcal{X}$, when $n \geq \frac{c\beta^2(\mu_{\mathbf{y}}+\beta)^2(d+\log\frac{8\log_2\sqrt{2}R_1 n+1}{\delta})}{\mu_{\mathbf{y}}^2\mu_{\mathbf{x}}^2}$ with probability at least $1 - \delta$

$$\Phi(\mathbf{x}) - \Phi(\mathbf{x}^*) \leq \frac{8\|\nabla\Phi_S(\mathbf{x})\|^2}{\mu_{\mathbf{x}}} + \frac{16\beta^2\mathbb{E}[\|\nabla_{\mathbf{y}}f(\mathbf{x},\mathbf{y}^*(\mathbf{x});\mathbf{z})\|^2]\log\frac{4}{\delta}}{\mu_{\mathbf{x}}\mu_{\mathbf{y}}^2 n}$$
$$+ \frac{16\mathbb{E}[\|\nabla_{\mathbf{x}}f(\mathbf{x}^*,\mathbf{y}^*;\mathbf{z})\|^2]\log\frac{8}{\delta}}{\mu_{\mathbf{x}}n} + \frac{2\left(\frac{2\beta B_{\mathbf{y}^*}}{\mu_{\mathbf{y}}}\log\frac{4}{\delta} + 2B_{\mathbf{x}^*}\log\frac{8}{\delta} + \mu_{\mathbf{x}}\right)^2}{\mu_{\mathbf{x}}n^2}.$$

Thus, with probability at least $1 - \delta$, we have

$$\Phi(\mathbf{x}) - \Phi(\mathbf{x}^*) - \frac{8\|\nabla\Phi_S(\mathbf{x})\|^2}{\mu_{\mathbf{x}}} \leq \frac{16\beta^2\mathbb{E}[\|\nabla_{\mathbf{y}}f(\mathbf{x},\mathbf{y}^*(\mathbf{x});\mathbf{z})\|^2]\log\frac{4}{\delta}}{\mu_{\mathbf{x}}\mu_{\mathbf{y}}^2 n}$$
$$+ \frac{16\mathbb{E}[\|\nabla_{\mathbf{x}}f(\mathbf{x}^*,\mathbf{y}^*;\mathbf{z})\|^2]\log\frac{8}{\delta}}{\mu_{\mathbf{x}}n} + \frac{2\left(\frac{2\beta B_{\mathbf{y}^*}}{\mu_{\mathbf{y}}}\log\frac{4}{\delta} + 2B_{\mathbf{x}^*}\log\frac{8}{\delta} + \mu_{\mathbf{x}}\right)^2}{\mu_{\mathbf{x}}n^2}.$$

According to the standard statistical analysis, Let $X$ be a random variable, if for some $v, c > 0, \mathbb{P}\{X > vt + ct^2\} \leq e^{-t}$ for every $t > 0$. Then we can easily derive that $\mathbb{E}[X] = \int_0^\infty \mathbb{P}\{X > x\}dx = v + 2c$. Thus, we have the expectation result

$$\mathbb{E}\|\Phi(\mathbf{x}) - \Phi(\mathbf{x}^*)\| \leq \frac{8\mathbb{E}[\|\nabla\Phi_S(\mathbf{x})\|^2]}{\mu_{\mathbf{x}}} + O\left(\frac{\mathbb{E}[\|\nabla_{\mathbf{x}}f(\mathbf{x}^*,\mathbf{y}^*;\mathbf{z})\|^2]}{\mu_{\mathbf{x}}n}\right.$$
$$\left. + \frac{\beta^2\mathbb{E}[\|\nabla_{\mathbf{y}}f(\mathbf{x},\mathbf{y}^*(\mathbf{x});\mathbf{z})\|^2]}{\mu_{\mathbf{x}}\mu_{\mathbf{y}}^2 n} + \frac{1}{n^2}\right).$$

The proof is complete. $\qquad\square$

Next, we give the proofs of the expectation result to relax the SC-SC assumptions given in Table 1. The proofs are similar with high probability format since we use the existing results for optimization error bounds with expectation format under NC-SC assumptions.

### E.1 GDA

**Lemma 20** (Optimization error bound for GDA in NC-SC minimax problems (Lin et al., 2020))**.** *Under Assumption 1, and letting the step sizes be chosen as $\eta_{\mathbf{x}} = \frac{1}{16(\frac{\beta}{\mu}+1)^2\beta}$ and $\eta_{\mathbf{y}} = \frac{1}{\beta}$, then the optimization error bound of Algorithm 1 can be bounded by*

$$\mathbb{E}\|\nabla\Phi_S(\mathbf{x}_T)\|^2 = O\left(\frac{\beta^3\Delta_\Phi}{\mu_{\mathbf{y}}^2 T} + \frac{\beta^3 D_{\mathcal{Y}}}{\mu_{\mathbf{y}} T}\right),$$

*where $\Delta_\Phi = \Phi_S(\mathbf{x}_0) - \min_{\mathbf{x}}\Phi_S(\mathbf{x})$.*

Using above optimization error bound, we can obtain the following theorem.

**Theorem 10.** *Suppose Assumption 1 and 3 hold. Assume that the population risk $F(\mathbf{x}, \mathbf{y})$ satisfies Assumption 4 with parameter $\mu_{\mathbf{x}}$. Let the step sizes choose as $\eta_{\mathbf{x}} = \frac{1}{16(\frac{\beta}{\mu}+1)^2\beta}$ and $\eta_{\mathbf{y}} = \frac{1}{\beta}$. When $T \asymp n$ and $n \geq \frac{c\beta^2(\mu_{\mathbf{y}}+\beta)^2(d+\log\frac{8\log_2\sqrt{2}R_1 n+1}{\delta})}{\mu_{\mathbf{y}}^2\mu_{\mathbf{x}}^2}$, where $c$ is an absolute constant, then the excess risk for primal functions of Algorithm 1 can be bounded by*

$$\mathbb{E}[\Phi(\mathbf{x}_T) - \Phi(\mathbf{x}^*)] = O\left(\frac{\mathbb{E}[\|\nabla_{\mathbf{x}} f(\mathbf{x}^*, \mathbf{y}^*; \mathbf{z})\|^2]}{n} + \frac{\mathbb{E}[\|\nabla_{\mathbf{y}} f(\mathbf{x}_T, \mathbf{y}^*(\mathbf{x}_T); \mathbf{z})\|^2]}{n} + \frac{1}{n^2}\right).$$

*Furthermore, Let $T \asymp n^2$ and $n \geq \max\left\{\frac{c\beta^2(\mu_{\mathbf{y}}+\beta)^2(d+\log\frac{8\log_2\sqrt{2}R_1 n+1}{\delta})}{\mu_{\mathbf{y}}^2\mu_{\mathbf{x}}^2}, \Theta\left(\frac{\beta^3}{\mu_{\mathbf{x}}\mu_{\mathbf{y}}^2}\right)\right\}$. Assume the function $f(\mathbf{x}, \mathbf{y}; \mathbf{z})$ is non-negative and $\Phi(\mathbf{x}^*) = O\left(\frac{1}{n}\right)$, we have*

$$\mathbb{E}[\Phi(\mathbf{x}_T) - \Phi(\mathbf{x}^*)] \approx O\left(\frac{1}{n^2}\right).$$

### E.2   SGDA

For SGDA settings, we introduce a weaker assumption comparing with Assumption 2.

**Assumption 6.** *Assume the existence of $\sigma > 0$ satisfies*

$$\mathbb{E}[\nabla f(\mathbf{x}, \mathbf{y}; \mathbf{z}) - \nabla F_S(\mathbf{x}, \mathbf{y})] = 0,$$
$$\mathbb{E}[\|\nabla f(\mathbf{x}, \mathbf{y}; \mathbf{z}) - \nabla F_S(\mathbf{x}, \mathbf{y})\|^2] \leq \sigma^2.$$

**Lemma 21** (Optimization error bound for SGDA in NC-SC minimax problems (Lin et al., 2020)). *Under Assumption 1 and 6, and letting the step sizes be chosen as $\eta_{\mathbf{x}} = \frac{1}{16(\frac{\beta}{\mu}+1)^2\beta}$ and $\eta_{\mathbf{y}} = \frac{1}{\beta}$, then the optimization error bound of Algorithm 1 can be bounded by*

$$\mathbb{E}\|\nabla\Phi_S(\mathbf{x}_T)\|^2 = O\left(\sqrt[5]{\frac{\beta^6}{\mu_{\mathbf{y}}^6 T^2}}\right).$$

Using above optimization error bound, we can obtain the following theorem.

**Theorem 11.** *Suppose Assumption 1, 3 and 6 hold. Assume that the population risk $F(\mathbf{x}, \mathbf{y})$ satisfies Assumption 4 with parameter $\mu_{\mathbf{x}}$. Let the step sizes choose as $\eta_{\mathbf{x}} = \frac{1}{16(\frac{\beta}{\mu}+1)^2\beta}$ and $\eta_{\mathbf{y}} = \frac{1}{\beta}$. When $T \asymp n^{\frac{5}{2}}$ and $n \geq \frac{c\beta^2(\mu_{\mathbf{y}}+\beta)^2(d+\log\frac{8\log_2\sqrt{2}R_1 n+1}{\delta})}{\mu_{\mathbf{y}}^2\mu_{\mathbf{x}}^2}$, where $c$ is an absolute constant, then the excess risk for primal functions of Algorithm 2 can be bounded by*

$$\mathbb{E}[\Phi(\mathbf{x}_T) - \Phi(\mathbf{x}^*)] = O\left(\frac{\mathbb{E}[\|\nabla_{\mathbf{x}} f(\mathbf{x}^*, \mathbf{y}^*; \mathbf{z})\|^2]}{n} + \frac{\mathbb{E}[\|\nabla_{\mathbf{y}} f(\mathbf{x}_T, \mathbf{y}^*(\mathbf{x}_T); \mathbf{z})\|^2]}{n} + \frac{1}{n^2}\right).$$

*Furthermore, Let $T \asymp n^5$ and $n \geq \max\left\{\frac{c\beta^2(\mu_{\mathbf{y}}+\beta)^2(d+\log\frac{8\log_2\sqrt{2}R_1 n+1}{\delta})}{\mu_{\mathbf{y}}^2\mu_{\mathbf{x}}^2}, \Theta\left(\frac{\beta^3}{\mu_{\mathbf{x}}\mu_{\mathbf{y}}^2}\right)\right\}$. Assume the function $f(\mathbf{x}, \mathbf{y}; \mathbf{z})$ is non-negative and $\Phi(\mathbf{x}^*) = O\left(\frac{1}{n}\right)$, we have*

$$\mathbb{E}[\Phi(\mathbf{x}_T) - \Phi(\mathbf{x}^*)] \approx O\left(\frac{1}{n^2}\right).$$

### E.3   AGDA

Alternating gradient descent ascent presented in Algorithm 3 was proposed recently to optimize nonconvex-nonconcave problems (Yang et al., 2020).

---

**Algorithm 3** Two-timescale AGDA for minimax problem

---

1: **Input:** $(\mathbf{x}_1, \mathbf{y}_1) = (0, 0)$, step sizes $\{\eta_{\mathbf{x}_t}\}_t > 0$, $\{\eta_{\mathbf{y}_t}\}_t > 0$ and dataset $S = \{\mathbf{z}_1, \ldots, \mathbf{z}_n\}$
2: **for** $t = 1, \ldots, T$ **do**
3:     update $\mathbf{x}_{t+1} = \mathbf{x}_t - \eta_{\mathbf{x}_t}\nabla_{\mathbf{x}} f(\mathbf{x}_t, \mathbf{y}_t; \mathbf{z}_{i_t})$
4:     update $\mathbf{y}_{t+1} = \mathbf{y}_t + \eta_{\mathbf{y}_t}\nabla_{\mathbf{y}} f(\mathbf{x}_{t+1}, \mathbf{y}_t; \mathbf{z}_{i_t})$

---

**Lemma 22** (Optimization error bound for AGDA in PL-SC minimax problems (Yang et al., 2020; Lei et al., 2021))**.** *Under Assumption 1 and assume that the population risk $F(\mathbf{x}, \mathbf{y})$ satisfies Assumption 4 with parameter $\mu_\mathbf{x}$. Let $\{\mathbf{x}_t, \mathbf{y}_t\}_t$ be the sequence produced by Algorithm 3 with the step sizes chosen as $\eta_{\mathbf{x}_t} \asymp \frac{1}{\mu_\mathbf{x} t}$ and $\eta_{\mathbf{y}_t} = \frac{1}{\mu_\mathbf{x} \mu_\mathbf{y}^2 t}$, then the optimization error bound of Algorithm 3 can be bounded by*

$$\mathbb{E}[\|\nabla \Phi_S(\mathbf{x}_T)\|^2] = O\left(\frac{1}{\mu_\mathbf{x}^2 \mu_\mathbf{y}^4 T}\right).$$

**Theorem 12.** *Suppose Assumption 1, 3 and 6 hold. Assume that the population risk $F(\mathbf{x}, \mathbf{y})$ satisfies Assumption 4 with parameter $\mu_\mathbf{x}$. Let $\{\mathbf{x}_t, \mathbf{y}_t\}_t$ be the sequence produced by Algorithm 3 with the step sizes chosen as $\eta_{\mathbf{x}_t} \asymp \frac{1}{\mu_\mathbf{x} t}$ and $\eta_{\mathbf{y}_t} \asymp \frac{1}{\mu_\mathbf{x} \mu_\mathbf{y}^2 t}$. When $T \asymp n$ and $n \geq \frac{c\beta^2(\mu_\mathbf{y}+\beta)^2(d+\log\frac{8\log_2\sqrt{2}R_1 n+1}{\delta})}{\mu_\mathbf{y}^2 \mu_\mathbf{x}^2}$, where $c$ is an absolute constant, then the excess risk for primal functions of Algorithm 3 can be bounded by*

$$\mathbb{E}[\Phi(\mathbf{x}_T) - \Phi(\mathbf{x}^*)] = O\left(\frac{\mathbb{E}[\|\nabla_\mathbf{x} f(\mathbf{x}^*, \mathbf{y}^*; \mathbf{z})\|^2]}{n} + \frac{\mathbb{E}[\|\nabla_\mathbf{y} f(\mathbf{x}_T, \mathbf{y}^*(\mathbf{x}_T); \mathbf{z})\|^2]}{n} + \frac{1}{n^2}\right).$$

*Furthermore, Let $T \asymp n^2$ and $n \geq \max\left\{\frac{c\beta^2(\mu_\mathbf{y}+\beta)^2(d+\log\frac{8\log_2\sqrt{2}R_1 n+1}{\delta})}{\mu_\mathbf{y}^2 \mu_\mathbf{x}^2}, \Theta\left(\frac{\beta^3}{\mu_\mathbf{x} \mu_\mathbf{y}^2}\right)\right\}$. Assume the function $f(\mathbf{x}, \mathbf{y}; \mathbf{z})$ is non-negative and $\Phi(\mathbf{x}^*) = O\left(\frac{1}{n}\right)$, we have*

$$\mathbb{E}[\Phi(\mathbf{x}_T) - \Phi(\mathbf{x}^*)] \approx O\left(\frac{1}{n^2}\right).$$

*Proof of Theorem 12.* Since the proofs of Theorem 10, 11 and 12 are similar, we only prove Theorem 12 as an example.

From Theorem 9, under Assumption 3 and 1, when we plug $\mathbf{x}_T$ into Theorem 9, when $n \geq \frac{c\beta^2(\mu_\mathbf{y}+\beta)^2(d+\log\frac{8\log_2\sqrt{2}R_1 n+1}{\delta})}{\mu_\mathbf{y}^2 \mu_\mathbf{x}^2}$, we have

$$\begin{aligned}
\mathbb{E}\|\Phi(\mathbf{x}_T) - \Phi(\mathbf{x}^*)\| \leq &\frac{8\mathbb{E}[\|\nabla\Phi_S(\mathbf{x}_T)\|^2]}{\mu_\mathbf{x}} + O\left(\frac{\mathbb{E}[\|\nabla_\mathbf{x} f(\mathbf{x}^*, \mathbf{y}^*; \mathbf{z})\|^2]}{\mu_\mathbf{x} n}\right.\\
&\left.+ \frac{\beta^2 \mathbb{E}[\|\nabla_\mathbf{y} f(\mathbf{x}_T, \mathbf{y}^*(\mathbf{x}_T); \mathbf{z})\|^2]}{\mu_\mathbf{x} \mu_\mathbf{y}^2 n} + \frac{1}{n^2}\right).
\end{aligned} \tag{54}$$

According to Lemma 22, we have

$$\mathbb{E}[\|\nabla\Phi_S(\mathbf{x}_T)\|^2] = O\left(\frac{1}{\mu_\mathbf{x}^2 \mu_\mathbf{y}^4 T}\right). \tag{55}$$

Plugging (55) into (54) and choose $T \asymp O(n)$, with probability at least $1 - \delta$

$$\mathbb{E}[\Phi(\mathbf{x}_T) - \Phi(\mathbf{x}^*)] = O\left(\frac{\mathbb{E}[\|\nabla_\mathbf{x} f(\mathbf{x}^*, \mathbf{y}^*; \mathbf{z})\|^2]}{\mu_\mathbf{x} n} + \frac{\beta^2 \mathbb{E}[\|\nabla_\mathbf{y} f(\mathbf{x}_T, \mathbf{y}^*(\mathbf{x}_T); \mathbf{z})\|^2]}{\mu_\mathbf{x} \mu_\mathbf{y}^2 n} + \frac{1}{n^2}\right).$$

Next, if we further assume $\Phi(\mathbf{x}^*) = O\left(\frac{1}{n}\right)$, According to Lemma 7, we have $\mathbb{E}\|\nabla_\mathbf{x} f(\mathbf{x}^*, \mathbf{y}^*; \mathbf{z})\|^2 \leq 4\beta\mathbb{E}[f(\mathbf{x}^*, \mathbf{y}^*; \mathbf{z})]$ and $\mathbb{E}\|\nabla_\mathbf{y} f(\mathbf{x}_T, \mathbf{y}^*(\mathbf{x}_T); \mathbf{z})\|^2 \leq 4\beta\mathbb{E}[f(\mathbf{x}_T, \mathbf{y}^*(\mathbf{x}_T); \mathbf{z})]$, then substituting (55) into (54) and choosing $T \asymp n^2$, we have

$$\mathbb{E}[\Phi(\mathbf{x}_T) - \Phi(\mathbf{x}^*)] = O\left(\frac{\beta^3 \mathbb{E}[\Phi(\mathbf{x}_T) - \Phi(\mathbf{x}^*)]}{\mu_\mathbf{x} \mu_\mathbf{y}^2 n} + \frac{\beta^3 \mathbb{E}[\Phi(\mathbf{x}^*)]}{\mu_\mathbf{x} \mu_\mathbf{y}^2 n} + \frac{\beta\mathbb{E}[\Phi(\mathbf{x}^*)]}{\mu_\mathbf{x} n} + \frac{1}{n^2}\right).$$

which implies that When $n \geq \max \left\{ \frac{c\beta^2(\mu_{\mathbf{y}}+\beta)^2(d+\log \frac{8 \log_2 \sqrt{2}R_1 n+1}{\delta})}{\mu_{\mathbf{y}}^2 \mu_{\mathbf{x}}^2}, \Theta\left(\frac{\beta^3}{\mu_{\mathbf{x}}\mu_{\mathbf{y}}^2}\right) \right\}$, we have

$$\mathbb{E}[\Phi(\mathbf{x}_T) - \Phi(\mathbf{x}^*)] \approx O\left(\frac{1}{n^2}\right).$$

The proof is complete.

$\square$