# OpenReview forum: "Uniform Localized Convergence and Sharper Generalization Bounds for Minimax Problems"
_ICLR.cc/2024/Conference — Submitted to ICLR 2024_

### Official Review · Reviewer_QX5y · 2023-10-24

**Soundness:** 2 fair
**Presentation:** 3 good
**Contribution:** 2 fair
**Rating:** 5
**Confidence:** 4

**Summary:**

This paper derives generalization bounds for learning with minimax objectives, assuming strong convexity of the dual problem and smoothness of the objective function. It establishes uniform convergence bounds for primal function gradients and provides convergence bounds for the empirical primal function to the primal risk function under a PL condition. The paper also applies these findings to algorithms like empirical saddle point, gradient descent ascent, and stochastic gradient descent ascent.

**Strengths:**

The paper presents a high-probability analysis for minimax problems which is more challenging than bounds in expectation.
It studies several popular algorithms such as empirical saddle point, gradient descent ascent, and stochastic gradient descent ascent.

**Weaknesses:**

some comments are listed below

1.  the novelty of the proof techniques is not clear to me. Most of them are just incremental compared with the existing work

2. The proof seems to have issues.  For example, the problem is in Eq (27). It only holds if $x^*$ is the projection of x to the set of solutions. That is if the problem does not have a unique solution, this x^* should be different for different x. Then one cannot apply Lemma 11 since Lemma 11 only holds for a fixed $x$.

Then Eq (24) does not hold since it is derived by applying Lemma 11 for all $x$

**Questions:**

See above

---

> ### Author Response · Authors · 2023-11-23
>
> Thanks for your time and thoughts. We will answer all your questions.
>
> Question 1: The novelty of the proof techniques is not clear to me ...
>
> Answer: There are two mainly challenges in our work in generalization bounds for minimax problems. On one hand, comparing with uniform convergence in [Zhang et al. 2022], we use a novel uniform localized convergence techniques [Xu and Zeevi, 2020] to construct a functional w.r.t. loss functions on minimax problems. This two layer structure involves difficulties. On the other hand, it is noteworthy that the optimal point $y^*(x) := argmax_{y \in \mathcal{Y}} F(x, y)$ for a given $x$ differs from $y_S^*(x) := argmax_{y \in \mathcal{Y}} F_S(x, y)$, thus introducing an additional error term $|| y^*(x) - y_S^*(x) ||$. Compared to [Zhang et al. 2022], they only need to bound this term with $O(\frac{1}{\sqrt{n}})$. But we need to reach the upper bound of order $O(\frac{1}{n})$ under certain assumptions.
>
> For applications, we found that all the related optimization papers in minimax problems focused on the iteration complexity (or gradient complexity). As a result, they only require $\mathbb{E}[||\nabla \Phi(x_T)||]$ for average of round $T$ with an expected outcome. We were compelled to derive the high probability empirical optimization bound ourselves using classical optimization methods under SC-SC conditions.
>
> Notice that even under SC-SC settings, achieving this for SGDA remains difficult. Drawing inspiration from [Lei et al., 2021]'s proof of Primal-Dual Risk optimization bound, we eventually derive the optimization bound for primal risk. The proofs for excess risks in applications differ from minimization problems [Li et al., 2021] and pose challenges. These challenges stem from errors in the inner and outer layers of minimax problems. Consequently, we can only achieve a result close to $O(1/n^2)$.
>
>
> Question 2: The proof seems to have issues. For example, ...
>
> Answer: We believe our proof is correct. Here we discuss why we don't require uniform convergence. For population risk $R(\theta) = E_{z} l(\theta, z)$, empirical risk $r(\theta) = \frac1n \sum_{i=1}^n l(\theta, z_i)$ and fixed $\theta$, we can directly apply concentration inequality. But we can't apply them to $\theta ' $(ERM) as it is a function of the dataset. We need to establish the sup that $R(\theta ') - r(\theta ') \leq \sup_{\theta_{sup}} [R(\theta_{sup}) - r(\theta_{sup})]$. Yet, as the dataset changes, the parameters $\theta_{sup}$ change accordingly. Thus, we require uniform convergence for function $R -r$. In Lemma 11 of our proof, for any $x$, when the dataset changes, $x$ and $y^*(x)$ remain unchanged, and $\nabla F_S(x, y^*(x))$ are the only altered random variables/vectors. Consequently, we can directly apply the Bernstein inequality, and Eq (32) holds for any $x$. In other words, we believe that we only need pointwise convergence for the function ${y}^*({x}) - {y}_S^*({x})$ w.r.t. ${x}$, which suffices.
>
> Furthermore, let $ {x}' = argmax_{{x}'}[{y}^*({x}') - {y}_S^*({x}')]$. Regardless of the parameters ${x}'$, we focus on the right side of the inequality in Lemma 11, when $E[||\nabla_y f(x', y^*(x'), z)||^2]=O(\frac{1}{\sqrt{n}})$, all results of this paper remain valid. Roughly, we can assume that $f$ is Lipschitz continuity. (If you want to use covering number, this assumption is necessary, but we derive sharper bounds. In fact, we only need to assume that the norm is bounded solely on the sup parameters $x'$). In any case, we are open to further discussions.
>
>
> References
>
>
>
> Zhang Siqi, Hu Yifan, Zhang Liang and He Niao. Uniform Convergence and Generalization for Nonconvex Stochastic Minimax Problems. 2022.
>
> Li Shaojie and Liu Yong. Improved learning rates for stochastic optimization: Two theoretical viewpoints. 2021.
>
> Lei Yunwen, Yang Zhenhuan, Yang Tianbao and Ying Yiming. Stability and generalization of stochastic gradient methods for minimax problems. 2021.
>
>
> Xu Yunbei and Zeevi Assaf. Towards optimal problem dependent generalization error bounds in statistical learning theory. 2020.

---

### Official Review · Reviewer_KhvS · 2023-10-27

**Soundness:** 3 good
**Presentation:** 3 good
**Contribution:** 2 fair
**Rating:** 5
**Confidence:** 4

**Summary:**

Post rebuttal comment: I would raise the score to 5.

===================
This paper investigates stochastic minimax optimization, focusing on high-probability generalization performance gauged by primal function gradients. It extends prior work by replacing Lipschitz continuity requirements with Bernstein conditions in NC-SC setting, broadening the SC-SC setting to PL-SC setting, demonstrating the generalization bounds for GDA and GDmax in SC-SC setting, and generalizing results from minimization problems to minimax cases where the optimal error is low.

**Strengths:**

The paper extends prior work by replacing Lipschitz continuity requirements with Bernstein conditions in NC-SC setting, broadening the SC-SC setting to PL-SC setting, demonstrating the generalization bounds for GDA and GDmax in SC-SC setting, and generalizing results from minimization problems to minimax cases where the optimal error is low.

**Weaknesses:**

The paper lacks a cohesive narrative, as the extensions presented are not tightly interconnected and could be considered in isolation. Specifically, Section 5 appears unrelated to the preceding sections, making the overall structure disjointed. This fragmentation complicates the assessment of the paper's technical contributions.

**Questions:**

1. How do the extensions contribute to a unified narrative on the generalization of minimax optimization?
2. What constitutes the paper's key technical contribution, and why is it challenging?
3. How does Theorem 6 differs from existing research, specifically [Lin et al., 2020]?
4. Why is the generalization error analysis limited to the SC-SC setting for GDA and SGDA, rather than extending to the NC-SC setting?
5. Given that the PL condition is a natural extension of strong convexity, how does the analysis in Section 5.1 differ from that in the SC-SC setting? Is a minor modification of SC-SC results sufficient?

---

> ### Author Response · Authors · 2023-11-23
> **Part I**
>
> Thank you for your review. We will answer all your questions.
>
> Question 1: How do the extensions ...
>
> Answer:  Apart from the state-of-the-art results, we introduce a novel generic chaining framework into minimax problems to relaxed the assumptions, propose a minimax version of Bernstein condition, use variance-type concentration inequality and address numerous challenges within the minimax problems, primarily related to the connections between inner and outer layers, the norm of the gradient or the sharper bounds.
>
>
>
>
> Question 2: What constitutes the paper's key technical contribution ...
>
> Answer: We have discussed in Introduction and Remark 6. [Zhang et al. 2022] firstly introduced a expectation generalization error for primal functions in minimax problems using complexity. Naturally, we want to create a high-probability version, preferably using local methods to introduce variance information and obtain a tighter upper bound. As Reviewer WZTS stats, we can certainly continue with the traditional localized approach and solve the problem with covering numbers. However, these technologies require additional bounded assumptions (Assumption 2), or need certain distributional assumptions for unbounded condition. For example, [Mei et al. 2018] introduced the 'Hessian statistical noise' assumption when using covering numbers. Fortunately, [Xu and Zeevi, 2020] developed a novel ``uniform localized convergence'' framework using generic chaining for the minimization problems and [Li and Liu, 2021] extended it to analyze stochastic algorithms.
>
> This novel framework can not only relax the bounded (or specific distribution) assumptions but also impose fewer restrictions on the surrogate function for the localized method, enabling us to design the measurement functional to achieve a sharper bound. Consequently, we introduce this remarkable framework into minimax problems. Our generalization bound uses weaker assumptions comparing with [Zhang et al. 2022] and is sharper in some conditions due to our utilization of variance information.
>
> Introducing this new framework into minimax problems is not straightforward. [Zhang et al. 2022] indeed established a connection between inner and outer layers with the loss of primal functions, but we need do this with a new generic chaining approach. Furthermore, while [Zhang et al. 2022] only needed to bound the error caused by the connection between inner and outer layer with $O(\frac{1}{\sqrt{n}})$. We need to introduce the variance for a sharper bound, necessitating the use variance-type concentration inequalities.
>
> Next, we turn to applications. Firstly, for a sharper excess risk bound, we need to introduce the PL-SC condition to establish a connection between excess risk and the gradient of primal functions, leading to our results for ESP. Unfortunately, for GDA and SGDA algorithms, we found that all the related optimization papers in minimax problems focused on the iteration complexity (or gradient complexity). As a result, they only require $E[||\nabla \Phi(x_T)||]$ for average of round $T$ with an expected outcome. We were compelled to derive the high probability empirical optimization bound ourselves using classical optimization methods under SC-SC conditions.
>
> Notice that even under SC-SC settings, achieving this for SGDA remains difficult. Drawing inspiration from [Lei et al. 2021]'s proof of Primal-Dual Risk optimization bound, we eventually derive the optimization bound for primal risk. Additionally, in nonconvex minimization problems, several results bound the $T$-th empirical optimization error with high probability. Their main approach involves constructing proper martingale difference sequences, which also presents a challenge in the minimax settings. This remains a topic for future research.
>
> The proofs for excess risks in applications differ from minimization problems and pose challenges. These challenges stem from errors in the inner and outer layers of minimax problems. Consequently, we can only achieve a result close to $O(\frac{1}{n^2})$.

---

> ### Author Response · Authors · 2023-11-23
> **Part II**
>
> Question 3: How does Theorem 6 differs from existing research, specifically [Lin et al., 2020]?
>
> Answer: [Lin et al., 2020] derived the empirical optimization error. Our paper in Theorem 6 derives the \emph{population} optimization error, which can be decomposed into the empirical optimization error and the generalization error. This definition is given in Preliminaries and our paper mainly studies the generalization error.
>
>
>
>
>
>
>
>
>
>
> Question 4: Why is the generalization error analysis limited to the SC-SC setting for GDA and SGDA ...
>
>
> Answer: We have discussed in above the definition of SC-SC assumption and Remark 10, although the generalization bounds we proved are in NC-SC settings, we require the SC-SC assumptions to derive the empirical optimization error bound of primal functions, to gain the high probability excess risk bound. Existing optimization results do not have a high probability optimization error bound under NC-SC conditions.
>
>
> Question 5: Given that the PL condition is a natural extension of strong convexity, how does ...
>
> Answer: On one hand, compared with the results of other works under stronger SC-SC conditions, we obtained the best results. On the other hand, although PL-SC condition is a natural extension of the SC-SC condition. Since PL condition does not necessarily satisfy convexity, properties related to convexity cannot be used in the proof process, which makes the proof difficult.
>
>
> Question 6: Issue in Weaknesses ``The paper lacks a cohesive narrative, as the extensions presented are not tightly ...''
>
> Answer: I believe our work is natural and connected. As we introduced in the introduction, we first introduce a novel ``uniform localized convergence'' framework using generic chaining to obtain better generalization error bounds. For further applications, we all know that excess error is a combination of generalization error and optimization error. So we hope to get better excess error through improved generalization error. Since there is currently no work about the high-probability optimization error under the NC-SC setting, we can only fall back on the excess error under the SC-SC setting. It is precisely due to the improvement of our generalization error that under the SC-SC condition, our excess error reaches $O(1/n^2)$, which is better than best existing results in SC-SC condition ($O(1/n)$) (See comparison in Table 1).
>
>
>
> References
>
> Zhang Siqi, Hu Yifan, Zhang Liang and He Niao. Uniform Convergence and Generalization for Nonconvex Stochastic Minimax Problems. 2022.
>
> Mei Song, Bai Yu and Montanari Andrea. The landscape of empirical risk for nonconvex losses. 2018.
>
> Xu Yunbei and Zeevi Assaf. Towards optimal problem dependent generalization error bounds in statistical learning theory. 2020.
>
> Li Shaojie and Liu Yong. Improved learning rates for stochastic optimization: Two theoretical viewpoints. 2021.
>
> Lei Yunwen, Yang Zhenhuan, Yang Tianbao and Ying Yiming. Stability and generalization of stochastic gradient methods for minimax problems. 2021.

---

> ### Comment · Reviewer_KhvS · 2023-12-01
> **Acknowledgement**
>
> I would like to thank the author for their detailed response. Given the answer to Q2, I am happy to raise the score to 5. I strongly encourage the authors to include the answer for Q2 into the revised version for better clarity.
>
> However, I am not fully convinced by the answers for other questions, particularly Q1 and Q6 as the extensions consider settings and techniques that are not strongly connected but rather developed in its own context.
>
> For Q3, I am pretty sure that lin et al 2020 have derived the population optimization error. See [Lin, Tianyi, Chi Jin, and Michael Jordan. "On gradient descent ascent for nonconvex-concave minimax problems." International Conference on Machine Learning. PMLR, 2020.]
>
> For Q5, the global convergence of several optimization algorithms (except for accelerated ones) in the SC setting usually only uses the PL condition. Lacking convexity does not seem to lead any major difficulties.

---

### Official Review · Reviewer_yKtG · 2023-11-02

**Soundness:** 3 good
**Presentation:** 2 fair
**Contribution:** 2 fair
**Rating:** 5
**Confidence:** 3

**Summary:**

The authors study generalization bounds for minimax problems. They provide high-probability bounds for the gradient of the primal function under a nonconvex strongly concave setting. They also provide a dimension-independent result if the outer layer satisfies PL condition. They apply their results to existing algorithms and establish sharp bounds for excess primal risk.

**Strengths:**

The authors have presented good theoretical results. The principal contribution appears to be the introduction of a high-probability variant (Theorem 1) of the generalization error bound, originally proposed by Zhang et al. (2022) in expectation. Additionally, Theorem 3 offers a dimension-independent bound, albeit with some additional assumptions.

**Weaknesses:**

The paper was challenging to follow due to its unclear presentation and the presence of cumbersome sentences. These issues disrupted the flow of the text and, on occasion, led to confusion.

- I encountered the term, "almost" $O(\cdot)$, in the paper, but I'm uncertain about its exact meaning. Does this imply that the order of certain elements is not calculated with high accuracy?
- Certain remarks in the paper are challenging to comprehend due to their presentation. For instance, Remark 5 suggests that specific quantities are small in Theorem 3 and (8), but it does not specify their orders or the precise conditions under which they are considered small.
- In Remark 8, it is stated that $\phi(x^*) = O(\frac{1}{n})$ is a common occurrence. Can you provide examples to illustrate this assertion?

It is not my intention to imply that these claims are untrue, but it would greatly enhance the paper's quality if they were substantiated mathematically, especially considering the predominantly theoretical nature of the contributions in the paper.

**Questions:**

Please check weaknesses.

---

> ### Author Response · Authors · 2023-11-23
>
> Thank you for your review. Here are the replies for all your questions.
>
> Question 1: I encountered the term, "almost" ...
>
> Answer: The expression``almost'' is used to describe the excess risk bound we finally get, which is $O(1/(n(n-c)))$ where $c$ is a constant depending on $\beta, \mu_x, \mu_y$ and $\log(1/\delta)$. We use this word to emphasize this constant term. When $n$ is large, it can be understood as $O(1/n^2)$, which is better than best previous result $O(1/n)$.
>
>
> Question 2: Certain remarks in the paper are challenging to comprehend due to their presentation ...
>
> Answer: When we refer to these terms as ''tiny'', we mean that the order can be $O(\frac{1}{n})$ under certain assumption. In fact, in application section (Section 5), we include the assumption that the optimal population primal function $\Phi(x^*) = O(\frac{1}{n})$ allowing us to derive the result to be approximately $O(\frac{1}{n^2})$. In this situation, these terms are actually ''tiny''.
>
> Question 3: In Remark 8, it is stated that ...
>
> Answer: Thanks for pointing that out. The assumption of an optimal population primal function $\Phi(x^*) = O(\frac1n)$ is common and weak. This assumption is natural as $F(x^*, y^*)$ represents the minimal population risk and many researches such as (Srebro et al, 2010; Zhang et al, 2017; Lei and Ying, 2020) were used to achieve tighter bound.
>
> References
>
> Nathan Srebro, Karthik Sridharan, and Ambuj Tewari. Optimistic rates for learning with a smooth loss. 2010.
>
> Lijun Zhang, Tianbao Yang, and Rong Jin. Empirical risk minimization for stochastic convex optimization. 2017.
>
> Yunwen Lei and Yiming Ying. Fine-grained analysis of stability and generalization for stochastic gradient descent. 2020.

---

### Official Review · Reviewer_PvDp · 2023-11-06

**Soundness:** 3 good
**Presentation:** 2 fair
**Contribution:** 2 fair
**Rating:** 5
**Confidence:** 4

**Summary:**

In this work, authors introduce a new framework of uniform localized convergence based on Bernstein condtion, which allows them to derive sharper high-probability generalization bounds for these problems. Their approach relaxes some standard assumptions, such as Lipschitz continuity, and provides dimension-independent results under certain conditions.

**Strengths:**

First high-probability convergence result compared to existing literature. Second, the paper relaxes some restrictive assumptions in the literature.

**Weaknesses:**

The Bernstein condition introduces an intriguing alternative to the usual bounded gradient assumption, but as far as I can see, here the new condition is more like a bounded (higher-order) moment setting while it is required to hold for any moment.
1. I'm intrigued by the role of $n$ here in your Assumption 3. Should we interpret it as the sample size? If that's the case, it would require practitioners to verify a prohibitive number of moments to ensure compliance with the assumption, and additionally, to define the parameter $B$.
2. Are there more examples, beyond the one in Remark 2, where the Bernstein condition is met but not the bounded gradient norm? It would be beneficial to see this condition applied in more varied contexts.

In comparison with Zhang et al., 2022, the authors claim a more refined result, yet this advantage is only evident only when $||x-x^*||\leq 1/n$, which basically suggests the point $x$ must be exceedingly close to the optimum.

3. Regarding the potentially very large sample size, I concerned that the feasibility of such a condition seems questionable, especially with large sample sizes, potentially diminishing the practical relevance of the results.
4. In the NC-SC setting, the optimal point $x^*$ may not be unique (should be a set $X^*$), while here the authors implicitly assume the uniqueness, I think a better notation can be $\text{dist}(x, X^*)$.

In conclusion, the paper ambitiously addresses a broad spectrum of minimax challenges. While the primary contributions could be articulated more clearly, and the practical application seems somewhat elusive. But I am open for further discussions. Thank you.

**Questions:**

See weakness.

---

> ### Author Response · Authors · 2023-11-23
>
> Thank you for your time and thoughts. Below, we reply to all the issues.
>
> Question 1: The Bernstein condition introduces an intriguing alternative to the usual bounded gradient assumption, but as far as I can see ...
>
> Answer: Bernstein condition is like a ($p$-th order) moment of random variable can be bounded by its second order moment. Since its second order moment can be unbounded, higher-order moment is also unbounded. Bernstein condition is a standard assumption when using Bernstein's inequality. This inequality considers the variance of random variables before the $\sqrt{\frac{1}{n}}$ term. When the variance is small, this term can be omitted. Thus, Bernstein condition and Bernstein's inequality are widely used to obtain sharper bounds.
>
>
> Question 1.1: I'm intrigued by the role of here in your Assumption 3. Should we interpret it as the sample size? If that's the case ...
>
> Answer: $n$ has nothing to do with sample size. Bernstein condition is like a ($p$-th order) moment of random variable can be bounded by its second order moment. We will change $n$ as $p$ in the finally version to avoid ambiguity.
>
> Question 1.2: Are there more examples, beyond the one in Remark 2 ...
>
> Answer: Yes, we have discussed in Remark 1 when introducing standard Bernstein condition.      ``Bernstein condition is milder than the bounded assumption of random variables and is also satisfied by various unbounded variables. For example, a random variable is sub-exponetial if it satisfies Bernsteion condition (Wainwright, 2019).''
>
>
>
> Question 2: In comparison with Zhang et al., 2022, the authors claim a more refined result ...
>
> Answer: This judgment that this advantage is only evident only when $||x_x^*|| \leq 1/n$ is not accurate. Zhang et al. [2022] considers the \emph{worst} parameters in the parameter space. However, in our case, whenever $x$ is closer to $x^*$, our results will be better. For example, when $||x_x^*|| \leq 1/\sqrt{n}$, our results is of order $O(1/n)$ comparing with $O(1/\sqrt{n})$ Zhang et al. [2022]. When $||x_x^*|| \leq 1/n$, our results is of order $O(1/\sqrt{n^3})$. The optimization algorithm in real scene will also make $x$ as close as possible to $x^*$, rather than choosing the worst result of the entire parameter space, so our generalization bound is more meaningful.
>
> Question 2.1: Regarding the potentially very large sample size, I concerned that ...
>
> Answer: Theorem 1 (our main Theorem) don't need the potentially very large sample size. For Theorem 3, large sample size is assumed to remove the dependence on the dimension $d$. Of course, we can also relax this assumption of sample size, but the final result will have dimension $d$.
>
>
>
> Question 2.2: In the NC-SC setting, the optimal point ...
>
>
> Answer:  Thanks for your suggestion. We will revise in the final version.
>
> References
>
> Zhang Siqi, Hu Yifan, Zhang Liang and He Niao. Uniform Convergence and Generalization for Nonconvex Stochastic Minimax Problems. 2022.

---

### Meta-Review · Area_Chair_sezM · 2023-12-11

**Metareview:**

to be filled

**Justification For Why Not Higher Score:**

to be filled

**Justification For Why Not Lower Score:**

N/A

---

### Decision · Program_Chairs · 2024-01-16

Reject